



# Inversely modeling homogeneous H$_2$SO$_4$-H$_2$O nucleation rate in exhaust-related conditions

Miska Olin[1], Jenni Alanen[1], Marja R.T. Palmroth[2], Topi Rönkkö[1], and Miikka Dal Maso[1]

[1]Aerosol Physics, Laboratory of Physics, Faculty of Natural Sciences, Tampere University of Technology, P.O. Box 692, 33101 Tampere, Finland
[2]Laboratory of Chemistry and Bioengineering, Faculty of Natural Sciences, Tampere University of Technology, P.O. Box 541, 33101 Tampere, Finland

**Correspondence:** Miska Olin (miska.olin@tut.fi)

**Abstract.** Homogeneous sulfuric acid-water nucleation rate in conditions related to vehicle exhaust was measured and modeled. The measurements were performed by evaporating pure sulfuric acid and water liquids and by diluting and cooling the sample vapor with a sampling system mimicking the dilution process occurring in a real-world driving situation. The nucleation rate inside the measurement system was modeled inversely using CFD (computational fluid dynamics) and the aerosol
dynamics code, CFD-TUTMAM (Tampere University of Technology Modal Aerosol Model for CFD). The nucleation exponents for the concentrations of sulfuric acid and water and for the saturation vapor pressure of sulfuric acid were found to be $1.9 \pm 0.1$, $0.50 \pm 0.05$, and $0.75 \pm 0.05$, respectively. With these exponents, nucleation rate can be expressed with a function of the concentrations of sulfuric acid and water and of temperature. Results imply that the nucleation process of volatile nanoparticles in real vehicle exhaust cannot be fully explained by sulfuric acid; instead, it is likely that other compounds, e.g.,
hydrocarbons, are involved as well. In general, the obtained nucleation rate function can be used to examine the nucleation mechanisms occurring in exhaust from different combustion sources (internal combustion engines, power plant boilers, etc.) or in the atmosphere and, furthermore, to improve air quality models.

## 1 Introduction

Airborne particles are related to adverse health effects (Dockery et al., 1993; Pope et al., 2002; Beelen et al., 2014; Lelieveld et al., 2015) and various effects on climate (Arneth et al., 2009; Boucher et al., 2013). In particular, adverse health effects are caused by the exposure to vehicle emissions which increase ultrafine particle concentration in urban air (Virtanen et al., 2006; Johansson et al., 2007; Pey et al., 2009) in the size range with high probability of lung deposition (Alföldy et al., 2009; Rissler et al., 2012).

Vehicles equipped with internal combustion engines generate nonvolatile particles (Rönkkö et al., 2007; Sgro et al., 2008; Maricq et al., 2012; Rönkkö et al., 2014; Chen et al., 2017); however, volatile particles are also formed after the combustion



process during exhaust cooling (Kittelson, 1998; Lähde et al., 2009), i.e., when the exhaust is released from the tailpipe. Thus, volatile particles are formed through nucleation process; hence, they are called here nucleation mode particles.

An important characteristic of fine particles is the particle size distribution, as it determines the behavior of particles in the atmosphere and particle deposition to the respiratory system. Modeling studies provide information on the formation and

evolution of exhaust-originated particles in the atmosphere (Jacobson et al., 2005; Stevens et al., 2012). To model the number concentration and the particle size of nucleation mode, the governing nucleation rate needs to be known.

The detailed nucleation mechanism controlling particle formation in cooling and diluting vehicle exhaust is currently unknown (Keskinen and Rönkkö, 2010). The nucleation mode particles contain at least water, sulfuric acid ($H_2SO_4$), and hydrocarbons (Kittelson, 1998; Tobias et al., 2001; Sakurai et al., 2003; Schneider et al., 2005). Therefore, it is likely that these

compounds are involved in the nucleation process, but, on the other hand, some of them can end up in the nucleation mode through the initial growth of the newly-formed clusters. The most promising candidate for the main nucleating component in the particle formation process occurring in diesel exhaust is $H_2SO_4$, as it has been shown that the $H_2SO_4$ vapor concentration in vehicle exhaust (Rönkkö et al., 2013; Karjalainen et al., 2014), fuel sulfur content (Maricq et al., 2002; Vogt et al., 2003; Vaaraslahti et al., 2005; Kittelson et al., 2008), lubricating oil sulfur content (Vaaraslahti et al., 2005; Kittelson et al., 2008),

and exhaust after-treatment system (Maricq et al., 2002; Vogt et al., 2003) correlate with nucleation mode number concentration, at least in the cases when the test vehicle has been equipped with an oxidative exhaust after-treatment system. The sulfur contents of fuel and lubricating oil are connected to the $H_2SO_4$ vapor concentration in the exhaust because the combustion of sulfur-containing compounds produces sulfur dioxide ($SO_2$) that is further oxidized to sulfur trioxide ($SO_3$) in an oxidative exhaust after-treatment system (Kittelson et al., 2008), and $SO_3$ finally produces $H_2SO_4$ when contacting with water ($H_2O$)

vapor (Boulaud et al., 1977).

Particle formation due to $H_2SO_4$ in real vehicle exhaust plumes and in laboratory sampling systems has been previously simulated by several authors (Uhrner et al., 2007; Lemmetty et al., 2008; Albriet et al., 2010; Liu et al., 2011; Arnold et al., 2012; Li and Huang, 2012; Wang and Zhang, 2012; Huang et al., 2014), but all of them have modeled nucleation as binary homogeneous nucleation (BHN) of $H_2SO_4$ and water. Other possible nucleation mechanisms are activation-type (Kulmala et al.,

2006), kinetic (McMurry and Friedlander, 1979), hydrocarbon-involving (Paasonen et al., 2010), and ion-induced nucleation (Raes et al., 1986). The choice of binary homogeneous $H_2SO_4$-$H_2O$ nucleation in studies involving vehicle exhaust is mainly made because it has been the only nucleation mechanism for which an explicitly defined formula for the nucleation rate ($J$) can be presented (Keskinen and Rönkkö, 2010). An explicit definition is required when the nucleation rate in cooling exhaust is modeled, as the nucleation rate has a steep temperature-dependency according to theory (Hale, 2005) and experiments (Wölk

and Strey, 2001). The nucleation rate of BHN is derived from classical thermodynamics; thus, the theory is called the classical nucleation theory (CNT). The nucleation rate according to the CNT is explicitly defined as a function of $H_2SO_4$ and $H_2O$ vapor concentrations ($[H_2SO_4]$ and $[H_2O]$) and temperature ($T$). Conversely, the nucleation rates of the other nucleation mechanisms are typically modeled as (Zhang et al., 2012)

$$J = k[H_2SO_4]^n, \tag{1}$$





where $k$ is an experimentally derived coefficient and $n$ is the nucleation exponent presenting the sensitivity of $[H_2SO_4]$ on $J$. The value for $k$ is typically a constant that includes the effect of $T$ and $[H_2O]$, i.e., relative humidity (RH), (Sihto et al., 2009; Stevens and Pierce, 2014). A constant coefficient can be a satisfactory approximation in atmospheric nucleation experiments, where $T$ and RH remain nearly constants. However, $T$ and RH in a cooling and diluting exhaust are highly variable; thus, a

constant coefficient cannot be used.

The derivation of the CNT contains, however, a lot of assumptions and it is thus quite uncertain (Vehkamäki and Riipinen, 2012). The largest uncertainty rises from the capillarity approximation, i.e. the physical properties of small newly-formed critical clusters can be expressed as the properties of bulk liquid (Wyslouzil and Wölk, 2016). Comparing experimental and theoretical nucleation rates, the CNT underestimates the temperature-dependency (Hung et al., 1989) and overestimates the

sensitivity of $[H_2SO_4]$ on $J$ (Weber et al., 1996; Olin et al., 2014). These discrepancies entail that theoretically derived nucleation rates need to be corrected with a factor, ranging in several orders of magnitude, to agree with experimental nucleation rates. The nucleation exponents, $n$, for $H_2SO_4$ obtained from the atmospheric nucleation measurements (Sihto et al., 2006; Riipinen et al., 2007) and from the atmospherically-relevant laboratory experiments (Brus et al., 2011; Riccobono et al., 2014) lie usually between $n = 1$ and $n = 2$, which are much lower than the theoretical exponents ($n \gtrsim 5$).

The first attempts to obtain the nucleation rate in vehicle exhaust, other than predicted by the CNT, were performed by Olin et al. (2015) and Pirjola et al. (2015). These modeling studies are based on the same experiments (Arnold et al., 2012; Rönkkö et al., 2013) where the exhaust of a diesel engine was sampled using a laboratory setup containing an engine dynamometer and a diluting sampling system (Ntziachristos et al., 2004). Modeling in these studies was performed inversely, i.e. an initial function for $J$ acts as an input to the model and is altered until the simulated particle concentration and distribution correspond

to the measured ones. Inverse modeling is a preferable method in examining nucleation rates in a diluting domain over the method based on calculating $J$ by dividing the measured number concentration with an estimated volume of a nucleation region, because the volume of a nucleation region depends on $n$ also. In the case of inverse modeling, there is no need to estimate the nucleation region because the model simulates $J$ at every time step, in a model using temporal coordinates, or in every computational cell, in a model using spatial coordinates. Pirjola et al. (2015) modeled the dilution system with an

aerosol dynamics model using temporal coordinates and concluded that hydrocarbons could be involved in the nucleation mechanism, and $n$ lies between $n = 1$ and $n = 2$. However, because particle formation in diluting vehicle emission involves strong gradients in temperature and the concentrations of the compounds involved, information in spatial dimensions is also required to fully understand the particle formation process. For this reason, Olin et al. (2015) simulated aerosol dynamics using computational fluid dynamics (CFD) and concluded that $n$ is 0.25 or 1, depending on the measurement set. These values are

very low compared to other studies and to the first nucleation theorem (Kashchiev, 1982) that implies $n$ is at least 1. Values below unity imply that there can be other compounds involved in the nucleation mechanism in addition to $H_2SO_4$.

Ammonia ($NH_3$) can also be involved in $H_2SO_4$-$H_2O$ nucleation if the $H_2SO_4$ concentration is low and the $NH_3$ concentration is high (Lemmetty et al., 2007; Kirkby et al., 2011). The $H_2SO_4$ concentration in the atmosphere is low enough for the effect of $NH_3$ to be relevant (Kirkby et al., 2011), but in vehicle exhaust higher $H_2SO_4$ concentrations make the effect of $NH_3$

probably negligible. However, more recent vehicles are equipped with the selective catalytic reduction (SCR) system which




decreases nitrogen oxide emissions but, on the other hand, increases $NH_3$ emissions. Therefore, $NH_3$ can be involved in the nucleation process occurring in vehicle exhaust of vehicles equipped with the SCR system (Lemmetty et al., 2007). The SCR system was not included in the experiments of Arnold et al. (2012) and Rönkkö et al. (2013) mentioned earlier; thus, other compounds involving in the nucleation process in those experiments are more likely hydrocarbons rather than $NH_3$.

In this paper, an improved aerosol dynamics model, CFD-TUTMAM (Tampere University of Technology Modal Aerosol Model for CFD), based on our previous model, CFD-TUTEAM (Tampere University of Technology Exhaust Aerosol Model for CFD) described in the reference Olin et al. (2015), is presented. The main improvement in the model is its capability to model the initial growth of the newly-formed clusters modally using our novel representation of the particle size distribution, the PL+LN (combined power law and log-normal distribution) model described in the reference Olin et al. (2016). Laboratory

experiments designed for nucleation rate modeling purposes are presented, in which the examination of the nucleation rate was aimed towards pure $H_2SO_4$-$H_2O$ nucleation instead of nucleation associated with some unknown compounds existing in real vehicle exhaust. Neglecting these unknown compounds is reasonable at this stage of nucleation studies because the knowledge of the nucleation mechanism of the pure $H_2SO_4$-$H_2O$ nucleation is still at a very low level, which should be examined more to better understand the nucleation process in a real vehicle exhaust.

A pure $H_2SO_4$-$H_2O$ nucleation was generated by evaporating pure $H_2SO_4$ and $H_2O$ liquids and using the dilution system that mimics a real-world dilution process of a driving vehicle (Ntziachristos et al., 2004). A similar principle of generating $H_2SO_4$ by evaporating it from a saturator has been used in the study of Neitola et al. (2015), in which the concentrations of $H_2SO_4$ and $H_2O$ and temperatures were kept in an atmospherically-relevant range. In this study, they were kept in a vehicle exhaust-relevant range; thus, the output is an explicitly defined formula for the $H_2SO_4$-$H_2O$ nucleation rate in exhaust-related

conditions. The formula is in the form of

$$J\left([H_2SO_4],[H_2O],T\right) = k\frac{[H_2SO_4]^{n_{sa}}[H_2O]^{n_w}}{p_{sa}{}^\circ(T)^{m_{sa}}}, \tag{2}$$

which is based on the formula hypothesized by Olin et al. (2015) but with an additional exponent $m_{sa}$ for the saturation vapor pressure of sulfuric acid ($p_{sa}{}^\circ$). In Eq. (2), $n_{sa}$ and $n_w$ represent the nucleation exponents for $[H_2SO_4]$ and $[H_2O]$, respectively. The exponents may also depend on the concentration levels but due to the unknown dependency, only constant values are

considered in this study.

The formulation obtained from this study helps in finding the nucleation mechanisms occurring in real vehicle exhaust or in the atmosphere. Similarly, it can be used to examine particle formation in coal-fired power plant exhaust, which is known to contain $H_2SO_4$ too (Stevens et al., 2012). Another use of the formulation is in studying the effect of $H_2SO_4$ on the urban air quality by using it in air quality models.

## 2   Laboratory experiments

Laboratory experiments were designed to enable the examination of the effects of three parameters ($[H_2SO_4]$, $[H_2O]$, and $T$) on the $H_2SO_4$-$H_2O$ nucleation rate. The experimental setup is presented in Fig. 1.



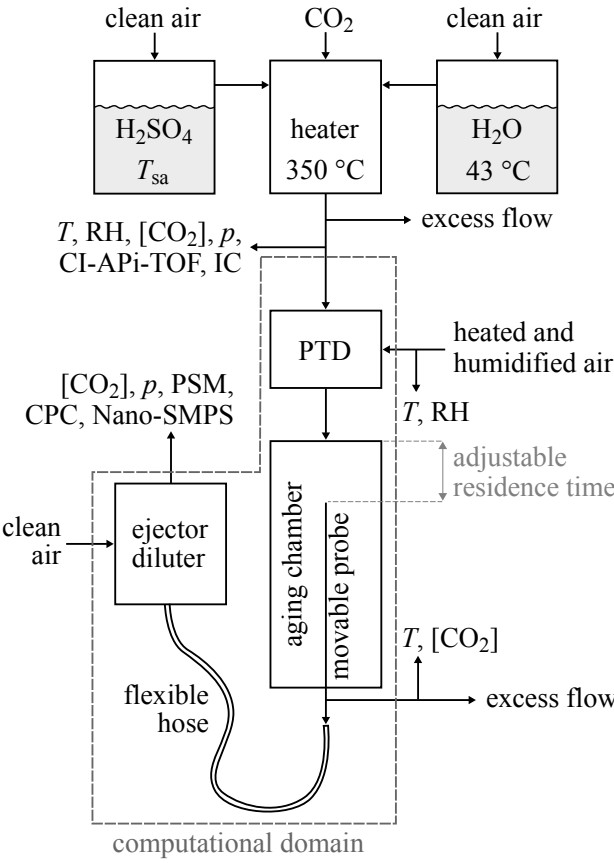

**Figure 1.** The experimental setup used to generate artificial exhaust and sample it with a diluting sampling system. The top part of the figure represents the artificial raw exhaust generation, which contains mixing and heating $H_2SO_4$ and $H_2O$ vapors evaporated from pure liquids. The bottom part of the figure represents the raw exhaust sampling system, which consists of a porous tube diluter (PTD), an aging chamber, and an ejector diluter (ED). The computational domain of the CFD simulation is also shown in the figure.

## 2.1 Artificial raw exhaust generation

The artificial raw exhaust sample was generated (the top part of Fig. 1) by evaporating 98 % $H_2SO_4$ liquid and deionized Milli-Q water. $H_2SO_4$ was held in a PTFE container and water in a glass bottle. The liquids were heated to temperatures $T_{sa}$ and 43 °C, respectively, which determine the concentrations in the gas phase theoretically through the saturation vapor pressure. Dry and filtered compressed air was flown through the evaporators and mixed before heating to 350 °C. 2.7 % of carbon dioxide ($CO_2$) was also mixed with a sample to act as a tracer to determine the dilution ratio (DR) of the diluters. $CO_2$ was selected because it has no effect on the particle formation process and because it exists in real exhaust as well.

The computational domain in the CFD simulation shown in the bottom part of Fig. 1 begins before the sample enters to the PTD; thus, the concentrations of $H_2SO_4$ and $H_2O$, temperature, pressure ($p$), and flow rate need to be known at that point





due to the requirement of the boundary conditions in the CFD simulation. $T$ and $p$ were measured at that point, $[H_2O]$ was calculated from the measured RH, and the flow rate was calculated from the DR of the porous tube diluter (PTD) with the aid of measured $CO_2$ concentrations.

The temperature of the raw sample was 243 °C and the mole fraction of $H_2O$ ($x_w$) was 0.036, in average. Temperature before the PTD was lower than the heater temperature, 350 °C, because the sample cooled in the sampling lines, but the temperature of 243 °C corresponds well with the temperature of real exhaust when released from the tailpipe. In NTP conditions, $x_w = 0.036$ corresponds to $[H_2O] = 9.0 \times 10^{17}\,\mathrm{cm^{-3}}$. The mole fractions in real diesel or gasoline exhaust range between 0.06 and 0.14, but the values higher than 0.036 with this experimental setup were not used, because a more humid sample caused the water vapor to condense as liquid water in the sampling lines.

The temperature of the $H_2SO_4$ evaporator, $T_{sa}$, was varied between 85 °C and 164.5 °C which correspond to the mole fractions ($x_{sa}$) between $2.2 \times 10^{-7}$ and $1.1 \times 10^{-5}$ in the raw sample. In NTP conditions, this range corresponds to the $[H_2SO_4]$ values between $5.7 \times 10^{12}\,\mathrm{cm^{-3}}$ and $2.8 \times 10^{14}\,\mathrm{cm^{-3}}$. These concentrations are higher than concentrations in real vehicle exhaust (typically between $10^8\,\mathrm{cm^{-3}}$ and $10^{14}\,\mathrm{cm^{-3}}$), because particle formation was not observed with the concentrations below $5.7 \times 10^{12}\,\mathrm{cm^{-3}}$. However, with real vehicle exhaust, in the same sampling system used here, particle formation has been observed even with the concentration of $2.5 \times 10^9\,\mathrm{cm^{-3}}$ (Arnold et al., 2012), indicating other compounds are probably involved in the nucleation process.

The determination of $[H_2SO_4]$ in the raw sample in our experiment was not straightforward due to the uncertainties involved in the measurement of $[H_2SO_4]$. The detailed information on measuring it is described in Sec. 2.2. Estimating $[H_2SO_4]$ theoretically through the saturation vapor pressure in the temperature of $T_{sa}$ provides some information on the dependency of $T_{sa}$ on $[H_2SO_4]$ in the raw sample. However, the absolute concentrations cannot be satisfactorily estimated due to high and uncertain diffusional losses of $H_2SO_4$ onto the sampling lines between the $H_2SO_4$ evaporator and the PTD. High losses are caused by a low flow rate from the $H_2SO_4$ evaporator (0.5 slpm) connected with high diffusion coefficient of $H_2SO_4$. The diffusional losses according to the equations reported by Gormley and Kennedy (1948) and to the humidity-dependent diffusion coefficient of $H_2SO_4$ reported by Hanson and Eisele (2000) are 98 % if the walls of the sampling lines are assumed fully condensing. However, some parts in the sampling lines between the evaporator and the PTD have high concentrations of $H_2SO_4$ with high temperature, especially with high $T_{sa}$ values. Therefore, these lines are probably partially saturated with $H_2SO_4$, which can act as preventing $H_2SO_4$ condensation onto the walls. Thus, the actual diffusional losses are estimated to be between 0 and 98 % and they can also depend on $T_{sa}$ and on the saturation status of the sampling lines during a previous measurement point. Therefore, the determination of $[H_2SO_4]$ in the raw sample was done through inverse modeling using measured particle diameter information (see Sec. 4.5). The output of the concentrations from inverse modeling denote the diffusional losses of 43 ... 95 % depending on $T_{sa}$.

## 2.2 Sulfuric acid measurement

The concentration of $H_2SO_4$ vapor in the raw sample was varied by altering the $H_2SO_4$ evaporator temperature and measured with a nitrate ion ($NO_3^-$) based chemical ionization Atmospheric Pressure interface Time-Of-Flight mass spectrometer (CI-





APi-TOF, Jokinen et al. (2012)) and Ion Chromatography (IC, Sulonen et al. (2015)). The measurements of $[H_2SO_4]$ were performed separately from the actual particle measurements due to the reduced availability of the CI-APi-TOF instrument and the required high sample flow rate and sampling time of the IC measurement.

The CI-APi-TOF used $NO_3^-$ ions as reagent ions to detect $H_2SO_4$ as bisulfate ions ($HSO_4^-$) and their clusters with nitric acid
($HNO_3$) in an APi-TOF mass spectrometer (Tofwerk AG, Switzerland and Aerodyne Research Inc., USA). The CI-APi-TOF outputs the concentrations of the measured ions as counts per second ($c_{ion}$) which need to be converted to absolute $H_2SO_4$ concentrations with the equation (Tröstl et al., 2016)

$$[H_2SO_4] =$$
$$C \cdot \ln \left( 1 + \frac{c_{HSO_4^-} + c_{HNO_3 \cdot HSO_4^-} + c_{H_2SO_4 \cdot HSO_4^-}}{c_{NO_3^-} + c_{HNO_3 \cdot NO_3^-} + c_{(HNO_3)_2 \cdot NO_3^-}} \right), \tag{3}$$

where $C$ is an experimentally determined calibration coefficient having the value of $1.3 \times 10^9 \, cm^{-3}$ for the device used. The CI-APi-TOF works well in measuring $H_2SO_4$ from the atmosphere; however, because the concentrations in this experiment were significantly higher, the raw sample needed to be diluted. The sample flow rate to the CI-APi-TOF was $(10.0 \pm 0.2)$ slpm and it was prepared by diluting the raw sample with compressed air heated to $300\,°C$ with the flow rate of almost 10 slpm. The DR, determined using $CO_2$ measurement, was $133 \pm 7$. This corresponds to the raw sample flow rate of $(0.075 \pm 0.004)$
slpm. The length of the sampling line before the dilution point was 70 mm and between the dilution point and the inlet of the CI-APi-TOF it was 1720 mm. According to the diffusional losses, only the fraction of $(6 \pm 2) \times 10^{-4}$ of $H_2SO_4$ penetrated to the CI-APi-TOF inlet, of which the major contribution resulted from the sampling line before the dilution point having very low flow rate.

The IC measurement was performed by sucking the raw sample with the flow rate of $(2.76 \pm 0.02)$ slpm through a gas
washing bottle with a fritted disc and analyzing $SO_4^{2-}$ ion concentration from the liquid sample with the IC instrument off-line. The length of the sampling line before the washing bottle was 525 mm, for which the calculated penetration due to diffusional losses is $(20.4 \pm 0.4)\,\%$. The effect of the line length on the diffusional losses was examined using also a sampling line having the length of 750 mm, for which the calculated penetration is $(12.9 \pm 0.4)\,\%$. However, according to the measured $[SO_4^{2-}]$, the line length had no effect on the penetrated fraction, implying over-predicted diffusional losses in the first part of the sampling
line calculated with the equations of Gormley and Kennedy (1948), probably due to saturating $H_2SO_4$ liquid onto the sampling lines. The gas washing bottle was filled with 130 ml of deionized Milli-Q water and the gas collecting time was 20 ... 360 min, depending on the expected $[H_2SO_4]$ in the raw sample. The collection efficiency of the gas washing bottle was measured by collecting the sample also with 80 ml of water having approximately half the bubbling height of 130 ml of water. According to the measured $[SO_4^{2-}]$, the amount of water had no effect on the results; thus, the collection efficiency was high, or at least the
maximum achievable in the measurement conditions.





## 2.3    Raw exhaust sampling system

The sampling system used to dilute and cool the raw exhaust, presented in the bottom part of Fig. 1, was a modified partial flow sampling system (Ntziachristos et al., 2004) mimicking the dilution process occurring in a real-world driving situation. It consists of a PTD, an aging chamber, and an ejector diluter (ED). The PTD dilutes and cools the sample rapidly, which leads to new particle formation. The aging chamber is used to grow the newly-formed particles to detectable sizes and to continue the nucleation process. The ED is used to stop the particle formation and growth processes and to obtain the conditions of the sample required for measurement devices.

Dilution air used with the PTD and the ED was filtered compressed air. The ED used only dry ($RH \approx 3.6\%$) and unheated ($T \approx 20\,°C$) dilution air, but the dilution air for the PTD was humidified ($RH_{PTD} = 2 \ldots 100\%$) and heated ($T_{PTD} = 27.5 \ldots 70\,°C$). Humidifying the dilution air of the PTD was done by directing the compressed air flow through a container filled with deionized Milli-Q water. $RH_{PTD}$ and $T_{PTD}$ are the variable parameters used in examining the effect of $[H_2O]$ and $T$ on $J$, which represent the conditions of the outdoor air acting in a dilution process in a real-world driving situation. The range of $T_{PTD}$ represent higher temperatures compared to the temperature of the outdoor air, but lower temperatures were not used because $27.5\,°C$ was the coldest temperature available with the laboratory setup having no cooling device.

In this experiment, the residence time in the aging chamber was made adjustable by a movable sampling probe inside the aging chamber. The sampling probe was connected to the ED with a flexible Tygon hose. The residence time from before the PTD to after the ED was altered within a range of $1.4 \ldots 2.8$ s. Using a movable probe to alter the residence time has only a minor effect on the flow and temperature fields compared to altering the residence time with changing the flow rate in the aging chamber. Maintaining constant flow and temperature fields when studying the effect of the residence time is important, because variable fields would alter the turbulence level and temperatures in the aging chamber, both having effects on the measured particle concentration and thus causing difficulties to separate the effect of the residence time from the effect of turbulence or temperature on measured particle concentrations.

The DR of the PTD ($DR_{PTD}$) was controlled by the excess flow rate after the aging chamber and calculated by the measured $[CO_2]$ before the PTD and after the aging chamber. The $DR_{PTD}$ was kept around 20 in all measurements. The DR of the ED ($DR_{ED}$) was controlled by the pressure of the dilution air used with the ED and calculated also using $CO_2$ measurements. The calculated $DR_{ED}$ was around 10. Because the dilution ratios varied between different measurement points, all the aerosol results are multiplied with the total DR thus making the results comparable.

## 2.4    Particle measurement

Particle number concentration and size distribution was measured after the ED using Airmodus PSM A11 (Particle Size Magnifier A10 using Airmodus Condensation Particle Counter A20 as the particle counter), TSI CPC 3775 (Ultrafine Condensation Particle Counter), and TSI Nano-SMPS (Nano Scanning Mobility Particle Sizer using TSI CPC 3776 as the particle counter). The PSM and the CPC 3775 measure the particle number concentration ($N_{PSM}$ and $N_{CPC}$) by counting particles with diameters larger than $\sim 1.15$ nm (PSM) or $\sim 2.15$ nm (CPC 3775). The $D_{50}$-cut-size (the particle diameter having the detection





efficiency of 50 %) of the PSM can be altered, by adjusting its saturator flow rate, within the diameter range of 1.3 ... 3.1 nm. Additionally, the CPC 3775 has the $D_{50}$-cut-size of 4.0 nm and the CPC 3776 of 2.4 nm. The detection efficiency curves of the particle counter used are presented in Fig. 2. The Nano-SMPS measured, with the settings used in this experiment, the particle size distribution within the diameter range of 2 ... 65 nm; however, the detection of particles having diameters smaller than $\sim 7$ nm are weakly detectable due to high diffusional losses of very small particles inside the device. Nevertheless, using the data from the different saturator flow rates of the PSM together with the data from the CPC 3775, information on the particle size distribution around the range of 1.15 ... 7 nm is also obtained.

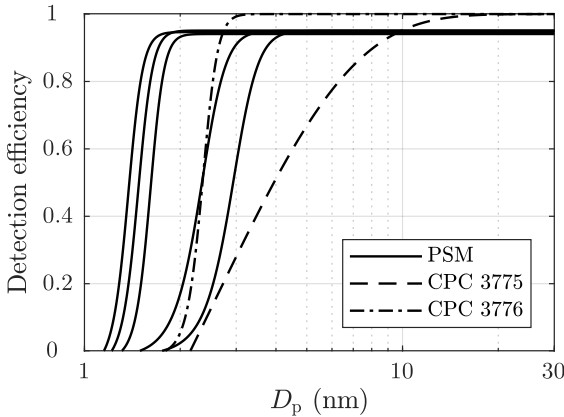

**Figure 2.** The detection efficiencies of the PSM, with five different saturator flow rates used in this experiment, and of the CPCs. The curves are exponential fittings based on the detection efficiencies reported by the manufacturers of the devices.

Due to too high particle number concentration for the PSM, aerosol measured with the PSM and the CPC 3775 was diluted with a bridge diluter (BD). It dilutes the concentration of larger particles ($D_{\mathrm{p}} > 10\,\mathrm{nm}$) with the ratio of 250, but the DR increases with decreasing particle size due to diffusional losses, finally to the ratio of 1200 ($D_{\mathrm{p}} = 1.15\,\mathrm{nm}$). The DR was measured with aerosol samples having the count median diameters (CMDs) of 2 ... 25 nm. The ratio of the sampling line length and the flow rate of the BD, a partially unknown variable, used in the diffusional losses function reported by Gormley and Kennedy (1948) was fitted to correspond with the DR measurement results; the obtained DRs are presented in Fig. 3.

## 2.5 Measurement sets

By varying [$H_2SO_4$] of the artificial raw exhaust sample and [$H_2O$] and $T$ of the dilution air separately and measuring the aerosol formed in the sampling system, the effects of the parameters on $J$ can be examined. The effects of the parameters are included in Eq. (2) simply with the exponents $n_{\mathrm{sa}}$, $n_{\mathrm{w}}$, and $m_{\mathrm{sa}}$. To obtain these three yet unknown values, at least three parameters were required to be varied in the experiments. Nevertheless, a fourth parameter, the residence time, was also varied to provide some validation for the obtained exponents. [$H_2O$] and $T$ of the dilution air were varied simply by humidifying and heating the dilution air flowing to the PTD and measuring RH and $T$ from the dilution air. Varying [$H_2SO_4$] of the raw sample





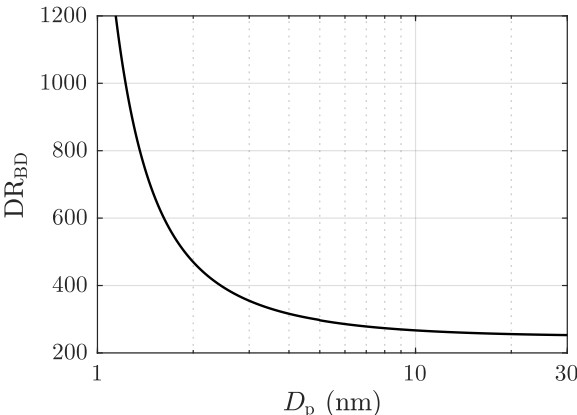

**Figure 3.** The dilution ratio of the bridge diluter with different particle diameters.

was done by varying $T_{sa}$. However, determining $[H_2SO_4]$ in the raw sample was more problematic due to the uncertainties involved in the measurement of $[H_2SO_4]$.

The varied conditions of the measurements are presented in Tab. 1, in which all the measurement points are divided according to the main outputs ($n_{sa}$, $n_w$, $m_{sa}$, and $\partial J/\partial t$) that measurement sets were designed to provide. Examining the effect of temperature ($m_{sa}$) was performed with the measurements of two types: varying $T_{PTD}$ while keeping $RH_{PTD}$ as a constant (Set 3a) and varying $T_{PTD}$ while keeping the mole fraction of $H_2O$ in the dilution air of the PTD ($x_{w,PTD}$) as a constant (Set 3b). The time-dependence of the nucleation rate ($\partial J/\partial t$) or, in the other words, the diminishment rate of $J$ in a diluting sampling system is mainly the product of the exponents $n_{sa}$ and $m_{sa}$ in the following way: $[H_2SO_4]$ decreases steeply due to dilution, losses to walls, and condensation to particles resulting in diminishing $J$ with the power of $n_{sa}$; simultaneously $T$ decreases due to dilution and cooling of the sampling lines resulting in strengthening $J$ with the power of $m_{sa}$. Examining the diminishment rate provides validation for the relation of $n_{sa}$ and $m_{sa}$ obtained from the simulations. We waited 2 ... 40 min for the particle size distributions to stabilize after the conditions was changed between the measurement points. When the particle formation process was satisfactorily stabilized, measurement data for each measurement point were recorded for 5 ... 40 min, depending on the stability of the particle generation.

## 3   Experimental results

### 3.1   Sulfuric acid concentrations

$H_2SO_4$ concentrations in the raw sample with different $H_2SO_4$ evaporator temperatures measured with the CI-APi-TOF and the IC are presented in Fig. 4. The values of $[H_2SO_4]_{raw}$ are presented as the values in NTP conditions rather than in a hot raw sample. The lossless theoretical curve represents $[H_2SO_4]_{raw,NTP}$ if diffusional losses onto the sampling lines between the evaporator and the $[H_2SO_4]_{raw}$ measurement point are neglected. Surprisingly, the CI-APi-TOF data are at a somewhat higher





**Table 1.** The varied conditions of the measurement points.

| Set | Main output | $T_{sa}$ (°C) | $T_{PTD}$ (°C) | $x_{w,PTD}$ ($10^{-3}$) | $RH_{PTD}$ (%) | Residence time (s) |
|---|---|---|---|---|---|---|
| 1 | $n_{sa}$ | 85 ... 164.5 | 27.5 | 7.7 | 22 | 2.8 |
| 2 | $n_{w}$ | 150 | 30 | 0.7 ... 42 | 2 ... 100 | 2.8 |
| 3a | $m_{sa}$ | 150 | 30 ... 70 | 9 ... 65 | 22 | 2.8 |
| 3b | $m_{sa}$ | 150 | 30 ... 70 | 44 | 22 ... 100 | 2.8 |
| 4 | $\partial J/\partial t$ | 135.5 ... 164.5 | 27.5 | 7.7 | 22 | 1.4 ... 2.8 |

level compared to the lossless level which is probably partially accounted by the calculated diffusional losses between the measurement point and the device, which have a large uncertainty due to very low sample flow rate. The IC data are at the level of about 5 % of the theoretical level, but there are also significant outliers at a higher level. The level of the IC data can be lowered due to the sample containing $CO_2$. $CO_2$ can lower the pH of the liquid sample in the gas washing bottle, which can

further decrease the collection efficiency of $SO_4^{2-}$. The 5 % level of the IC data and the direction of the effect of $CO_2$ would denote maximum diffusional losses onto the sampling lines between the evaporator and the $[H_2SO_4]_{raw}$ measurement point of 95 %, which lies in the range of calculated diffusional losses of 0 ... 98 %. The theoretical level with full diffusional losses is also shown in Fig. 4. The reason why the CI-APi-TOF data lies at the lossless level is presumably due to the direction of adjusting $T_{sa}$ which was from high to low temperatures during the CI-APi-TOF measurement and the time waited for the CI-

APi-TOF signal to stabilize was short with respect to the equilibration time of the sampling line. Performing higher saturator temperatures first can saturate the walls of the sampling lines with $H_2SO_4$ which could later act as preventing diffusional losses using lower saturator temperatures and thus results into the lossless level.

Nevertheless, both the data sets agree well with the shape of the theoretical curve, which implies that $[H_2SO_4]_{raw}$ can be estimated using $T_{sa}$. However, the absolute value for $[H_2SO_4]_{raw}$ cannot be satisfactorily estimated using $T_{sa}$ due to the

discrepancy of the measured concentrations. Therefore, the simulations of this study do not use these measured concentrations as the boundary conditions; instead, the $[H_2SO_4]_{raw}$ values are obtained through inverse modeling.

### 3.2 Particle concentrations and size distributions

Figure 5 represents examples of particle size distributions measured with different $H_2SO_4$ evaporator temperatures, $T_{sa}$. The PSM+CPC data are calculated using the number concentrations measured with different saturator flow rates of the PSM and

with the CPC 3775, i.e. with different $D_{50}$-cut-sizes. To properly compare the data measured with different dilution ratios and sampling line lengths, the comparison requires backwards-corrected data, i.e., all data in the figure are corrected with the DR of the BD and with the diffusional losses caused by the sampling lines between the ED and the measurement devices. However, correcting the distributions backwards from the measured data to the distributions after the ED is not simple because that requires the shapes of the distributions within the whole diameter range to be known. The data of the PSM and the CPC




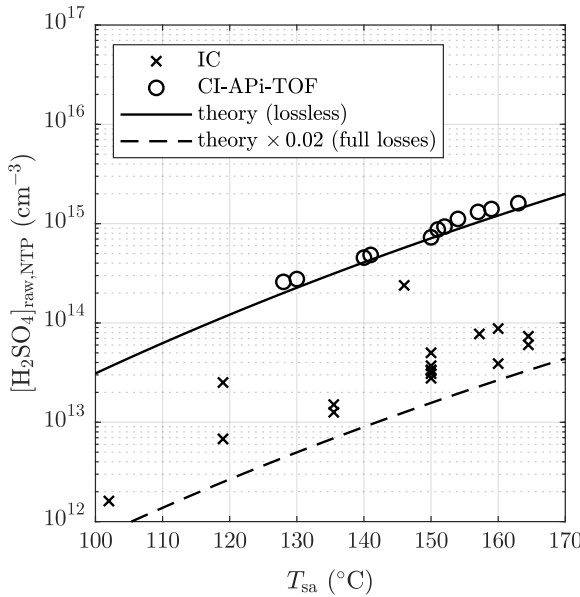

**Figure 4.** Measured and theoretical sulfuric acid concentrations in the raw sample with different sulfuric acid evaporator temperatures. The lossless theoretical level represents the concentrations if the diffusional losses onto the sampling lines between the $H_2SO_4$ evaporator and the $[H_2SO_4]_{raw}$ measurement point are neglected. The theoretical level with full losses represent the concentrations if the sampling lines are fully condensing.

3775 cannot always provide real size distributions because the cumulative nature of the method using particle counters as the size distribution measurement can suffer from noise in measured concentration. For example, the PSM+CPC data with $T_{sa} = 157.2\,°C$ shown in Fig. 5 implies that the concentration could increase with decreasing particle size, but the placing of the data points can be caused by the noise in the measured concentrations. On the other hand, the data implies that there are no particles smaller than $\sim 2.5$ nm in diameter, but the data of the smaller particles can be invisible due to the noise in the measured concentrations. Hence, the unknown concentration of the particles smaller than $\sim 2.5$ nm in diameter can have a significant effect on the total number concentration after the ED calculated from the measured data because these particles play the major role in the effect of the diffusional losses in the sampling lines and in the BD. Due to these uncertainties, the backwards-corrected data (denoting the distributions right after the ED) are not used when comparing the measured results with the simulated results later in this article. Nevertheless, the backwards-corrected data are used when presenting the distributions from all the aerosol devices together because the distributions cannot be presented without correcting them backwards due to different particle losses in the sampling lines of the different devices.

It can be observed that though the Nano-SMPS data are in a nearly log-normal form, there are also size distributions in the PSM+CPC diameter range. Particles generated with lower $T_{sa}$ are lower in concentration and smaller than ones with higher



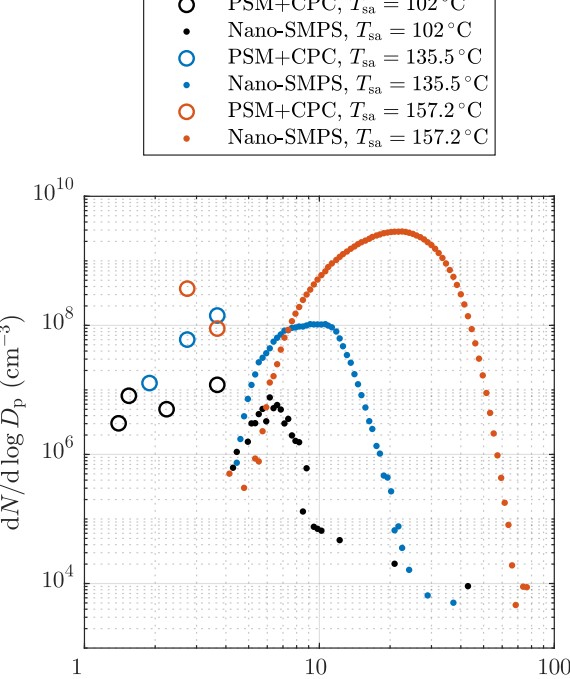

**Figure 5.** Examples of particle size distributions after the ED measured with different $H_2SO_4$ evaporator temperatures in the measurement set 1. The data are corrected with the DR of the BD and with the diffusional losses in the sampling lines after the ED. The concentrations are multiplied with the total DR of the sampling system.

$T_{sa}$; and higher amount of particles are in the PSM+CPC diameter range. The smaller diameter edges of the log-normal size distributions measured by the Nano-SMPS do not connect with the distributions measured by the PSM and the CPC 3775 due to high diffusional losses of very small particles inside the Nano-SMPS device. Thus, the smaller diameter edges of the measured log-normal size distributions are not accurate. By examining the combination of the size distributions measured by the PSM and the CPC 3775 and the size distributions measured by the Nano-SMPS, the real size distributions are not in a log-normal form.

The particle number concentrations measured with the highest saturator flow rate of the PSM ($N_{PSM}$), i.e. the particles with diameters larger than $\sim 1.3$ nm, and the diameters with the average mass ($D_{\bar{m}}$) of the measurement set 1 are presented in Fig. 6. $D_{\bar{m}}$ are calculated using the size distributions measured with the combination of the PSM, the CPC 3775, and the Nano-SMPS which are corrected with the diffusional losses in the sampling lines. The figure consists of data measured at two different days. Due to the uncertainties involved in the $[H_2SO_4]$ measurement, $N_{PSM}$ and $D_{\bar{m}}$ are presented as a function of the simulated $[H_2SO_4]$ in the raw sample. The simulations are presented later in Sec. 4. It can be observed that $N_{PSM}$ increases steeply with increasing $[H_2SO_4]_{raw}$ with lower $[H_2SO_4]_{raw}$ values, but the steepness decreases with increasing $[H_2SO_4]_{raw}$ due




to increasing self-coagulation rate. With lower $[\mathrm{H_2SO_4}]_\mathrm{raw}$ values the slope of $N_\mathrm{PSM}$ versus $[\mathrm{H_2SO_4}]_\mathrm{raw}$ in a log-log scale,

$$n_{N_\mathrm{PSM}\,\mathrm{vs.}\,[\mathrm{H_2SO_4}]_\mathrm{raw}} = \frac{\partial \ln N_\mathrm{PSM}}{\partial \ln [\mathrm{H_2SO_4}]_\mathrm{raw}}, \tag{4}$$

is approximately 10, but decreases to approximately 0.4 with decreasing $[\mathrm{H_2SO_4}]_\mathrm{raw}$. The slope of $J$ versus $[\mathrm{H_2SO_4}]$ is, by the definition of $J$ (Eq. (2)),

$$5 \quad n_{J\,\mathrm{vs.}\,[\mathrm{H_2SO_4}]} = \frac{\partial \ln J}{\partial \ln [\mathrm{H_2SO_4}]} = n_\mathrm{sa}, \tag{5}$$

which is also the nucleation exponent for $[\mathrm{H_2SO_4}]$. The slope $n_{N_\mathrm{PSM}\,\mathrm{vs.}\,[\mathrm{H_2SO_4}]_\mathrm{raw}}$ can provide a rough estimate of the slope $n_\mathrm{sa}$ but due to the other aerosol processes, especially coagulation, having effects on the particle concentrations, the estimated slope can differ a lot from the real $n_\mathrm{sa}$ in the nucleation rate function. The slope at higher $[\mathrm{H_2SO_4}]_\mathrm{raw}$ values is usually decreased due to coagulation and the slope at lower $[\mathrm{H_2SO_4}]_\mathrm{raw}$ values can be increased due to decreased particle detection efficiency of smaller particles. Therefore, $n_\mathrm{sa}$ is expected to be within the range of 0.4 ... 10. Additionally, the estimated slope can also differ from $n_\mathrm{sa}$ because $n_{N_\mathrm{PSM}\,\mathrm{vs.}\,[\mathrm{H_2SO_4}]_\mathrm{raw}}$ is based on $[\mathrm{H_2SO_4}]$ in the raw sample rather than the value of $[\mathrm{H_2SO_4}]$ in a specific location: $[\mathrm{H_2SO_4}]$ decreases from the concentration in the raw sample several orders of magnitude during the dilution process.

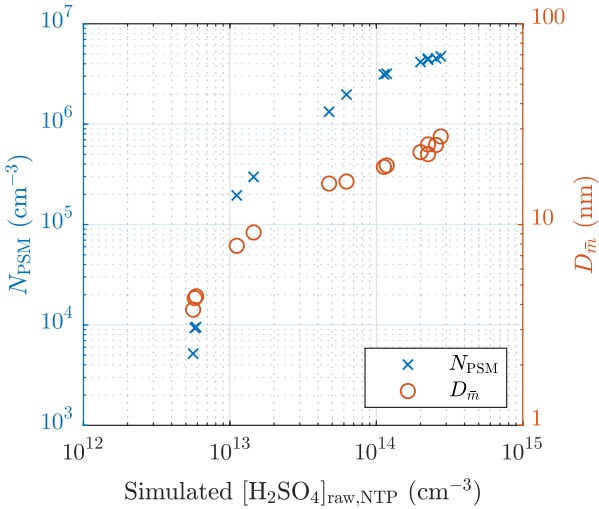

**Figure 6.** The measured number concentrations of the particles larger than $\sim 1.3$ nm and the diameters with the average mass of the measurement set 1 as a function of the simulated $\mathrm{H_2SO_4}$ concentration in the raw sample. The concentrations are multiplied with the total DR of the sampling system.

The effect of humidity on the particle concentration (Set 2) is shown in Fig. 7. The slope of $N_\mathrm{PSM}$ versus $\mathrm{RH_{PTD}}$ in a log-log scale,

$$15 \quad n_{N_\mathrm{PSM}\,\mathrm{vs.}\,\mathrm{RH_{PTD}}} = \frac{\partial \ln N_\mathrm{PSM}}{\partial \ln \mathrm{RH_{PTD}}}, \tag{6}$$





is roughly between 0.1 and 0.2. The slope $n_{N_{\mathrm{PSM}}\,\mathrm{vs.}\,\mathrm{RH}_{\mathrm{PTD}}}$ nearly equals the slope of $N_{\mathrm{PSM}}$ versus $[\mathrm{H_2O}]_{\mathrm{PTD}}$ ($n_{N_{\mathrm{PSM}}\,\mathrm{vs.}\,[\mathrm{H_2O}]_{\mathrm{PTD}}}$) because $T_{\mathrm{PTD}}$ is nearly a constant. The slope $n_{N_{\mathrm{PSM}}\,\mathrm{vs.}\,[\mathrm{H_2O}]_{\mathrm{PTD}}}$ corresponds to the slope $n_{\mathrm{w}}$ with the same uncertainties as involved with the slopes $n_{N_{\mathrm{PSM}}\,\mathrm{vs.}\,[\mathrm{H_2SO_4}]_{\mathrm{raw}}}$ and $n_{\mathrm{sa}}$. Nevertheless, the effect of decreased particle detection is not involved because, in this case, particle size has only a weak dependency of $\mathrm{RH}_{\mathrm{PTD}}$. Additional uncertainty in estimating $n_{\mathrm{w}}$ arises from

the origin of $\mathrm{H_2O}$ vapor in the system, which is both the dilution air and the raw sample. Because $[\mathrm{H_2O}]$ in the raw sample was kept constant, it has a higher effect on the total $[\mathrm{H_2O}]$ with lower values of $\mathrm{RH}_{\mathrm{PTD}}$; thus, the estimated $n_{\mathrm{w}}$ is lower than the real $n_{\mathrm{w}}$ in the nucleation rate function.

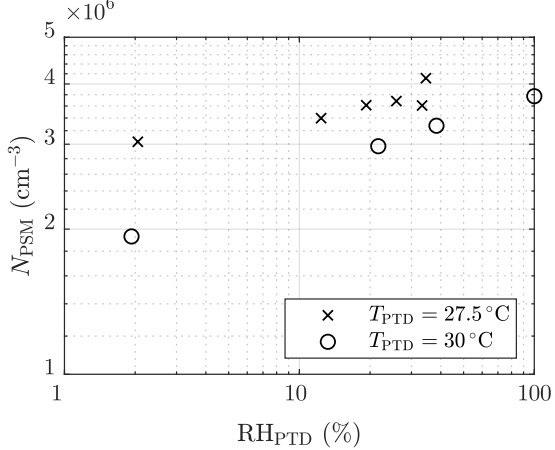

**Figure 7.** The measured number concentrations of the particles larger than $\sim 1.3$ nm of the measurement set 2 as a function of the RH of the PTD dilution air. The concentrations are multiplied with the total DR of the sampling system.

The effect of $T_{\mathrm{PTD}}$ can be observed in Figs. 7 and 8. Lower temperatures result in higher concentrations of $N_{\mathrm{PSM}}$. However, the examination is problematic because keeping $\mathrm{RH}_{\mathrm{PTD}}$ as a constant while increasing $T_{\mathrm{PTD}}$ (Set 3a) increases $[\mathrm{H_2O}]$, which

results in lower $N_{\mathrm{PSM}}$ with lower temperatures. Therefore, keeping $x_{\mathrm{w,PTD}}$ as a constant (Set 3b) is better in examining $m_{\mathrm{sa}}$. One of the measurements with $T_{\mathrm{PTD}} = 50\,^{\circ}\mathrm{C}$ is, however, a significant outlier in Set 3b. Estimating the exponent $m_{\mathrm{sa}}$ from the slope in Fig. 8 is not straightforward because temperature is included also in the concentrations having yet unknown exponents.

The effect of the residence time on the particle concentrations is presented in Tab. 2. With $T_{\mathrm{sa}} = 135.5\,^{\circ}\mathrm{C}$ the ratio of $N$ with the residence times of 1.4 s and with the residence time of 2.8 s is below unity, but above unity with higher temperatures. The

ratio below unity denotes that the nucleation process is not diminished yet at the time of 1.4 s, e.g., the ratio of 0.74 denoting 74% of particles are formed within the time range of 0 ... 1.4 s and the remaining 26% within the time range of 1.4 ... 2.8 s. With higher temperatures the ratio is above unity because self-coagulation begins to decrease the number concentration, especially at the later times where the number concentration is the highest. The nucleation process may continue after 1.4 s but it cannot be easily seen with higher temperatures. Because coagulation has no effect on the mass concentrations ($M$), the

ratios of $M$ measured with the combination of the PSM, the CPC 3775, and the Nano-SMPS with the residence time of 1.4 s and with the residence time of 2.8 s are near unity with higher temperatures. The effect of particle growth and wall losses,




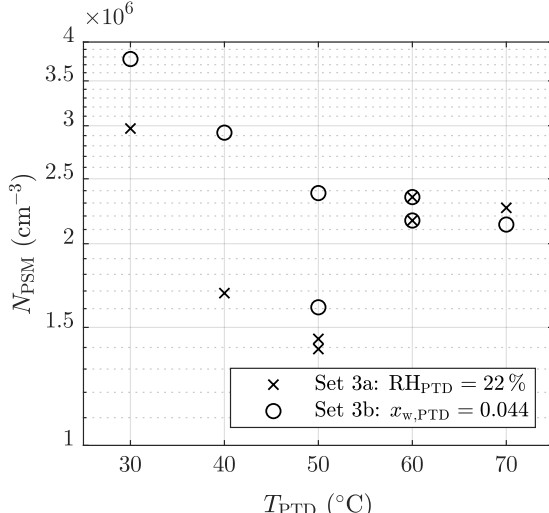

**Figure 8.** The measured number concentrations of the particles larger than $\sim 1.3$ nm of the measurement set 3 as a function of the $T$ of the PTD dilution air. The concentrations are multiplied with the total DR of the sampling system.

however, have effects on the ratios too. The temperature with which the coagulation process would eliminate the effect of the nucleation process, resulting in the number concentration ratio of unity, is near $142\,^{\circ}\mathrm{C}$.

**Table 2.** The ratios of the measured number concentrations and mass concentrations with the residence times of 1.4 s and 2.8 s, in the measurement set 4. The values are corrected with the DR of the BD and with the diffusional losses in the sampling lines after the ED; thus, the values correspond with the distributions existing after the ED.

| $T_{\mathrm{sa}}$ ($^{\circ}$C) | $\frac{N(1.4\,\mathrm{s})}{N(2.8\,\mathrm{s})}$ | $\frac{M(1.4\,\mathrm{s})}{M(2.8\,\mathrm{s})}$ |
|---:|---|---|
| 135.5 | 0.74 | 0.28 |
| 150 | 1.29 | 0.92 |
| 160 | 1.72 | 0.96 |
| 164.5 | 1.74 | 1.10 |

## 4 Simulations

Every measurement point presented in Tab. 1 was simulated with the model consisting of four phases: (1) the CFD simulations
5  to solve the flow and the temperature field of the sampling system, (2) the CFD-TUTMAM simulations to solve the aerosol processes in the sampling system, (3) correcting the particle sizes decreasing rapidly in the dry ED, and (4) calculating the pen-



etration of the particles due to diffusional losses in the sampling lines after the sampling system and the detection efficiencies of the particle counting devices.

## 4.1 CFD model

The CFD simulations to solve the flow and the temperature fields for every simulation case were performed with a commercially

available software ANSYS FLUENT 17.2. It is based on a finite volume method in which the computational domain is divided into a finite amount of cells. Governing equations of the flow are solved in every computational cell iteratively until sufficient convergence is reached. In this study, the governing equations in the first phase are continuity, momentum, energy, radiation, and turbulence transport equations.

    The computational domain in the CFD simulations is an axial symmetric geometry consisting of the PTD, the aging chamber,

and the ED (Fig. 1). An axial symmetric geometry was selected over a three-dimensional geometry due to high computational demand of the model and a nearly axial symmetric profile of the real measurement setup. The domain was divided into $\sim 8 \times 10^5$ computational cells, of which the major part was located inside the PTD where the smallest cells are needed due to the highest gradients. The smallest cells were 20 µm in side lengths and were located in the beginning of the porous section, where the hot exhaust and the cold dilution air meet.

In contrast to our previous study (Olin et al., 2015), the ED was also included in the computational domain though the ED has only a minor effect on nucleation (Lyyränen et al., 2004; Giechaskiel et al., 2009). Because the ED has a high speed nozzle that cools the flow locally to near -30 °C, including it in the domain provides partial validation for $m_{sa}$ in the following way: if too high value for $m_{sa}$ were used, nucleation would be observed in the ED, being in contradiction with the former studies. The internal fluid inside the sampling lines is modeled as a mixture of air, $H_2O$ vapor, and $H_2SO_4$ vapor. The sampling lines are

modeled as solid zones of steel or Tygon. 10 cm of the external fluid, modeled as air, is also included in the domain to simulate natural cooling of the sampling lines.

    Flow rate and temperature boundary conditions for the simulated sampling system were set for the each simulation case to the measured values. Due to steady-state conditions and high computational demand, all governing equations were time-averaged; thus, the simulations were performed with a steady-state type. Turbulence was modeled using the SST-$k$-$\omega$ model,

which is one of the turbulence models used with a steady-state simulation. It produced the most reliable results of the available steady-state turbulence models based on the pressure drop in the porous section. Turbulence, however, can have a significant role in the wall losses of the vapors and the particles in the regions where the turbulence level is high. In this sampling system, the turbulence level is high in the upstream part of the aging chamber where the diameter of the sampling line increases steeply. Validating the suitability of the turbulence model for this geometry would require a measurement of, e.g., solid seed particle

concentrations after and before the sampling system without any aerosol processes, such as nucleation, condensation, and coagulation. However, that kind of measurement has not been performed yet.



## 4.2 CFD-TUTMAM

The main functionality of the CFD-TUTMAM based on the previous aerosol model, CFD-TUTEAM, is described by Olin et al. (2015). However, because the measured distributions are not in a log-normal form, the inclusion of the PL+LN model (Olin et al., 2016) was beneficial. The PL+LN model simulates the initial growth of newly-formed very small particles by modeling

the particle size distribution with the combination of a power law (PL) and a log-normal (LN) distribution. Newly-formed particles are first put to the PL distribution, after which they are transferred to the LN distribution by particle growth.

The CFD-TUTMAM adds three governing equations per a distribution (denoted by $j$) to the CFD model using a modal representation of the particle size distribution, i.e. the distributions are modeled by three variables: number ($M_{j,0} = N_j$), surface area-related ($M_{j,2/3}$), and mass ($M_{j,1}$) moment concentrations. $M_{j,1}$ are further divided into different components in

a multi-component system. Due to small particle size and low particle loading, the aerosol phase has only a minor effect on the gas phase properties. Therefore, continuity, momentum, energy, radiation, and turbulence transport equations can be excluded from the computation after the flow and temperature fields are solved, and only gas species equations and the aerosol model equations are solved. The governing equation of the aerosol model for the concentration of a $k$th moment of a distribution $j$ is

$$\frac{\partial M_{j,k}}{\partial t} = - \nabla \cdot (M_{j,k} \boldsymbol{u}) + \nabla \cdot \left( \rho_{\mathrm{f}} \overline{D}_{j,k,\mathrm{eff}} \nabla \frac{M_{j,k}}{\rho_{\mathrm{f}}} \right)$$

$$+ \mathrm{nucl}_{j,k} + \mathrm{cond}_{j,k} + \mathrm{coag}_{j,k} + \mathrm{transfer}_{j,k}, \qquad (7)$$

where $\boldsymbol{u}$, $\rho_{\mathrm{f}}$, and $\overline{D}_{j,k,\mathrm{eff}}$ are the fluid velocity vector, the fluid density, and the $k$th moment-weighted average of the particle effective diffusion coefficient, respectively. The last terms in Eq. (7) represent source terms for nucleation, condensation, coagulation, and intermodal particle transfer. In this study, aerosol is modeled with two distributions: a PL distribution ($j = \mathrm{PL}$) and a LN distribution ($j = \mathrm{LN}$). In this study, two gas species equations, which model the internal fluid mixture as the mass

fractions of $H_2O$ and $H_2SO_4$, are built in the CFD model, but the opposite numbers of the source terms of nucleation and condensation are added to them to maintain the mass closure of the species.

After each iteration step of the CFD-TUTMAM simulation, the parameters of the distributions are calculated for every computational cell by using the three moment concentrations. The parameters for the PL distribution are the number concentration ($N_{\mathrm{PL}}$), the slope parameter ($\alpha$), and the largest diameter ($D_2$). The smallest diameter ($D_1$) has a fixed value of 1.15 nm which

is the smallest detectable particle diameter with the devices used. The density function for the PL distribution is

$$\left. \frac{\mathrm{d}N}{\mathrm{d}\ln D_{\mathrm{p}}} \right|_{\mathrm{PL}} = \begin{cases} N_{\mathrm{PL}} \left( \frac{D_{\mathrm{p}}}{D_2} \right)^{\alpha} \beta_0, & D_1 \leq D_{\mathrm{p}} \leq D_2 \\ 0, & \text{otherwise} \end{cases}, \qquad (8)$$

where $\beta_0$ is a function

$$\beta_l \left( \alpha, \frac{D_1}{D_2} \right) = \begin{cases} \frac{\alpha + l}{1 - \left( \frac{D_1}{D_2} \right)^{\alpha + l}}, & \alpha \neq -l \\ \frac{1}{-\ln\left( \frac{D_1}{D_2} \right)}, & \alpha = -l \end{cases}. \qquad (9)$$



The parameters for the LN distribution are the number concentration ($N_{\mathrm{LN}}$), the geometric standard deviation ($\sigma$), and the geometric mean diameter ($D_{\mathrm{g}}$). An analytical solution exists for the reconstruction of the parameters from the moment concentrations for the LN distribution but not for the PL distribution; thus, it is solved numerically. A numerical solution is obtained by using the Levenberg-Marquardt iteration algorithm, in contrast to a slower method using a pre-calculated interpolation table

described by Olin et al. (2016).

The nucleation source terms in Eq. (7) for different moments are

$$\mathrm{nucl}_{\mathrm{PL},0} = J$$
$$\mathrm{nucl}_{\mathrm{PL},2/3} = J\left(m_{\mathrm{sa}}^{*} + m_{\mathrm{w}}^{*}\right)^{2/3}$$
$$\mathrm{nucl}_{\mathrm{PL},1,\mathrm{sa}} = J m_{\mathrm{sa}}^{*} \tag{10}$$

$$\mathrm{nucl}_{\mathrm{PL},1,\mathrm{w}} = J m_{\mathrm{w}}^{*}$$
$$\mathrm{nucl}_{\mathrm{LN},k} = 0,$$

where $J$ is the nucleation rate as in Eq. (2) and $m_{\mathrm{sa}}^{*}$ and $m_{\mathrm{w}}^{*}$ are the masses of $H_2SO_4$ and $H_2O$ in a newly-formed particle. The value of $D_1 = 1.15\,\mathrm{nm}$ was chosen for the diameter of the newly-formed particles. A particle of this diameter is in equilibrium with water uptake in the temperature of 300 K and in the relative humidity of 22 % if the mass fraction of $H_2SO_4$ in the

particle is 0.71. This constant value is used with nucleation though the mass fraction would vary between 0.5 and 1 if the whole temperature and humidity range were considered, but the major part of nucleation occurs in the conditions having the equilibrium mass fraction of near 0.71. This mass fraction and particle diameter corresponds to a cluster containing 5.7 $H_2SO_4$ molecules and 12.4 $H_2O$ molecules.

Diffusion, condensation, and coagulation are modeled as described in the reference Olin et al. (2015) and intermodal particle

transfer as described in the reference Olin et al. (2016). Condensation is modeled with the growth by $H_2SO_4$ from which immediately follows the water uptake until the water equilibrium is achieved. The water equilibrium procedure is also described in the reference Olin et al. (2015). The coagulation modeling includes intramodal coagulation within the both distribution and intermodal coagulation between the distributions.

Intermodal particle transfer includes condensational transfer and coagulational transfer from the PL distribution to the LN

distribution. In contrast to a constant condensational transfer factor $\gamma$ of the PL+LN model described in the reference Olin et al. (2016), a function of $\alpha$, $D_1/D_2$, and $k$ is used in the CFD-TUTMAM due to a more complex particle growth modeling. The




function used here is

$$
\gamma\left(\alpha, \frac{D_1}{D_2}, k\right) =
\begin{cases}
0.1\alpha + 0.5, & \alpha \geq 0 \\
0, & \alpha < 0
\end{cases}
\times
\begin{cases}
\frac{3}{\beta_0}, & k = 0 \\
\frac{2}{\beta_1} + \frac{1}{\beta_2}, & k = \frac{2}{3} \\
\frac{3}{\beta_2}, & k = 1
\end{cases}
. \tag{11}
$$

The functional form of $\gamma$ is derived so that the condensational transfer eliminates the effect of increasing $\alpha$ by the condensation
process and also tries to keep $\alpha$ positive because a PL distribution with a negative $\alpha$ in combination with a LN distribution represents a distribution having a nonphysical local minimum between the distributions. The form of $\gamma$ also restricts $\alpha$ increasing
too high, which would cause numerical difficulties. Particles are not lost or altered during the intermodal particle transfer, it is
only controlling the ratio of particles represented in the PL distribution and in the LN distribution. Higher values of $\gamma$ result in
lower $N_{\mathrm{PL}}/N$ ratio.

Deposition of particles and condensation of vapors onto the inner walls of the sampling lines have direct effect on the aerosol
concentrations at the measurement devices. The particle deposition was modeled by setting the boundary conditions for the
aerosol concentrations at the walls to zero, which represents deposition driven by diffusion and turbulence. Condensation
of $H_2O$ and $H_2SO_4$ vapors onto the walls was modeled by setting the boundary conditions for the mass fractions of $H_2O$
and $H_2SO_4$ at the walls to saturation mass fractions in an aqueous solution of $H_2SO_4$, in contrast to the simpler method in
the previous study (Olin et al., 2015). The simpler method caused $H_2SO_4$ to be completely non-condensing onto the walls
because the saturation factor of the pure vapor never exceeded unity. Instead, the method using the saturation mass fractions
in the solution induces some condensation because the vapor pressure of a hygroscopic liquid over an aqueous solution is
lower than over a pure liquid. This method provides also smoother behavior of the boundary conditions on the walls. The
method is, however, strongly dependent of the chosen activity coefficient functions of the vapors, which have large differences
between each other due to their exponential nature. Activity coefficients used here are based on the values reported by Zeleznik
(1991). However, due to exponential and non-monotonic nature of activity coefficients, they cause numerical difficulties in
CFD modeling; thus, a monotonic van Laar type equation fitted by Taleb et al. (1996) from the data of Zeleznik (1991) was
used.

### 4.3 Dry particle model

The main trend of the RH inside the sampling system is increasing due to decreasing temperature. This results in increasing
water uptake rate during the particle growth process, which can be modeled by the condensation rate of $H_2O$ that is simply
the condensation rate of $H_2SO_4$ multiplied with a suitable factor (the water equilibrium procedure described by Olin et al.
(2015)). However, when the sample enters to the ED, the RH decreases rapidly due to a dry dilution air, but the growth process
by the condensation of $H_2SO_4$ still continues. This results in increasing $H_2SO_4$ amount in the particles but rapidly decreasing




H$_2$O amount, which cannot be modeled with the water uptake model. Hence, the particles after the ED simulated by the CFD-TUTMAM contain incorrectly too much water.

All the simulated particle size distributions output by the CFD-TUTMAM were corrected to correspond the water amount that would be in the conditions after the ED ($T \approx 23\,°\text{C}$ and RH $\approx 3.6\,\%$). These conditions are mainly caused by the conditions of compressed air directed to the ED. Additionally, the particle size measurement device (Nano-SMPS) used room air, having nearly equal conditions as compressed air, as the sheath flow air. Dry sheath flow air also dries particles rapidly inside the device. The theory behind the dry particle model equals the theory behind the water uptake model in the CFD-TUTMAM, but the drying process is significantly faster and in opposite direction, in contrast to the water uptake connected to the condensation rate of H$_2$SO$_4$ in the CFD-TUTMAM. Figure 9 represents examples of particle diameters in different humidities, e.g., a particle with the diameter of 40 nm in the RH of 60 % shrinks to the diameter of 30 nm when sampled with the ED.

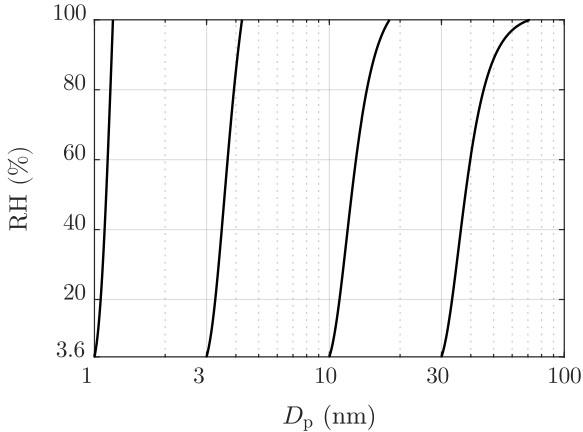

**Figure 9.** Examples of particle diameters in different humidities in the temperature of 23 °C. The lowest RH value represents the RH of the dilution air of the ED.

### 4.4 Penetration and detection efficiency model

Particle size distributions output by the CFD-TUTMAM and corrected with the dry particle model were also corrected according to the penetration and detection efficiency model. Particle penetration in the sampling lines between the ED and the measurement devices were calculated with the equations of Gormley and Kennedy (1948). All the internal diameters of the used sampling lines were sufficiently large to keep the flows laminar to minimize the diffusional losses. The penetration-corrected size distributions were multiplied with the detection efficiency curves presented in Fig. 2 to simulate the measured number concentrations by the PSM and the CPC 3775 and the measured size distribution by the Nano-SMPS.



## 4.5 Inverse modeling

The simulated number concentrations measurable by the PSM with different saturator flow rates and by the CPC 3775 and the simulated size distributions measurable by the Nano-SMPS were compared with the measured ones during inverse modeling. The exponents $n_{\mathrm{sa}}$, $n_{\mathrm{w}}$, and $m_{\mathrm{sa}}$ were altered until the simulated and the measured variables corresponded satisfactorily in all simulated cases. The proportionality coefficient $k$ in Eq. (2) is unknown and depends on the exponents. Because the value of $k$ affects directly on the nucleation rate magnitude, it was obtained by fitting until the simulated and the measured number concentrations corresponded.

Due to the uncertainties involved in the measurement of $[\mathrm{H_2SO_4}]_{\mathrm{raw}}$, the boundary conditions for $[\mathrm{H_2SO_4}]$ in the CFD-TUTMAM simulations could not be set reliably. Hence, it was considered a fitting parameter also. It was estimated by comparing the aerosol mass concentrations because $[\mathrm{H_2SO_4}]_{\mathrm{raw}}$ has a direct effect on the particle sizes, but it also affects on $J$. In conclusion, the inverse modeling requires fitting all the five parameters ($n_{\mathrm{sa}}$, $n_{\mathrm{w}}$, $m_{\mathrm{sa}}$, $k$, and $[\mathrm{H_2SO_4}]_{\mathrm{raw}}$) to obtain the function for $J$. The first four parameters were fitted in a way they have the same value for every simulation case, but the last parameter, $[\mathrm{H_2SO_4}]_{\mathrm{raw}}$, was fitted in every simulation case separately. In the simulations related to the measurement sets 2 ... 4, $T_{\mathrm{sa}}$ was not altered between the measurement points; therefore, the value of $[\mathrm{H_2SO_4}]_{\mathrm{raw}}$ in the simulations was constant. Because only one parameter was fitted separately, only one of the outputs, the aerosol number or mass concentration, could correspond with the measured value exactly. In this study, the number concentration was chosen as the main output of which correspondence is preferred over the correspondence of the mass concentration because nucleation process is connected more straightly to the number concentration.

The uncertainties involved in modeling turbulence and the condensation of the vapors onto the walls affect the number and mass concentrations in the measurement devices. Nevertheless, these uncertainties become partially insignificant because $k$ and $[\mathrm{H_2SO_4}]_{\mathrm{raw}}$ are considered fitting parameters, which partially neglect incorrectly modeled losses of particles and vapors.

## 5 Simulation results

In this section, the outputs of the simulations performed using the nucleation rate function with the best correspondence between the measured and the simulated data are described firstly. Finally, the used nucleation rate function is presented.

## 5.1 Sulfuric acid concentrations

Figure 10 represents the comparison of inversely modeled $[\mathrm{H_2SO_4}]_{\mathrm{raw}}$ with the measured and theoretical ones. The simulated concentrations vary between 0.05 and 0.57 times the theoretical concentrations where the lowest values are observed with lower $T_{\mathrm{sa}}$ values probably due to the effect of increasingly saturating $\mathrm{H_2SO_4}$ liquid onto the sampling lines with higher temperatures that can decrease the diffusional losses onto the sampling lines. All values lie between the theoretical level assuming full diffusional losses and the lossless theoretical level. A weak agreement of the simulated concentrations with 0.15 times




the theoretical curve can be seen, which implies the diffusional losses of 85 % onto the sampling lines between the $H_2SO_4$ evaporator and the PTD.

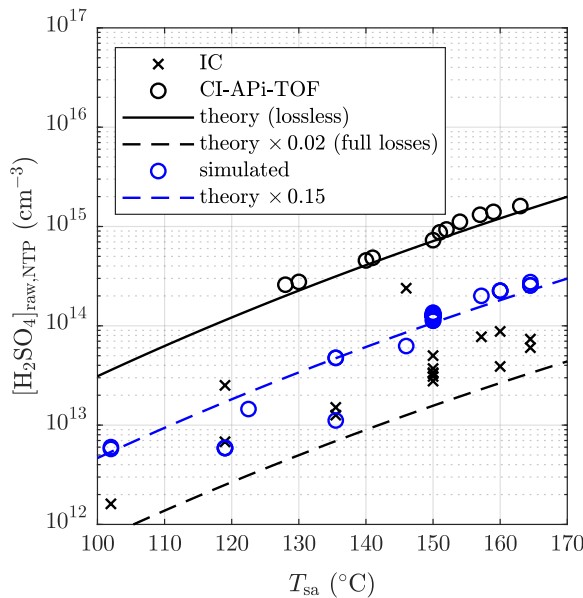

**Figure 10.** Simulated sulfuric acid concentrations in the raw sample compared with the measured and the theoretical concentrations with different sulfuric acid evaporator temperatures.

## 5.2 Particle size distributions

Examples of measured and simulated particle concentrations and size distributions of the measurement set 1 are presented in
5  Fig. 11. The panes (a) and (c) in the figure represent the concentrations measured/measurable with the PSM and the CPC 3775. Because the concentrations decrease with increasing cut diameter in the case with $T_{sa} = 102\,°C$ (a), particle size distribution exists within this diameter range, which is also seen in the simulated data. However, the concentration measured with the cut diameter of 3.1 nm is two-fold compared to the simulated one, implying that the real distribution is not a pure PL+LN distribution or the shape of the distribution is modeled incorrectly near the diameter of 3.1 nm. Conversely, in the case with
10  $T_{sa} = 157.2\,°C$ (c), the concentrations are in the same level, which implies no size distribution within that diameter range.

The panes (b) and (d) in Fig. 11 represent examples of measured and simulated Nano-SMPS data. The case with $T_{sa} = 102\,°C$, (b) represent an example of one of the worst agreements of measured and simulated size distributions. While the simulated total number concentration agrees with the measured one in that case, the particle diameter is underestimated with the factor of $\sim 2$. The disagreement is discussed later in this section. Conversely, in the case with $T_{sa} = 157.2\,°C$ (d), the
15  distributions agree well, except that the model predicts higher particle concentration in the diameter range of 2 ... 10 nm.



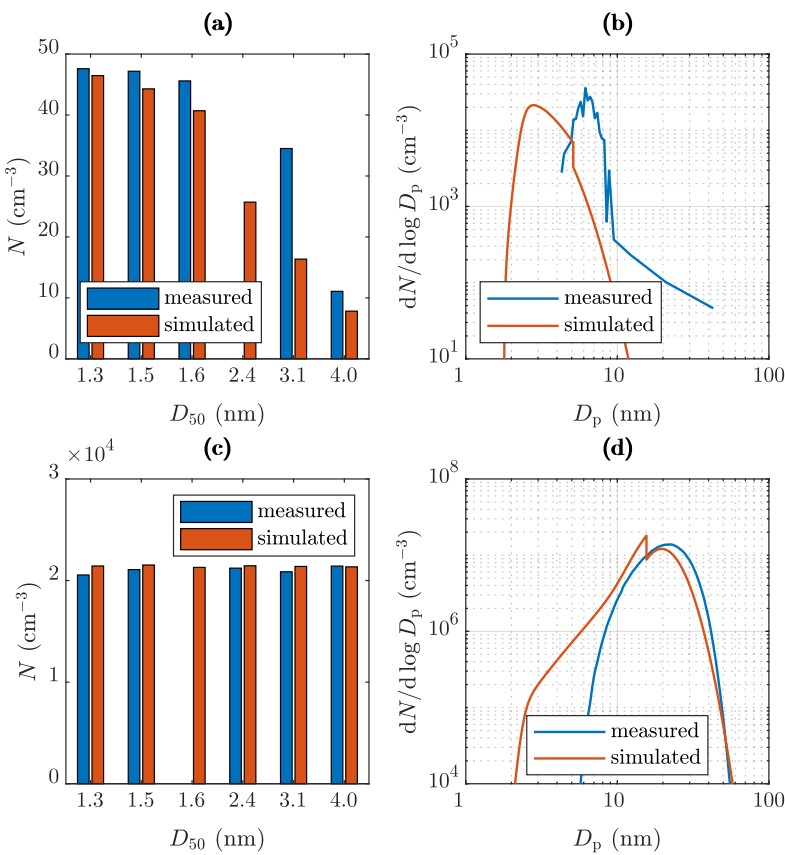

**Figure 11.** Examples of measured and simulated (a) number concentrations from the PSM and the CPC 3775 with $T_{\mathrm{sa}} = 102\,^\circ$C, (b) size distributions from the Nano-SMPS with $T_{\mathrm{sa}} = 102\,^\circ$C, (c) number concentrations from the PSM and the CPC 3775 with $T_{\mathrm{sa}} = 157.2\,^\circ$C, and (d) size distributions from the Nano-SMPS with $T_{\mathrm{sa}} = 157.2\,^\circ$C. The $D_{50}$ values in the range of $1.3\ldots3.1$ nm represent the cut-sizes of the PSM with different saturator flow rates and the $D_{50}$ value of 4.0 nm represents the cut-size of the CPC 3775.





This disagreement can be due to the decreased particle detection efficiency of the Nano-SMPS with very small particles due to very high diffusional losses inside the device. These diffusional losses are not included in the penetration and detection efficiency model and are thus not seen in the simulated distributions. Because the detection efficiency curve of the CPC 3776 is included in the model, the simulated size distributions measurable with the Nano-SMPS decrease steeply with decreasing

5   particle diameter near the particle diameter of $D_{50} = 2.4\,\text{nm}$. The sharp peak at the diameter of $\sim 20$ nm in the simulated distribution in (d) is caused by the nature of the PL+LN model where the PL distribution ends at the diameter of $D_2 \approx 20\,\text{nm}$. While Fig. 11 represents the data at the measurement devices, Fig. 12 represents the example distributions after the ED. From the latter figure the PL distribution is seen as a whole, starting from the diameter of $D_1 = 1.15\,\text{nm}$.

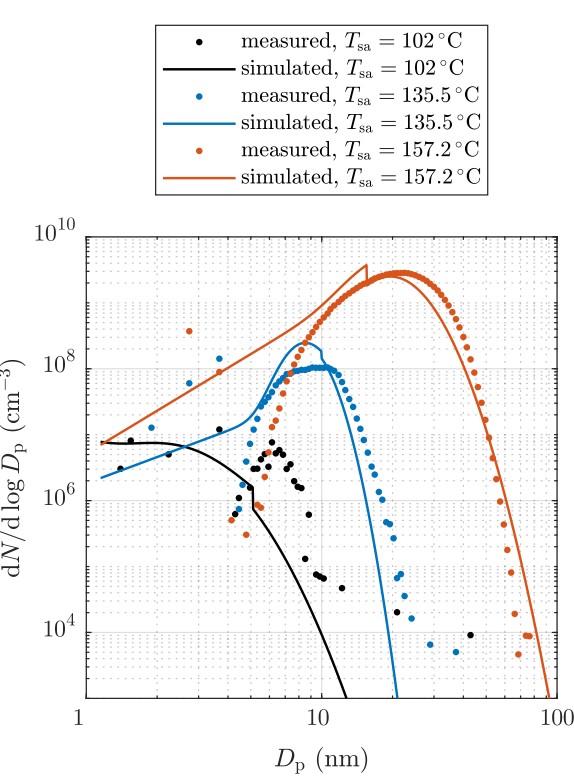

**Figure 12.** Examples of measured and simulated particle size distributions after the ED. The measured data are corrected with the DR of the BD and with the diffusional losses in the sampling lines after the ED. Additionally, all concentrations are multiplied with the total DR of the diluting sampling system.

The requirement of the PL+LN model can be observed from Fig. 13, in which the particle number concentrations and sizes

10   of a single simulation case with different values of $[\text{H}_2\text{SO}_4]_{\text{raw}}$ are presented. With low values of $[\text{H}_2\text{SO}_4]_{\text{raw}}$, both $N$ and $D_{\bar{m}}$ behave discontinuously if only the LN distribution is simulated: particles are first small and in low concentration when $[\text{H}_2\text{SO}_4]_{\text{raw}}$ increases, and then suddenly rise to higher levels. This is, however, not seen with the PL+LN model, which has a





smoother behavior. Therefore, by simulating with the LN distribution only, it is impossible to produce, e.g., a size distribution with $N = 10^4 \, \text{cm}^{-3}$ or $D_{\bar{m}} = 3 \, \text{nm}$ with this simulation setup, whereas with the PL+LN model it is possible.

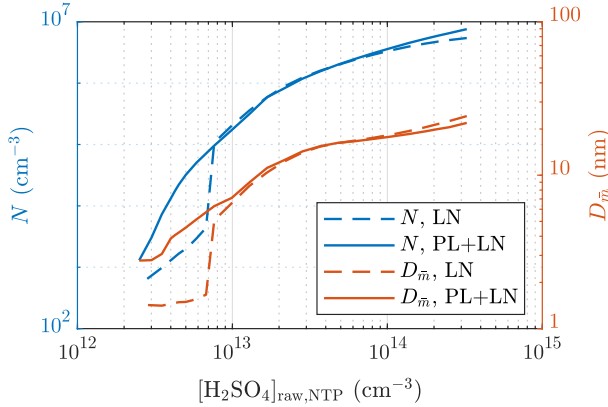

**Figure 13.** Comparison of the particle number concentrations and the diameters with the average mass after the ED simulated using the LN distribution only and using both the PL and the LN distributions.

## 5.3 Particle concentrations and sizes

Figure 14 represents the comparison of the simulated and the measured $N_{\text{PSM}}$ and $D_{\bar{m}}$ values after the ED. The blue crosses in the pane (a) correspond well with the measured concentrations because they represent the cases for which $N_{\text{PSM}}$ was obtained by fitting the value of $[\text{H}_2\text{SO}_4]_{\text{raw}}$. The black crosses have more deviations because they represent all the other cases, the $N_{\text{PSM}}$ values of which originate from the simulations, e.g., simulated with different $\text{RH}_{\text{PTD}}$, $T_{\text{PTD}}$, or residence times. Nevertheless, all the simulated $N_{\text{PSM}}$ values correspond with the measured values relatively well. The optimal scenario would be that all the $N_{\text{PSM}}$ values would correspond exactly with the measured values, but that would imply the exponents $n_{\text{w}}$ and $m_{\text{sa}}$ in the nucleation rate function can be modeled exactly with constant values within the concentration and temperature ranges of this study. However, it is not expected that the constant exponents would represent exactly the nucleation rate function in all concentration and temperature ranges.

The blue crosses in the pane (a) of Fig. 14 correspond moderately with the measured $D_{\bar{m}}$ values. It can be observed that with lower and higher values of $D_{\bar{m}}$ the model underestimates the particle sizes. There are several issues which can cause this discrepancy: (1) the exponent $n_{\text{sa}}$ varies with $[\text{H}_2\text{SO}_4]$, (2) a problem in calculating $D_{\bar{m}}$ from the measurement data, (3) a problem in estimating a proper $N_{\text{PL}}/N$ ratio in the PL+LN model, and (4) an uncertainty in simulating the condensation process. The most possible explanation is (1) because according to the CNT, $n_{\text{sa}}$ decreases with increasing $[\text{H}_2\text{SO}_4]$. This can be seen as underestimated particle sizes because larger particle sizes would require higher $[\text{H}_2\text{SO}_4]_{\text{raw}}$ but that would cause overestimated $N_{\text{PSM}}$. To overcome the overestimated $N_{\text{PSM}}$ in low and high $[\text{H}_2\text{SO}_4]$ values, $k$ should be decreased in low and high $[\text{H}_2\text{SO}_4]$ values, which indicates decreasing $n_{\text{sa}}$ with increasing $[\text{H}_2\text{SO}_4]$. The point (2) can explain at least the discrepancy of the lower values of $D_{\bar{m}}$ because calculating $D_{\bar{m}}$ from the measured PSM, CPC 3775, and Nano-SMPS data





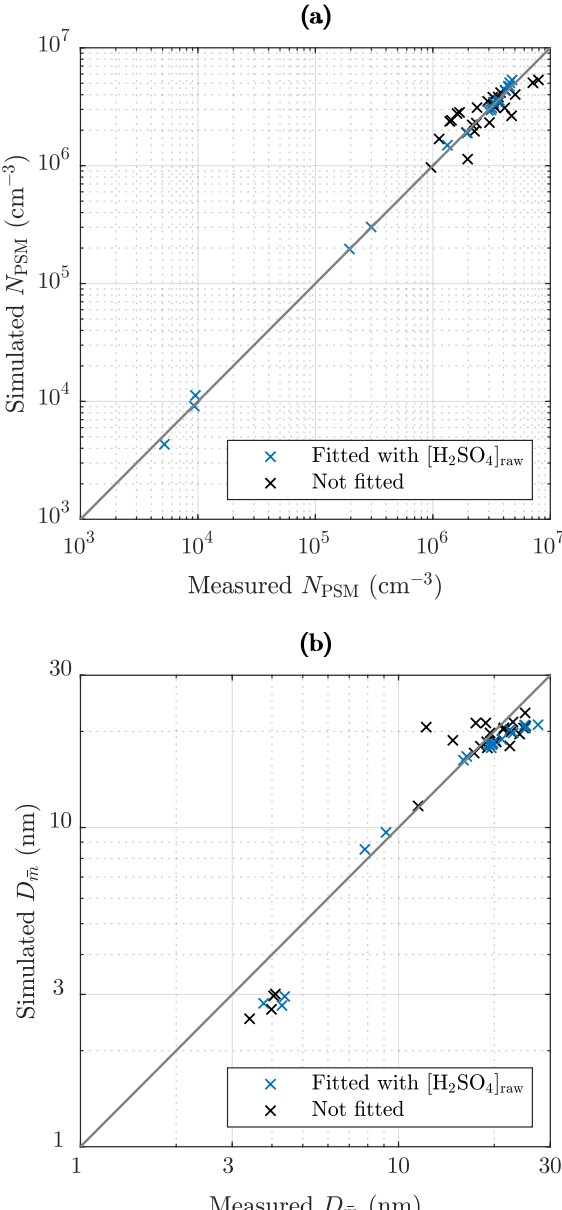

**Figure 14.** Comparison of the simulated and the measured (a) number concentrations of the particles larger than $\sim 1.3$ nm and (b) the diameters with the average mass after the ED. The blue crosses represent the cases for which $N_{\mathrm{PSM}}$ and $D_{\bar{m}}$ were obtained by fitting the value of $[\mathrm{H_2SO_4}]_{\mathrm{raw}}$. The black crosses represent the cases of the measurement sets 2 ... 4 in which the values of $[\mathrm{H_2SO_4}]_{\mathrm{raw}}$ originated from an another case of the measurement set having the same $T_{\mathrm{sa}}$ value.





is not straightforward, especially with the lower values of $D_{\bar{m}}$ in which the distributions measured by the Nano-SMPS are cut from the smaller diameter edge due to diffusional losses. Therefore, $D_{\bar{m}}$ calculated from the measurement data may be overestimated with the lower values of $D_{\bar{m}}$. However, by comparing the measured and the simulated size distributions with $T_{sa} = 102\,^{\circ}C$ in Fig. 12 (measured $D_{\bar{m}} = 4.3\,nm$, simulated $D_{\bar{m}} = 2.8\,nm$), it can be seen that the larger diameter edges of

the distribution do not correspond satisfactorily either, which implies (1) being the most possible explanation. Conversely, the discrepancy of the higher values of $D_{\bar{m}}$ can be partially explained by (3) because simulating those cases with the LN distribution only higher values of $D_{\bar{m}}$ are output. That implies the PL+LN model overestimates the $N_{PL}/N$ ratio. The $N_{PL}/N$ ratio is controlled by the value of $\gamma$, the proper functional form of which is still under development in the PL+LN model. The last point (4) can also explain the discrepancies but the direction of a discrepancy could be in one way or another. The

black crosses follow mainly the same curve as the blue crosses with the exception of four cases in which the values of $D_{\bar{m}}$ are overestimated. These cases belong to the measurement set 3 and have high $T_{PTD}$. This discrepancy raises the point (4) because there are clearly some uncertainties involved in the condensation process modeling when $T_{PTD}$ is high. It can be related, e.g., to the activity coefficient function of $H_2SO_4$ because too low activity coefficient would cause too low vapor pressure of $H_2SO_4$ at the surface of a particle, which would cause too large particles.

Table 3 represents the ratios of the simulated $N$ and $M$ with the residence times of 1.4 s and 2.8 s. The simulated ratios follow the same behavior as the measured ratios: with a low $T_{sa}$ value the ratios are below unity and with higher $T_{sa}$ values the ratio of $N$ increases but the ratio of $M$ stays near unity. The ratios with a low $T_{sa}$ value correspond well with the measured values, but according to the simulations, the ratio of $N$ does not increase with increasing $T_{sa}$ equally with the measured ratios. This implies the coagulation rate is underestimated in the model but the reason for that is unknown. The temperature with

which the coagulation process would eliminate the effect of the nucleation process, resulting in the number concentration ratio of unity, is near 148 $^{\circ}C$ (near 142 $^{\circ}C$ according to the measurements).

**Table 3.** The ratios of the simulated number concentrations and mass concentrations after the ED with the residence times of 1.4 s and 2.8 s, in the measurement set 4. The values in parentheses denote the measured values as presented in Tab. 2.

| $T_{sa}$ (°C) | $\frac{N(1.4\,s)}{N(2.8\,s)}$ | $\frac{M(1.4\,s)}{M(2.8\,s)}$ |
|---|---|---|
| 135.5 | 0.66 (0.74) | 0.25 (0.28) |
| 150 | 1.04 (1.29) | 0.88 (0.92) |
| 160 | 1.07 (1.72) | 0.99 (0.96) |
| 164.5 | 1.06 (1.74) | 0.96 (1.10) |



## 5.4 Nucleation rate function

The nucleation rate function with the best correspondence between the measured and the simulated data having a type of Eq. (2) used in the simulations has the parameters presented in Tab. 4 and is thus

$$J\left([H_2SO_4],[H_2O],T\right)=5.8\times 10^{-26}\frac{[H_2SO_4]^{1.9}[H_2O]^{0.5}}{p_{sa}{}^{\circ}(T)^{0.75}}, \qquad (12)$$

5   where the concentrations are given in $cm^{-3}$, the saturation vapor pressure in Pa, and the nucleation rate is output in $cm^{-3}$. This function was applied within the environmental parameter ranges presented in Tab. 5. The ranges can be considered the ranges within which Eq. (12) is defined. However, because the major part of the nucleation occurs when $[H_2SO_4]$ is high (nearer to the upper boundary than to the lower boundary), a wrong formulation of $J$ in the $[H_2SO_4]$ values lower than $2\times 10^{11}$ $cm^{-3}$ would have only a minor effect on the model outputs. Therefore, an alternative range having $2\times 10^{11}$ $cm^{-3}$ as a minimum

10   boundary for $[H_2SO_4]$ is a more credible range within which the obtained function for $J$ produces reliable results.

**Table 4.** The parameters of the nucleation rate function with the best correspondence between the measured and the simulated data. The ranges of variation represent the resolution with which the exponents were altered during inverse modeling.

| Parameter | Value |
|---|---|
| $k$ | $5.8\times 10^{-26}$ |
| $n_{sa}$ | $1.9\,(\pm 0.1)$ |
| $n_{w}$ | $0.50\,(\pm 0.05)$ |
| $m_{sa}$ | $0.75\,(\pm 0.05)$ |

**Table 5.** The environmental parameter ranges within which the nucleation rate function was applied.

| Parameter | Unit | Lower boundary | Upper boundary |
|---|---|---|---|
| $T$ | °C | $-30$ | $250$ |
| $[H_2SO_4]$ | $cm^{-3}$ | $0\,(2\times 10^{11})^{a}$ | $2\times 10^{14}$ |
| $x_{sa}$ | | $0\,(10^{-8})^{a}$ | $1.1\times 10^{-5}$ |
| $[H_2O]$ | $cm^{-3}$ | $2\times 10^{16}$ | $10^{18}$ |
| $x_{w}$ | | $8\times 10^{-4}$ | $0.04$ |
| RH | % | $0.1$ | $100$ |

[a] Alternative range

Because $p_{sa}{}^{\circ}(T)$ has nearly equal exponential form with the saturation vapor pressure of $H_2O$ ($p_{w}{}^{\circ}(T)$), $p_{sa}{}^{\circ}(T)$ can be expressed approximately using $p_{w}{}^{\circ}(T)$ with

$$p_{sa}{}^{\circ}(T)\approx 2.6\times 10^{-10}p_{w}{}^{\circ}(T)^{2}. \qquad (13)$$



Hence, the magnitude of $J$ remains as in Eq. (12) if it is expressed with $p_\mathrm{w}{}^\circ(T)$ using the form

$$J\left([H_2SO_4],[H_2O],T\right) = 8.9 \times 10^{-19} \frac{[H_2SO_4]^{1.9}[H_2O]^{0.5}}{p_\mathrm{w}{}^\circ(T)^{1.5}} \tag{14}$$

or with both $p_\mathrm{sa}{}^\circ(T)$ and $p_\mathrm{w}{}^\circ(T)$ using, e.g., the form

$$J\left([H_2SO_4],[H_2O],T\right) = 1.4 \times 10^{-23} \frac{[H_2SO_4]^{1.9}[H_2O]^{0.5}}{p_\mathrm{sa}{}^\circ(T)^{0.5}\,p_\mathrm{w}{}^\circ(T)^{0.5}} \tag{15}$$

or a different form

$$J\left([H_2SO_4],[H_2O],T\right) =$$
$$4.0 \times 10^{-25} \left(\frac{[H_2SO_4]}{p_\mathrm{sa}{}^\circ(T)^{0.35}}\right)^{1.9} \left(\frac{[H_2O]}{p_\mathrm{w}{}^\circ(T)^{0.35}}\right)^{0.5}. \tag{16}$$

The exponent $n_\mathrm{sa} = 1.9$ is in agreement with the former studies ($n_\mathrm{sa} = 1\dots2$) and corresponds best with the kinetic nucleation theory (McMurry and Friedlander, 1979) where $n_\mathrm{sa} = 2$. Estimating $n_\mathrm{sa}$ from the measured particle number concentration provided the slope $n_{N_\mathrm{PSM}\,\mathrm{vs.}\,[H_2SO_4]} = 0.4\dots10$. The exponent $n_\mathrm{w}$ estimated from the measurement data is $n_{N_\mathrm{PSM}\,\mathrm{vs.}\,\mathrm{RH_{PTD}}} = 0.1\dots0.2$, which is remarkably lower than the inversely modeled exponent $n_\mathrm{w} = 0.5$. The slope of $N_\mathrm{PSM}$ versus $T_\mathrm{PTD}$ of the measurement set 3b in Fig. 8 is

$$n_{N_\mathrm{PSM}\,\mathrm{vs.}\,T_\mathrm{PTD}} = \frac{\partial \ln N_\mathrm{PSM}}{\partial \ln T_\mathrm{PTD}} = -6\dots-4 \tag{17}$$

but the inversely modeled exponent $m_\mathrm{sa} = 0.75$ corresponds to the slope of $-27$, which is remarkably more negative than $n_{N_\mathrm{PSM}\,\mathrm{vs.}\,T_\mathrm{PTD}}$ due to the same uncertainties as involved with the slopes $n_{N_\mathrm{PSM}\,\mathrm{vs.}\,[H_2SO_4]}$ and $n_{N_\mathrm{PSM}\,\mathrm{vs.}\,\mathrm{RH_{PTD}}}$. In conclusion, inverse modeling provides significantly more accurately the exponents over the method based on the measurement data only.

Nucleation rate was the highest in the PTD where the hot sample and the cold dilution air met. The major part of nucleation occurred in the beginning part of the aging chamber. No noticeable nucleation occurred in the ED though temperature reaches $-30\,^\circ\mathrm{C}$ locally, which is in agreement with the former studies. It provides partial validation for the obtained $m_\mathrm{sa}$ value.

# 6 Conclusions

Homogeneous $H_2SO_4$-$H_2O$ nucleation rate measurements using the modified partial flow sampling system mimicking the dilution process occurring in a real-world driving situation were performed. The aerosol formed in the diluting and cooling sampling system was measured using the PSM, the CPC 3775, and the Nano-SMPS. The particle size distribution near the detection limit of the Nano-SMPS showed clear disagreement with the PSM and the CPC3775 data, with major underestimation of the smaller particles and distortion of the size distribution shape. Thus, due to small particle sizes and simultaneous nucleation and particle growth, the Nano-SMPS data unrealistically suggest log-normal shape for the size distributions. The measurements was simulated with the aerosol dynamics code CFD-TUTMAM using nucleation rate which is explicitly defined as a function of temperature and the concentrations of $H_2SO_4$ and $H_2O$. Equation (2) was used as the functional form



of nucleation rate. The parameters for Eq. (2) which resulted in the best prediction for particle number concentrations and size distributions were $n_{\mathrm{sa}} = 1.9$, $n_{\mathrm{w}} = 0.5$, and $m_{\mathrm{sa}} = 0.75$, thus providing the nucleation rate function Eq. (12) (or any of Eqs. (14) – (16)). As discussed in Sec. 5.3, the obtained exponent $n_{\mathrm{sa}} = 1.9$ may be slightly overestimated in high concentrations and slightly underestimated in low concentrations. Estimating these exponents using only the measured particle concentrations

resulted in markedly higher uncertainties when compared to modeling them inversely using the CFD-TUTMAM code.

The raw sample was generated by evaporating pure $H_2SO_4$ and $H_2O$ liquids. The concentration of $H_2SO_4$ was controlled by adjusting the temperature of the liquid, $T_{\mathrm{sa}}$, and measured by the IC and the CI-APi-TOF. However, there were differences between these measurements and the concentrations estimated using the saturation vapor pressure of $H_2SO_4$ in the temperature of $T_{\mathrm{sa}}$. The differences were caused by high diffusional losses onto the walls of the sampling lines. Due to these differences,

$[H_2SO_4]_{\mathrm{raw}}$ was handled as a fitting parameter to correspond the simulated size distributions with the measured ones. Particle sizes were small with low $T_{\mathrm{sa}}$ and the size distributions were not in a log-normal form. Therefore, the PL+LN model was used to represent the size distributions in the simulations.

In these measurements, particle formation was not observed with the $H_2SO_4$ concentrations below $5.7\times10^{12}\,\mathrm{cm}^{-3}$ at exhaust condition temperatures. However, with real vehicle exhaust, in the same sampling system used here, particle formation

has been observed even with the concentration of $2.5\times10^{9}\,\mathrm{cm}^{-3}$ (Arnold et al., 2012). This indicates that $H_2SO_4$ and $H_2O$ cannot fully control the nucleation process; instead, other compounds, such as hydrocarbons, existing in real exhaust are likely to be involved in the nucleation process as well.

The obtained exponent $n_{\mathrm{sa}} = 1.9$ is in agreement with the former studies ($n_{\mathrm{sa}} = 1\ldots2$) and corresponds best with the kinetic nucleation theory. However, the effects of $[H_2O]$ and $T$ obtained here may differ from the former studies because the effects

are not extensively studied in them. The functional form for the homogeneous $H_2SO_4$-$H_2O$ nucleation rate obtained in this study helps in finding the currently unknown nucleation mechanism occurring in real vehicle or power plant boiler exhaust or in the atmosphere. It provides also the starting point for examining the $H_2SO_4$-$H_2O$ nucleation involved with hydrocarbons, which is likely occurred in real vehicle exhaust. It can also be used in studying the effect of $H_2SO_4$ on the urban air quality by using it in air quality models.

*Author contributions.* M.O., J.A., T.R., and M.D.M. designed the experiments and M.O. and J.A. carried them out. M.O. analyzed the measurement data, developed the model code, and performed the simulations. M.R.T.P. designed the IC analysis. M.O. prepared the manuscript with contributions from all co-authors.

*Competing interests.* The authors declare that they have no conflict of interest.





*Acknowledgements.* This work was funded by Tampere University of Technology Graduate School and by the Maj and Tor Nessling Foundation (project number 2014452). The authors thank CSC and TCSC for the computational time. We also thank prof. Mikko Sipilä from University of Helsinki for lending the CI-inlet for the APi-TOF, the tofTools team for providing tools for mass spectrometry analysis, and M.Sc. Kalle Koivuniemi for Ion Chromatography measurements.



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
