# Peer review of "Inversely modeling homogeneous $H_2SO_4$ - $H_2O$ nucleation rate in exhaust-related conditions"

_Atmospheric Chemistry and Physics, 2018_

## Referee Comment (RC1) · Anonymous Referee #1 · 2 Nov 2018

The manuscript entitled "Inverse modeling homogeneous H2SO4-H2O nucleation rate in exhaust-related conditions" by Miska Olin et al. investigates particle formation in typical vehicle exhaust conditions both experimentally and theoretically. Combined measurements of particle number concentration and size distribution by PSM and nanoSMPS are taken to determine nucleation rate as a function of sulfuric acid concentration (measured by CI-API-ToF-MS and ion chromatography), RH and sulfuric acid saturator temperature. Modeling of the exhaust sampling system together with the aerosol model CFD-TUTMAM lets the authors conclude that binary sulfuric acid-water nucleation is unlikely the dominant mechanism in the particle formation of real-world driving situations. Instead, additional compounds such as hydrocarbons are claimed necessary to explain the observed particle concentrations. While I was positive about

this manuscript initially as this topic is of high relevance to the ACP readership and inverse modeling is also becoming more and more popular nowadays I regret to say that I cannot recommend publication in ACP at this stage.

Beginning with section 2 reporting on the experiments I got the feeling that the authors themselves have no faith in their own experiments/results. Two different instruments for the measurement of sulfuric acid concentration yield substantially different results and it is not clear what the reason for this is. Similarly, the combined size distribution measurements from PSM and nanoSMPS do not produce satisfying agreement in the overlapping size regions. My guess is that the used sizing instruments do not provide the features needed to capture correctly the time scale of the particle dynamics taking place in this exhaust type experiment. To me it seems critical that the experimental data need a much higher confidence level to allow comparison to the modeling results. The concluding section clearly documents the reasons for my concerns about this study. It nicely summarizes what has been done and what problems were encountered but it does not come up with major and firm scientific advances that I would expect from a paper in ACP. The main conclusion seems to be the inappropriate binary H2SO4-H2O nucleation mechanism and that other vapors would be needed for multicomponent nucleation. But this claim is firstly vague, and secondly, not unexpected and thus limited in novelty. In the end I feel that this manuscript in its present form may better fit a technology focused journal but it is certainly not suitable for ACP.

---

## Author Comment (AC1) · 9 Nov 2018

We thank the referee for reading the manuscript and for bringing up the important issues of the manuscript.

We agree that the measurement of sulfuric acid had high uncertainty and the two measurement approaches (CI-APi-TOF and IC) yielded different concentrations (varied within 2 orders of magnitude) due to high diffusional losses when sampling with the CI-APi-TOF. In general, measuring sulfuric acid concentrations is still always a challenging task to date; the major contribution of the uncertainty arising from high losses of sulfuric acid onto the sampling lines before the chemical ionization inside the CI-APi-TOF. For these reasons, we did not use the measured sulfuric acid concentrations

in the simulations; instead, the sulfuric acid concentrations were obtained through inverse modeling by using the measured particle size information. Inverse modeling of the vapor concentrations is possible due to the condensational growth of particles, and the estimation of condensing vapor from the observed growth is widely used method in atmospheric nucleation studies. These simulations could have been performed even without the sulfuric acid measurement still producing the same output. We provided the results from the sulfuric acid measurement as additional, supporting information on the sulfuric acid concentrations for the manuscript. Based on the referee's response, we see that the presentation of the measurement of sulfuric acid would have been better suited as a supplement for the manuscript.

The case of the disagreement of the overlapping size regions measured by the PSM (and the CPC) and the Nano-SMPS is observed elsewhere also (Kulmala et al., 2013; Alanen et al., 2015; Rönkkö et al., 2017). It is caused by drastically underestimated particle concentrations at the smallest particle sizes measured by the Nano-SMPS due to high diffusional losses of small particles inside the device, due to low charging efficiency of small particles, and due to the inversion algorithm which favors features from a log-normal size distribution. It should not be related to the time scale of the particle dynamics in an exhaust type experiment, because this disagreement has been seen in atmospheric measurements also (Kulmala et al., 2013).

The referee found the main conclusion of the manuscript being that binary $H_2SO_4$-$H_2O$ nucleation is unlikely to be the mechanism in vehicle exhaust, with the opinion that this is not a novel conclusion. We would like to state that the main output of the study was meant to present the binary $H_2SO_4$-$H_2O$ nucleation rate as a function of sulfuric acid concentration, water vapor concentration (or RH), and temperature in high sulfuric acid concentrations and temperatures, based on experimental data. Such a rate is currently not available for the relevant temperature and concentration ranges, and therefore we consider that our result is novel and of importance to atmospheric science and thus the manuscript should be suitable for ACP. We agree that the inappropriateness of binary

H2SO4-H2O nucleation mechanism occurring both in the atmosphere and in vehicle exhaust is not a novel finding (Saito et al., 2002; Vaaraslahti et al., 2004; Meyer and Ristovski, 2007; Kerminen et al., 2010; Sipilä et al., 2010); thus, it was not meant to be the main conclusion, but act as additional information and stronger support to the conclusion of former studies related to the nucleation mechanism occurring in vehicle exhaust. It is an additional result of our study; our experiments are also, to our knowledge, the first ones that show directly and experimentally that the formation rate of particles at high sulfuric acid concentrations and in the absence of additional compounds is lower than in H2SO4-containing vehicle exhaust. In light of the referee's comments, we agree that this aspect of our study was highlighted too much in the manuscript to give the feeling of it being the main conclusion. Should we have the opportunity to prepare a revision, we will take the strong criticism of the referee on board.

References:

Alanen, J., Saukko, E., Lehtoranta, K., Murtonen, T., Timonen, H., Hillamo, R., Karjalainen, P., Kuuluvainen, H., Harra, J., Keskinen, J., and Rönkkö, T.: The formation and physical properties of the particle emissions from a natural gas engine, Fuel, 162, 155-161, https://doi.org/10.1016/j.fuel.2015.09.003, 2015.

Kerminen, V.-M., Petäjä, T., Manninen, H. E., Paasonen, P., Nieminen, T., Sipilä, M., Junninen, H., Ehn, M., Gagné, S., Laakso, L., Riipinen, I., Vehkamäki, H., Kurten, T., Ortega, I. K., Dal Maso, M., Brus, D., Hyvärinen, A., Lihavainen, H., Leppä, J., Lehtinen, K. E. J., Mirme, A., Mirme, S., Hõrrak, U., Berndt, T., Stratmann, F., Birmili, W., Wiedensohler, A., Metzger, A., Dommen, J., Baltensperger, U., Kiendler-Scharr, A., Mentel, T. F., Wildt, J., Winkler, P. M., Wagner, P. E., Petzold, A., Minikin, A., Plass-Dülmer, C., Pöschl, U., Laaksonen, A., and Kulmala, M.: Atmospheric nucleation: highlights of the EUCAARI project and future directions, Atmos. Chem. Phys., 10, 10829-10848, https://doi.org/10.5194/acp-10-10829-2010, 2010.

Kulmala, M., Kontkanen, J., Junninen, H., Lehtipalo, K., Manninen, H. E., Nieminen,

T., Petäjä, T., Sipilä, M., Schobesberger, S., Rantala, P., Franchin, A., Jokinen, T., Järvinen, E., Äijälä, M., Kangasluoma, J., Hakala, J., Aalto, P. P., Paasonen, P., Mikkilä, J., Vanhanen, J., Aalto, J., Hakola, H., Makkonen, U., Ruuskanen, T., Mauldin, R. L., Duplissy, J., Vehkamäki, H., Bäck, J., Kortelainen, A., Riipinen, I., Kurtén, T., Johnston, M. V., Smith, J. N., Ehn, M., Mentel, T. F., Lehtinen, K. E. J., Laaksonen, A., Kerminen, V.-M., and Worsnop, D. R.: Direct Observations of Atmospheric Aerosol Nucleation, Science, 339, 943-946, https://doi.org/10.1126/science.1227385, 2013.

Meyer, N. K. and Ristovski, Z. D.: Ternary Nucleation as a Mechanism for the Production of Diesel Nanoparticles:  Experimental Analysis of the Volatile and Hygroscopic Properties of Diesel Exhaust Using the Volatilization and Humidification Tandem Differential Mobility Analyzer, Environ. Sci. Technol., 41, 7309-7314, https://doi.org/10.1021/es062574v, 2007.

Rönkkö, T., Kuuluvainen, H., Karjalainen, P., Keskinen, J., Hillamo, R., Niemi, J. V., Pirjola, L., Timonen, H. J., Saarikoski, S., Saukko, E., Järvinen, A., Silvennoinen, H., Rostedt, A., Olin, M., Yli-Ojanperä, J., Nousiainen, P., Kousa, A., and Dal Maso, M.: Traffic is a major source of atmospheric nanocluster aerosol, P. Natl. Acad. Sci. USA, 201700830, https://doi.org/10.1073/pnas.1700830114, 2017.

Saito, K., Shinozaki, O., Seto, T., Kim, C.-S., Okuyama, K., Kwon, S.-B., and Lee, K. W.: The Origins of Nanoparticle Modes in the Number Distribution of Diesel Particulate Matter, SAE Tech. Paper Ser., 2002-01-1008, https://doi.org/10.4271/2002-01-1008, 2002.

Sipilä, M., Berndt, T., Petäjä, T., Brus, D., Vanhanen, J., Stratmann, F., Patokoski, J., Mauldin, R. L. III, Hyvärinen, A.-P., Lihavainen., H., and Kulmala, M.: The Role of Sulfuric Acid in Atmospheric Nucleation, Science, 327, 1243-1246, https://doi.org/10.1126/science.1180315, 2010.

Vaaraslahti, K., Virtanen, A., Ristimäki, J., and Keskinen, J.: Nucleation Mode Formation in Heavy-Duty Diesel Exhaust with and without a Particulate Filter, Environ. Sci.

Technol., 38, 4884-4890, https://doi.org/10.1021/es0353255, 2004.

---

## Referee Comment (RC2) · Anonymous Referee #2 · 3 Dec 2018

New particle formation in vehicle or other exhaust flows as well in atmosphere, is one of the least understood aerosol processes. Regarding vehicle exhaust, the measurements indicate that although sulfuric acid participates in the production of volatile particles, it cannot explain the measured particle number size distributions. So, volatile and/or semi-volatile condensable vapours other than sulfuric acid are required to explain the measurements. This is the general story of the exhaust nanoparticles science. And the questions are: Which are these other vapours? And, which are the alternative mechanisms? Several studies, both experimental and theoretical, have been conducted to answer these questions and, although the levels of instrumentation and of modelling sophistication and complexity have significantly increased over the recent years, this is not the case for the level of a clear understanding of the processes. At

least this is my perspective, and I think that this is also the view of the authors, at least as indicated by their conclusions (page 31/38, lines 15-17): "[...] This indicates that H2SO4 and H2O cannot fully control the nucleation process; instead, other compounds, such as hydrocarbons, existing in real exhaust are likely".

Irrespective of the above general statement, the article is interesting, it can contribute to further research and can be considered for publication in the journal. Nevertheless, I think that the authors should be more convincing when presenting the need for the proposed study. Below, my detailed comments:

Title – good, reflects the content of the paper

Abstract – good, describes the essential information in the work.

Introduction – It is rather well written and in sufficient detail. Very good presentation of the work already done in the field. I have some remarks, however: • Pages (2-3)/38: It will also be useful if the authors reconstruct slightly the text. More specifically: it would be better for the reader if the authors had continued the description of binary homogeneous nucleation (BHN) after line 33 in page 2 with the text of page 3 (lines 6-12: "The derivation of the CNT contains, however . . .. with experimental nucleation rates") and had started a new distinct paragraph with the details of the other nucleation mechanisms (i.e., from "Conversely, the nucleation rates of the other nucleation mechanisms" up to "thus, a constant coefficient cannot be used." An then from (page 3 line 12) : "The nucleation exponents, n, for H2SO4 obtained" up to "much lower than the theoretical exponents (n > 5) (here, references are needed)." • Page 3/38, line 1: Explaining equation 1, the authors write "[...] and n is the nucleation exponent presenting the sensitivity of [H2SO4] on J." It would be useful for the reader to understand better sensitivity if the authors add that, in theory, the nucleation exponent n represents the number of nucleating molecules in the critical nuclei. • Page 3/38, line 15: "The first attempts to obtain the nucleation. . .." This is not exact. Vouitsis et al. (Modelling of Diesel Exhaust Aerosol during Laboratory Sampling. Atmos. Environ.39, 2005) showed that the

barrier-free nucleation scheme, where clusters are always stable against evaporation, could predict the nucleation mode particles concentration rather well for low sulphur fuel (<10 ppm), whereas a nucleation rate proportional to the square of sulphuric acid saturation vapour pressure was more appropriate for high sulphur fuel (250 ppm).

The research question is stated in a somewhat unclear and contradictory manner. The authors note (page 4/38 lines 9-14) that their research focuses on "pure H2SO4-H2O nucleation instead of nucleation associated with some unknown compounds existing in real vehicle exhaust" on the basis of the very low level knowledge of the H2SO4-H2O nucleation mechanism; however, some lines further (lines29-29) they state that "The formulation obtained from this study helps in finding the nucleation mechanisms occurring in real vehicle exhaust or in the atmosphere, etc". I think that the argument must be rephrased in a more proper statement.

Materials, methods, discussion - The methodology presented in the manuscript and the analysis provided are both accurate and appropriate. The experimental section is robust and the methods applied are presented in sufficient detail. The observations made are discussed comprehensively and the data presented are adequate to support the conclusions. The same is valid for the simulation section of the study. The authors have developed a very interesting model which accounts for the flow and the temperature field of the sampling system, for the solution the aerosol processes, for the correction of water amount present in the particles and for diffusional losses in the sampling as well as for the detection efficiencies of particle counting devices. One comment only: in two cases, (page 30/38, line 8 and page 3/38m line 18), the authors write "The obtained exponent nsa = 1:9 is in agreement with the former studies". However, they do not refer to any study specifically, unless if they mean the article of McMurry and Friedlander (1979), on which they referred in the first case. If this is true, they must be more specific in relation with the observations of McMurry and Friedlander and their connection with the results of this study.

Conclusions – I have a similar and related with the above comment in the introduction

section concerning the scope and clarity of the study. The authors write: "[. . .] nucleation rate obtained in this study helps in finding the currently unknown nucleation mechanism occurring in real vehicle or power plant boiler exhaust or in the atmosphere". The authors must clarify better how this help is provided.

---

## Author Comment (AC2) · 16 Jan 2019

We thank the referee for careful reading of the manuscript and for giving detailed comments for clarifying it.

The referee firstly highlighted a sentence from our conclusions about the inappropriateness of pure binary H2SO4-H2O nucleation mechanism controlling particle formation in vehicle exhaust. As we already replied on the first referee's comment about this inappropriateness being the main conclusion of this study, it was not meant to be the main conclusion; instead, presenting the function for the binary H2SO4-H2O nucleation rate in exhaust-related conditions was meant. Neither was the finding of the other vapors or nucleation mechanisms participating in the particle formation process in vehicle ex-

haust the purpose of this study pointed out by the second referee. Instead, because even the physics behind the pure binary $H_2SO_4$-$H_2O$ nucleation mechanism is very uncertain to date, we focused on the nucleation rate of that, which could act as a starting point in examining the actual nucleation rate in the future. We see that, according to the comments from the both referees, this conclusion was highlighted too much in the manuscript to give the feeling of it being the main conclusion; nevertheless, we will rephrase it in the revised manuscript.

The current version of the manuscript did not tell clearly enough how the nucleation rate function resulted from this study can help in finding the actual nucleation mechanism occurring in vehicle exhaust. Firstly, performing inverse modeling with four free parameters ($k$, $nsa$, $nw$, and $msa$) to be fitted using CFD is already computationally very expensive. Thus, including other compounds, such as hydrocarbons, in the nucleation mechanism to be examined, will increase the computational expense significantly. Therefore, using the nucleation rate function, obtained in this study, as a starting point for a more complex nucleation rate function, will decrease the computational time in that kind of inverse modeling task. Secondly, the parameters of the obtained function for pure $H_2SO_4$-$H_2O$ nucleation rate can also give details on the currently poorly understood physics behind the pure $H_2SO_4$-$H_2O$ nucleation mechanism, for example, with the values of the nucleation exponents of 1.9 and 0.5 for $H_2SO_4$ and for $H_2O$, respectively. The exponent of 1.9 for $H_2SO_4$ suggests that the nucleation can be driven by kinetic nucleation, which is the collisions between two $H_2SO_4$ molecules; however, the exponent of 0.5 for $H_2O$ does not match with any currently known nucleation mechanisms and raises thus new ideas for the nucleation theories. Finally, the obtained function can be used to improve the effect of traffic or power generation on particle concentrations in urban air. These clarifications will be added to the revised manuscript.

Other minor comments from the referee, such as reordering of the text describing different nucleation mechanisms, noting that the nucleation exponent is related to the

number of nucleating molecules in the critical nuclei, and correcting and adding appropriate references, will be taken into account when preparing the revised manuscript.

---

## Author Response (AR1)

**Authors' response to comments of Olin et al.: "Inversely modeling homogeneous $H_2SO_4$-$H_2O$ nucleation rate in exhaust-related conditions"**

We thank the referees for their detailed and very useful comments, and have corrected the manuscript according to them.

Referee reports are in *black italic* and authors' responses are in blue roman font. The changes to the manuscript provided as the marked-up manuscript, the new version of the manuscript, and the Supplement are included at the end of this file.

**Referee 1 comments:**

*The manuscript entitled "Inverse modeling homogeneous H2SO4-H2O nucleation rate in exhaust-related conditions" by Miska Olin et al. investigates particle formation in typical vehicle exhaust conditions both experimentally and theoretically. Combined measurements of particle number concentration and size distribution by PSM and nanoSMPS are taken to determine nucleation rate as a function of sulfuric acid concentration (measured by CI-API-ToF-MS and ion chromatography), RH and sulfuric acid saturator temperature. Modeling of the exhaust sampling system together with the aerosol model CFD-TUTMAM lets the authors conclude that binary sulfuric acid-water nucleation is unlikely the dominant mechanism in the particle formation of real-world driving situations. Instead, additional compounds such as hydrocarbons are claimed necessary to explain the observed particle concentrations. While I was positive about this manuscript initially as this topic is of high relevance to the ACP readership and inverse modeling is also becoming more and more popular nowadays I regret to say that I cannot recommend publication in ACP at this stage.*

*Beginning with section 2 reporting on the experiments I got the feeling that the authors themselves have no faith in their own experiments/results. Two different instruments for the measurement of sulfuric acid concentration yield substantially different results and it is not clear what the reason for this is.*

We agree that the measurement of sulfuric acid had high uncertainty and the two measurement approaches (CI-APi-TOF and IC) yielded different concentrations (varied within 2 orders of magnitude) due to high diffusional losses when sampling with the CI-APi-TOF. In general, measuring sulfuric acid concentrations is still always a challenging task to date; the major contribution of the uncertainty arising from high losses of sulfuric acid onto the sampling lines before the chemical ionization inside the CI-APi-TOF. For these reasons, we did not use the measured sulfuric acid concentrations in the simulations; instead, the sulfuric acid concentrations were obtained through inverse modeling by using the measured particle size information. Inverse modeling of the vapor concentrations is possible due to the condensational growth of particles, and the estimation of condensing vapor from the observed growth is widely used method in atmospheric nucleation studies. These simulations could have been performed even without the sulfuric acid measurement still producing the same output. We provided the results from the sulfuric acid measurement as additional, supporting information on the sulfuric acid concentrations for the manuscript. Based on the referee's response, we see that the presentation of the measurement of sulfuric acid would have been better suited as a supplement for the manuscript. The part of the manuscript related to the measurement of sulfuric acid is now moved to the Supplement.

*Similarly, the combined size distribution measurements from PSM and nanoSMPS do not produce satisfying agreement in the overlapping size regions. My guess is that the used sizing instruments do not provide the features needed to capture correctly the time scale of the particle dynamics taking place in this exhaust type experiment. To me it seems critical that the experimental data need a much higher confidence level to allow comparison to the modeling results.*

The case of the disagreement of the overlapping size regions measured by the PSM (and the CPC) and the Nano-SMPS is observed elsewhere also (Kulmala et al., 2013; Alanen et al., 2015; Rönkkö et al., 2017). It is caused by drastically underestimated particle concentrations at the smallest particle sizes measured by the Nano-SMPS due to high diffusional losses of small particles inside the device, due to low charging efficiency of small particles, and due to the inversion algorithm which favors

features from a log-normal size distribution. It should not be related to the time scale of the particle dynamics in an exhaust type experiment, because this disagreement has been seen in atmospheric measurements also (Kulmala et al., 2013).

The text related to this issue is now changed from: "The smaller diameter edges of the log-normal size distributions measured by the Nano-SMPS do not connect with the distributions measured by the PSM and the CPC 3775 due to high diffusional losses of very small particles inside the Nano-SMPS device. Thus, the smaller diameter edges of the measured log-normal size distributions are not accurate." to: "The smaller diameter edges of the log-normal size distributions measured by the Nano-SMPS do not connect with the distributions measured by the PSM and the CPC 3775 due to high diffusional losses of very small particles inside the Nano-SMPS device, due to low charging efficiency of small particles, and due to the inversion algorithm of the device which favors features from a log-normal size distribution. Thus, the smaller diameter edges of the measured log-normal size distributions are not accurate. Similar disagreements of the data from these devices have been observed elsewhere also both in exhaust-related (Alanen et al., 2015; Rönkkö et al., 2017) and in atmospherically-related studies (Kulmala et al., 2013).".

*The concluding section clearly documents the reasons for my concerns about this study. It nicely summarizes what has been done and what problems were encountered but it does not come up with major and firm scientific advances that I would expect from a paper in ACP. The main conclusion seems to be the inappropriate binary H2SO4- H2O nucleation mechanism and that other vapors would be needed for multicomponent nucleation. But this claim is firstly vague, and secondly, not unexpected and thus limited in novelty. In the end I feel that this manuscript in its present form may better fit a technology focused journal but it is certainly not suitable for ACP.*

The referee found the main conclusion of the manuscript being that binary $H_2SO_4$-$H_2O$ nucleation is unlikely to be the mechanism in vehicle exhaust, with the opinion that this is not a novel conclusion. We would like to state that the main output of the study was meant to present the binary $H_2SO_4$-$H_2O$ nucleation rate as a function of sulfuric acid concentration, water vapor concentration (or RH), and temperature in high sulfuric acid concentrations and temperatures, based on experimental data. Such a rate is currently not available for the relevant temperature and concentration ranges, and therefore we consider that our result is novel and of importance to atmospheric science and thus the manuscript should be suitable for ACP. We agree that the inappropriateness of binary $H_2SO_4$-$H_2O$ nucleation mechanism occurring both in the atmosphere and in vehicle exhaust is not a novel finding (Saito et al., 2002; Vaaraslahti et al., 2004; Meyer and Ristovski, 2007; Kerminen et al., 2010; Sipilä et al., 2010); thus, it was not meant to be the main conclusion, but act as additional information and stronger support to the conclusion of former studies related to the nucleation mechanism occurring in vehicle exhaust. It is an additional result of our study; our experiments are also, to our knowledge, the first ones that show directly and experimentally that the formation rate of particles at high sulfuric acid concentrations and in the absence of additional compounds is lower than in $H_2SO_4$-containing vehicle exhaust. In light of the referee's comments, we agree that this aspect of our study was highlighted too much in the manuscript to give the feeling of it being the main conclusion. The strong criticism of the referee has now been taken on board in the revised manuscript.

All changes to the manuscript related to this issue can be seen in the marked-up manuscript at the end of this file. For example, in Abstract, the sentence: "Results imply that the nucleation process of volatile nanoparticles in real vehicle exhaust cannot be fully explained by sulfuric acid; instead, it is likely that other compounds, e.g., hydrocarbons, are involved as well." is replaced with the sentences: "The obtained function can be used as a starting point for inverse modeling studies of more complex nucleation mechanisms involving extra compounds in addition to sulfuric acid and water. More complex nucleation mechanisms, such as hydrocarbon-involving, are observed with real vehicle exhaust and are also supported by the results obtained in this study." Also, in Introduction, the sentence: "Neglecting these unknown compounds is reasonable at this stage of nucleation studies ..." is changed to: "Although the pure binary nucleation seems not to be the principal nucleation mechanism in real exhaust (Saito et al., 2002; Vaaraslahti et al., 2004; Meyer and Ristovski, 2007; Pirjola et al., 2015), neglecting the unknown compounds is reasonable at this stage of nucleation studies ..." and sentences: "Additionally, although there are studies suggesting that other compounds are involved in the nucleation process in real vehicle exhaust, it has not yet been directly shown that nucleation rate would be lower or higher with the absence of those compounds. Comparing the experiments with pure $H_2SO_4$-$H_2O$ nucleation to the experiments with real exhaust can provide information on that." are added to clarify that the inappropriateness of the binary $H_2SO_4$-$H_2O$ nucleation mechanism is not a new result from this study but the lower

nucleation rate of pure $H_2SO_4$-$H_2O$ nucleation compared to nucleation rate in real exhaust is. Finally, in Conclusions, the part: "... other compounds, such as hydrocarbons, existing in real exhaust are likely to be involved in the nucleation process as well." is complemented with a sentence: "... which is in agreement with the former exhaust-related nucleation studies (Saito et al., 2002; Vaaraslahti et al., 2004; Meyer and Ristovski, 2007; Pirjola et al., 2015; Olin et al., 2015)."

**Referee 2 comments:**

*New particle formation in vehicle or other exhaust flows as well in atmosphere, is one of the least understood aerosol processes. Regarding vehicle exhaust, the measurements indicate that although sulfuric acid participates in the production of volatile particles, it cannot explain the measured particle number size distributions. So, volatile and/or semi-volatile condensable vapours other than sulfuric acid are required to explain the measurements. This is the general story of the exhaust nanoparticles science. And the questions are: Which are these other vapours? And, which are the alternative mechanisms? Several studies, both experimental and theoretical, have been conducted to answer these questions and, although the levels of instrumentation and of modelling sophistication and complexity have significantly increased over the recent years, this is not the case for the level of a clear understanding of the processes. At least this is my perspective, and I think that this is also the view of the authors, at least as indicated by their conclusions (page 31/38, lines 15-17): "[...] This indicates that H2SO4 and H2O cannot fully control the nucleation process; instead, other compounds, such as hydrocarbons, existing in real exhaust are likely".*

As already replied on the first referee's comment about the inappropriateness of pure binary $H_2SO_4$-$H_2O$ nucleation mechanism controlling particle formation in vehicle exhaust being the main conclusion of this study, it was not meant to be the main conclusion; instead, presenting the function for the binary $H_2SO_4$-$H_2O$ nucleation rate in exhaust-related conditions was meant. Neither was the finding of the other vapors or nucleation mechanisms participating in the particle formation process in vehicle exhaust the purpose of this study. Instead, because even the physics behind the pure binary $H_2SO_4$-$H_2O$ nucleation mechanism is very uncertain to date, we focused on the nucleation rate of that, which could act as a starting point in examining the actual nucleation rate in the future. We see that, according to the comments from the both referees, this conclusion was highlighted too much in the manuscript to give the feeling of it being the main conclusion; nevertheless, it is now rephrased in the revised manuscript. All changes to the manuscript related to this issue can be seen in the marked-up manuscript at the end of this file, and the most important changes were pointed out in the previous blue text.

*Irrespective of the above general statement, the article is interesting, it can contribute to further research and can be considered for publication in the journal. Nevertheless, I think that the authors should be more convincing when presenting the need for the proposed study. Below, my detailed comments:*

*Title – good, reflects the content of the paper*

*Abstract – good, describes the essential information in the work.*

*Introduction – It is rather well written and in sufficient detail. Very good presentation of the work already done in the field. I have some remarks, however: Pages (2-3)/38: It will also be useful if the authors reconstruct slightly the text. More specifically: it would be better for the reader if the authors had continued the description of binary homogeneous nucleation (BHN) after line 33 in page 2 with the text of page 3 (lines 6-12: "The derivation of the CNT contains, however ... . with experimental nucleation rates") and had started a new distinct paragraph with the details of the other nucleation mechanisms (i.e., from "Conversely, the nucleation rates of the other nucleation mechanisms" up to "thus, a constant coefficient cannot be used." An then from (page 3 line 12) : "The nucleation exponents, n, for H2SO4 obtained" up to "much lower than the theoretical exponents (n > 5) (here, references are needed)."*

The proposed changes have now been made to the revised manuscript. The reference (Vehkamäki et al., 2003) for the theoretical exponents ($n > 5$) is also added.

*Page 3/38, line 1: Explaining equation 1, the authors write "[...] and n is the nucleation exponent presenting the sensitivity of [H2SO4] on J." It would be useful for the reader to understand better sensitivity if the authors add that, in theory, the nucleation exponent n represents the number of nucleating molecules in the critical nuclei.*

A sentence: "According to the first nucleation theorem (Kashchiev, 1982), $n$ is also connected to the number of molecules in a critical cluster; however, due to assumptions included in the theorem, $n$ is not exactly the number of molecules in a critical cluster in realistic conditions (Kupiainen-Määttä et al., 2014)." is now added after the sentence pointed out.

*Page 3/38, line 15: "The first attempts to obtain the nucleation... ." This is not exact. Vouitsis et al. (Modelling of Diesel Exhaust Aerosol during Laboratory Sampling. Atmos. Environ.39, 2005) showed that the barrier-free nucleation scheme, where clusters are always stable against evaporation, could predict the nucleation mode particles concentration rather well for low sulphur fuel (<10 ppm), whereas a nucleation rate proportional to the square of sulphuric acid saturation vapour pressure was more appropriate for high sulphur fuel (250 ppm).*

Because Vouitsis et al. (2005) did not obtain the nucleation rates inversely, but tested what their model outputs using previously determined nucleation rates, we changed the sentence: "The first attempts to obtain the nucleation rate in vehicle exhaust, other than predicted by the CNT, were performed by Olin et al. (2015) and Pirjola et al. (2015)" to: "The first step in examining nucleation mechanisms, other than the CNT, in vehicle exhaust using experimental data was performed by Vouitsis et al. (2005). They concluded that nucleation mechanisms having $n = 2$, including barrierless kinetic nucleation mechanism, can predict nucleation rates in vehicle exhaust. Later, Olin et al. (2015) and Pirjola et al. (2015) focused on obtaining nucleation rates inversely ..."

*The research question is stated in a somewhat unclear and contradictory manner. The authors note (page 4/38 lines 9-14) that their research focuses on "pure H2SO4-H2O nucleation instead of nucleation associated with some unknown compounds existing in real vehicle exhaust" on the basis of the very low level knowledge of the H2SO4- H2O nucleation mechanism; however, some lines further (lines29-29) they state that "The formulation obtained from this study helps in finding the nucleation mechanisms occurring in real vehicle exhaust or in the atmosphere, etc". I think that the argument must be rephrased in a more proper statement.*

The issue related to the main conclusion of this study is discussed in the last blue text of the first referee's comments.

*Materials, methods, discussion - The methodology presented in the manuscript and the analysis provided are both accurate and appropriate. The experimental section is robust and the methods applied are presented in sufficient detail. The observations made are discussed comprehensively and the data presented are adequate to support the conclusions. The same is valid for the simulation section of the study. The authors have developed a very interesting model which accounts for the flow and the temperature field of the sampling system, for the solution the aerosol processes, for the correction of water amount present in the particles and for diffusional losses in the sampling as well as for the detection efficiencies of particle counting devices. One comment only: in two cases, (page 30/38, line 8 and page 3/38m line 18), the authors write "The obtained exponent nsa = 1:9 is in agreement with the former studies". However, they do not refer to any study specifically, unless if they mean the article of McMurry and Friedlander (1979), on which they referred in the first case. If this is true, they must be more specific in relation with the observations of McMurry and Friedlander and their connection with the results of this study.*

The reference (McMurry and Friedlander, 1979) was not considered "the former studies" here. It was meant to be the reference for the kinetic nucleation theory, which has the exponent of 2. To clarify this, the first occurrence: "The exponent $n_{sa} = 1.9$ is in agreement with the former studies ($n_{sa} = 1...2$) and corresponds best with the kinetic nucleation theory (McMurry and Friedlander, 1979) where $n_{sa} = 2$." is now replaced with: "The exponent $n_{sa} = 1.9$ is in agreement with the former nucleation studies related to vehicle exhaust (Vouitsis et al., 2005) or to the atmosphere (Sihto et al., 2006; Riipinen et al., 2007; Brus et al., 2011; Riccobono et al., 2014) where $n_{sa}$ lies usually between 1 and 2. The exponent $n_{sa} = 1.9$ corresponds best with the kinetic nucleation theory (McMurry and Friedlander, 1979) where $n_{sa} = 2$.".

*Conclusions – I have a similar and related with the above comment in the introduction section concerning the scope and clarity of the study. The authors write: "[...] nucleation rate obtained in this study helps in finding the currently unknown nucleation mechanism occurring in real vehicle or power plant boiler exhaust or in the atmosphere". The authors must clarify better how this help is provided.*

A clarifying sentence on how the results of this study help in finding the nucleation mechanism is now added to Introduction: "E.g., the values of the nucleation exponents obtained in this study can provide information on the nucleation mechanisms because the values differ with respect to different nucleation mechanisms." Additionally, a clarification on how the results improve air quality models is also added: "Another use of the formulation is in improving air quality models by using it to model the effect of sulfuric acid-emitting traffic and power generation on the particle concentration in urban air".

**Additional changes:**

There are also some minor language corrections in the revised manuscript which are also seen in the marked-up manuscript at the end of this file.

**References**

[revised manuscript text omitted]

---

## Author Response (AR2)

**Authors' response to the referee's comment for Olin et al.: "Inversely modeling homogeneous $H_2SO_4$-$H_2O$ nucleation rate in exhaust-related conditions"**

We thank the referee for a beneficial comment and have corrected the manuscript according to it.

Referee report is in *black italic* and authors' response is in blue roman font. The marked-up manuscript and Supplement showing the changes and the new versions are included at the end of this file.

**Referee 1 comments:**

*The authors have carefully addressed my criticism and revised the manuscript accordingly. Its message has been significantly clarified now and it might therefore be suitable for publication in ACP. Still, I have some concerns about the complexity of uncertainties in this study. It's not only the measurement of the sulfuric acid concentration that is still a challenging task but also size distribution measurement in the few nanometer size range is typically far from being quantitative. The size distribution results presented in this study are the best example. The authors themselves summarize all the critical aspects of size distribution measurement such as charging probability, sampling losses and instrument related transfer functions. The measured signal (number concentration at a certain DMA voltage) needs a sophisticated INVERSION algorithm to finally obtain a size distribution. In that sense it may be questioned whether the inverse modeling really brings the benefit one would hope for. Does the inversion of the inversion reduce error bars after all? I guess a careful analysis of measurement uncertainties is obligatory to get some feeling on the reliability of the results.*

We have made a more careful examination on the particle size distributions in the smallest particle size range. Firstly, we found that there has been an error in our code used to import the Nano-SMPS data: the particle diameter vector has been misaligned with the concentration matrix; thus, all size distributions have been shifted towards larger diameters with a factor of about 1.2. After correcting this error, all Nano-SMPS distributions are now in 1.2 times smaller sizes. Fortunately, this correction narrows the gaps between the distributions obtained from the PSM+CPC system and from the Nano-SMPS, as is seen in Fig. AR1. Additionally, we have changed the CPC 3776 detection efficiency curve from the one reported by the manufacturer to the curve measured by Mordas et al. (2008) because it seems that the curve can deviate clearly from the manufacturer's curve, as seen by Hermann et al. (2007) and Mordas et al. (2008).

[Figure]

**Figure AR1.** The change of the example size distributions after the correction of the error in the code.

However, there are still some discrepancies between the distributions after the correction. We have made a careful analysis of the uncertainties involved in the size distribution measurements by calculating the error bars for all the size distribution points at specific diameters. For the PSM+CPC system, systematical errors, due to the uncertainties in the detection efficiency curves and in the diffusional loss correction function, are taken into account. For the Nano-SMPS system, systematical errors, due to the uncertainties in the radioactive charger efficiency, in the CPC 3776 detection efficiency curve, and in the diffusional loss correction function, are taken into account. Random error, caused by the instability in the particle generation and by the low counting statistics of the Nano-SMPS for the particle sizes having very low detection efficiency or low concentration, are also taken into account for the both devices. The detailed information on calculating the error bars is now included in the Supplement; the error bars for the example distributions are also shown here in Fig. AR2. Error bars are now added also to some main manuscript's figures for which the clarity of the figures can be maintained; the remaining figures with the error bars are added to the Supplement.

High error bars in the PSM+CPC distributions arise when the standard deviation of the measured concentrations are at a same level as the difference between the concentrations measured with the adjacent cut-diameters. Therefore, using all the measured concentrations typically cause high error bars if there is instability in the measured signal. Alternative way to obtain the size distribution in the PSM+CPC size range is to use only the concentrations measured with the smallest and with the largest cut-diameter. The alternative method will produce smaller error bars (shown as green shaded areas), but this will, of course, diminish the information on the shape of the distribution within that size range. Considering the error bars of the distributions, it seems that in the cases in panes (a) and (b), there are particles in the PSM+CPC size range although the Nano-SMPS distributions show log-normal-like edges for the smaller sizes. In the case in pane (c), although there are two size distribution points in the PSM+CPC distribution, particles in that size range are, according to the alternative method, inexistent. However, high error bars for the alternative method, caused by the standard deviation of the measured concentrations due to instabilities in particle generation, denote that the probability of the existence of particle in that size range is high. Nevertheless, the fraction of particles in that size range compared to the total particle count is, definitely, some orders of magnitude smaller than in the cases in panes (a) and (b). Due to the incapability of the Nano-SMPS in determining the size distribution reliably

in sub-10 nm diameter range, in all of the cases studied here, the PSM+CPC system is more suitable than the Nano-SMPS in measuring the size distribution for that particle size range.

[Figure]

**Figure AR2.** The example size distributions shown with the error bars.

The inverse modeling in this work is based on predicting the concentrations measured with the different saturator flow rates of the PSM and with the CPC 3775 and on predicting the size distribution which is reported by the Nano-SMPS software. In other words, the inverse modeling does not try to predict the concentrations measured by the CPC 3776, acting as a particle counter in the Nano-SMPS system, as a function of time, or as a function of a specific DMA voltage. Concluding, there are two distinct parts of inversion involved: (1) the inverse modeling performed in this work and (2) the inversion algorithm which is included in the software of the Nano-SMPS device. These two inversion parts are not overlapped; thus, there is no "inversion of the inversion" in the analysis. The inverse modeling in this work takes the diffusional losses in the sampling lines and the detection efficiencies of the particle counters into account, while the inversion algorithm of the Nano-SMPS device takes at least the charger efficiency and the diffusional losses inside the device into account.

A consequence from correcting the error in the Nano-SMPS importing code is that because the particle diameters are now smaller, the diameters with the average mass of the distributions are also smaller now. Therefore, Fig. 13(b) and its interpretation is slightly changed now, as seen in Fig. AR3. The measured $D_{\bar{m}}$ values are now shifted towards the smaller diameters, which causes that the smallest diameters have now better agreement with the simulated diameters, but however, some diameters become less agreed with the simulated ones. The lining of the points is, nevertheless, the same: the fitted points form a slightly curved line in which the mid-ranged sizes are slightly overestimated. As some of the points now lie on the other side of the 1:1 line, minor changes to the text related to this figure are also made, but the final interpretation still remains as before.

[Figure]

**Figure AR3.** The change of the diameters with the average mass after the correction of the error in the code. The error bars are also added, and the colors are also changed for better clarity.

**References**

Hermann, M., Wehner, B., Bischof, O., Han, H.-S., Krinke, T., Liu, W., Zerrath, A., and Wiedensohler, A.: Particle counting efficiencies of new TSI condensation particle counters, J. Aerosol Sci., 38, 674 – 682, https://doi.org/https://doi.org/10.1016/j.jaerosci.2007.05.001, http://www.sciencedirect.com/science/article/pii/S0021850207000705, 2007.

[revised manuscript text omitted]

Nevertheless, both the measured data sets agree well with the shape of the theoretical curve, which implies that $[H_2SO_4]_{raw}$ can be estimated using $T_{sa}$. However, the absolute value for $[H_2SO_4]_{raw}$ cannot be satisfactorily estimated using neither $T_{sa}$ nor the measured concentrations due to the discrepancy of the measured concentrations. Therefore, the simulations of this study did not use the measured concentrations as the boundary conditions; instead, the $[H_2SO_4]_{raw}$ values were obtained through inverse modeling.

[Figure]

**Figure S1.** Simulated sulfuric acid concentrations in the raw sample compared to the measured and the theoretical concentrations with different sulfuric acid evaporator temperatures.

**2  Error estimation for particle size distribution measurements**

The disagreement of sub-6 nm particle size distributions measured by the combination of the PSM and the CPC 3775 and by the Nano-SMPS is examined by estimating the sources and the magnitudes of errors in measured concentrations for these devices.

**2.1  Calculating the error bars for the combination of the PSM and the CPC 3775**

The particle number size distributions are calculated using the step-wise method according to Lehtipalo et al. (2014) . Backwards-correcting the measured distributions to represent the distributions after the ED requires multiplying the data with the DR of the BD and dividing by the penetration efficiency of particles in the sampling lines between the ED and the measurement devices. Finally, the equation to obtain the distribution at particle size $D_{p,i}$ after the ED from the measured concentrations is

$$\nu(D_{p,i}) = \frac{N_i^{smaller} - N_i^{larger}}{\log(D_{p,i}^{larger}/D_{p,i}^{smaller}) \cdot p(D_{p,i}, L/Q)} \tag{S2}$$

where $N_i^{\text{smaller}}$ and $N_i^{\text{larger}}$ are particle number concentrations measured with the $D_{50}$-cut-sizes of $D_{\text{p},i}^{\text{smaller}}$ and $D_{\text{p},i}^{\text{larger}}$, respectively. $D_{\text{p},i}$ is the geometric mean diameter of the $D_{50}$-cut-sizes and $p(D_{\text{p},i}, L/Q)$ is the penetration efficiency of particles with a diameter of $D_{\text{p},i}$ in a sampling line with a length of $L_{\text{lines}} = L$ and flow rate of $Q_{\text{lines}} = Q$, according to diffusional losses calculated with the equations of Gormley and Kennedy (1948). This penetration efficiency takes also the BD into account because its operation principle is also based on diffusional losses; thus, $L$ denotes the effective length of the combined effect of the sampling lines and the BD.

Systematic errors for the calculated size distributions after the ED include the uncertainty of the cut-diameters and the uncertainty of the value of $L/Q$. Because the detection efficiency curves of the PSM and CPC 3775 are measured using particles having a different composition than $H_2SO_4$-$H_2O$, as in these measurements, and because environmental parameters, such as temperature, can have effects on the detection efficiency curves, the reported cut-diameters may not hold exactly. The uncertainty of 20 % is estimated for the cut-diameters and also for the ratio of the cut-diameters $D_{\text{p},i}^{\text{larger}}/D_{\text{p},i}^{\text{smaller}}$ because it is expected that if one of the cut-diameters is deviated towards smaller or larger particle sizes, another one is deviated towards the same direction. 10 % uncertainty is estimated for the value of $L/Q$, which includes the measurement uncertainty for both $L$ and $Q$ and the uncertainty in the equations of Gormley and Kennedy (1948).

The standard deviations of the measured concentrations caused by the instability of the particle generation are in the range of 1 ... 25 %, depending on the concentration level and particle sizes: higher concentrations and larger particle sizes provided more stable particle generation compared to lower concentrations and smaller particle sizes.

The error bars for $\nu(D_{\text{p},i})$ can be calculated with the equation

$$\frac{\Delta\nu}{\nu} = \sqrt{\left(\frac{\partial\nu}{\partial N_i^{\text{smaller}}}\frac{\Delta N_i^{\text{smaller}}}{\nu}\right)^2 + \left(\frac{\partial\nu}{\partial N_i^{\text{larger}}}\frac{\Delta N_i^{\text{larger}}}{\nu}\right)^2 + \left(\frac{\partial\nu}{\partial\log(D_{\text{p},i}^{\text{larger}}/D_{\text{p},i}^{\text{smaller}})}\frac{\Delta\log(D_{\text{p},i}^{\text{larger}}/D_{\text{p},i}^{\text{smaller}})}{\nu}\right)^2 + \left(\frac{\partial\nu}{\partial p}\frac{\Delta p}{\nu}\right)^2}$$

$$= \sqrt{\left(\frac{\Delta N_i^{\text{smaller}}}{N_i^{\text{smaller}} - N_i^{\text{larger}}}\right)^2 + \left(\frac{\Delta N_i^{\text{larger}}}{N_i^{\text{smaller}} - N_i^{\text{larger}}}\right)^2 + \left(\frac{\Delta\log(D_{\text{p},i}^{\text{larger}}/D_{\text{p},i}^{\text{smaller}})}{\log(D_{\text{p},i}^{\text{larger}}/D_{\text{p},i}^{\text{smaller}})}\right)^2 + \left(\frac{\Delta p}{p}\right)^2} \tag{S3}$$

where $\Delta N_i^{\text{smaller}}$ and $\Delta N_i^{\text{smaller}}$ are the standard deviations of the concentrations, depending on the measurement case, the third term is $0.2^2$ because $\Delta(D_{\text{p},i}^{\text{larger}}/D_{\text{p},i}^{\text{smaller}})/(D_{\text{p},i}^{\text{larger}}/D_{\text{p},i}^{\text{smaller}}) = 20\,\%$, and $\Delta p$ depends on the particle size and is calculated with the equation

$$\frac{\Delta p}{p} = \sqrt{\left(\frac{\partial p}{\partial(L/Q)}\frac{\Delta(L/Q)}{p}\right)^2 + \left(\frac{\partial p}{\partial D_{\text{p},i}}\frac{\Delta D_{\text{p},i}}{p}\right)^2} \tag{S4}$$

using $\Delta(L/Q)/(L/Q) = 10\,\%$ and $\Delta D_{\text{p},i}/D_{\text{p},i} = 20\,\%$.

**2.2 Calculating the error bars for the Nano-SMPS**

The particle number size distributions reported by the Nano-SMPS device have already went through the manufacturer's inversion algorithm. Thus, the inverse modeling of this work does not try to predict the concentration measured as a function of time measured by the CPC 3776, acting as a particle counter in the Nano-SMPS system. Instead, the inverse modeling takes only the diffusional losses in the sampling lines and the CPC 3776 detection efficiency curve into account, but not, e.g., the radioactive charger efficiency and the diffusional losses inside the device. It is partly unknown what is included in the manufacturer's inversion algorithm, but at least the charger efficiency and the diffusional losses inside the device are included. The inversion algorithm probably includes also the CPC 3776 detection efficiency curve, $f_{\text{CPC}}$, but it is, however, included in the inverse modeling of this work because it seems that it may differ significantly from the curve reported by the manufacturer, according to Hermann et al. (2007) and Mordas et al. (2008), as presented in Fig. S2. Unfortunately, the curve for the device used in these measurements is not measured; therefore, the inversion modeling uses the one reported by

Mordas et al. (2008) because it lies between the other two curves, representing an average one. The uncertainty of the detection efficiency at a specific diameter is calculated from the maximum range of variation of the detection efficiencies from these three different sources.

[Figure]

**Figure S2.** The CPC 3776 detection efficiency curves as a function of particle size reported by the manufacturer, Hermann et al. (2007) , and Mordas et al. (2008) . Additionally, a hypothetical curve correcting the disagreement between sub-6 nm particle size distributions measured by the Nano-SMPS and PSM+CPC system is presented.

The hypothetical detection efficiency curve presented in Fig. S2 is based on the curve reported by Mordas et al. (2008) but with different parameters. If this hypothetical curve is the actual curve of the device used, the size distributions as in Fig. 4 will be corrected to the distributions presented in Fig. S3, from which it can be seen that the size distributions measured by different devices correspond clearly better, at least for the two cases having the lowest $T_{sa}$. The PSM+CPC distribution for the case having the highest $T_{sa}$ is probably overestimated because, according to Fig. 10 (c), the concentrations measured with different cut-diameters are on the same level, implying that there is not a notable amount of particles in that size range.

Other systematical errors, in addition to the uncertainty involved in the CPC 3776 detection efficiency curve, include the uncertainties of the charger efficiency, the diffusional losses, and the particle sizes interpreted by the manufacturer's inversion algorithm. The charger of the Nano-SMPS used was a TSI 3077 radioactive Kr-85 charger, which is based on charging particles bipolarly to the charge equilibrium state. The inversion algorithm uses the positive charge distribution function, $f_{charger}$, reported by Wiedensohler (1988) . It is a semi-empirical function in which the mobilities and masses of positive and negative ions in the carrier gas are fitted based on the charge distribution measurements (Hussin et al., 1983; Adachi et al., 1985; Wiedensohler et al., 1986) made for particles larger than 5 nm in diameters. Alonso et al. (1997) have measured the charge distributions down to particle diameters of 2.5 nm. Unfortunately, the charger distributions from all these measurements differ, especially for the smallest particle sizes, and have thus different ion parameters, due to different particle compositions, carrier gas compositions, and the accuracies of the particle size measurements. Therefore, the charger efficiency function selected in the manufacturer's inversion algorithm is not exact. Based on the differences between the results of these charge distribution measurements, the uncertainties of 30, 20, and 10 % for the charger efficiencies at particle diameters of 6, 10, and 20 nm, respectively, are estimated. Another factor causing uncertainty for the charger efficiency is how satisfactorily the charge distribution is developed to the equilibrium state. If the residence time inside the charger is too short (Alonso et al., 1997) , the activity of the charger is too low (de La Verpilliere et al., 2015) (e.g, if the activity of the charger is depleted due to a long operating life), or if the particle concentration is too high compared to the ion concentration (Wiedensohler et al., 2012) , the equilibrium state may not be reached and the charger efficiency is overestimated. According to the deviations of the charge distributions in the measurements of de La Verpilliere et al. (2015) from the charge distribution function of Wiedensohler (1988) , the uncertainties of 40, 30, and 20 % for the charger efficiencies at particle diameters of

[Figure]

**Figure S3.** The corrected particle size distributions as in Fig. 4 if the detection efficiency curve of the CPC 3776 would be the hypothetical curve presented in Fig. S2.

2, 6, and 10 nm, respectively, are estimated. Because the particles before the Nano-SMPS are supposedly uncharged in this work, the possible incomplete reaching of the charge equilibrium state causes that the particles are less charged than predicted. Therefore, the concentrations would be underestimated, and thus the error bars related to this are considered only towards the positive direction.

The diffusional losses of the particles in the sampling lines of this work are based on the equations of Gormley and Kennedy (1948) using the $L_\mathrm{lines}/Q_\mathrm{lines}$ parameter as in the case of the PSM+CPC system (the uncertainty of 10 % for the $L_\mathrm{lines}/Q_\mathrm{lines}$ parameter in this case is again estimated). The correction of the diffusional losses inside the Nano-SMPS device is also based on those equations in the manufacturer's inversion algorithm. The algorithm uses an empirically fitted $L_\mathrm{device}/Q_\mathrm{device}$ value which included the whole route of the particles inside the device even though the route is not a perfect laminar circular tube flow, for which the analytical solution by Gormley and Kennedy (1948) is based on. Therefore, the penetration function for particles inside the device, $p_\mathrm{device}$, may not be exact and the uncertainty of 10 % for the $L_\mathrm{device}/Q_\mathrm{device}$ parameter is estimated.

The correction factor assumed to exist in the inversion algorithm of the Nano-SMPS, to which the penetration in the sampling lines, $p_\mathrm{lines}$, is added, is

$$C(D_{\mathrm{p},i}) = \frac{1}{f_\mathrm{charger}(D_{\mathrm{p},i}) \cdot f_\mathrm{CPC}(D_{\mathrm{p},i}) \cdot p_\mathrm{device}(D_{\mathrm{p},i}, L_\mathrm{device}/Q_\mathrm{device}) \cdot p_\mathrm{lines}(D_{\mathrm{p},i}, L_\mathrm{lines}/Q_\mathrm{lines})}. \tag{S5}$$

The concentration measured with a specific DMA (Differential Mobility Analyzer) voltage at a specific time, related to the particle diameter of $D_{\mathrm{p},i}$ (obtained though the inversion algorithm), is multiplied with $C(D_{\mathrm{p},i})$ in order to obtain the size

distribution in a location of $L_{\text{lines}}$ before the device, i.e., after the ED in this case. For very small particles, all the four functions in Eq. (S5) have very low value; and thus, the value of $C(D_{\text{p},i})$ is extremely high. This is illustrated in Fig. S4 from which it can be observed that the value for sub-6 nm particles is several orders of magnitude. Very high correction factor denotes very low number of particle counts detected by the CPC at a specific diameter, and very low counts do not provide good accuracy due to statistics: there may be only a few randomly detected single particles or there may be even not a single detection at all during the time dedicated to that particle size, even though multiple scans have been performed for one measurement case. In the case of no or very low detection of single particles, the error bars cannot be calculated. Because the correction factor increases very steeply with decreasing particle size, the uncertainties involved in the functions in Eq. (S5) can deviate it in high extent. Another consequence of the steep behavior of the correction factor is that if there is even a minor error in the interpreted particle diameters, the value of the correction factor can be significantly misestimated. There are several factors that can cause error to the particle diameters measured by the Nano-SMPS (Wiedensohler et al., 2012); here, the uncertainty of 5 % is estimated for the diameters.

[Figure]

**Figure S4.** Nano-SMPS correction factor, as in Eq. (S5), used to correct the measured particle concentrations in the data inversion.

The error bars for the size distributions, $\nu(D_{\text{p},i})$, can be calculated with the equation

$$\frac{\Delta\nu}{\nu} = \sqrt{\left(\frac{\Delta\nu'}{\nu'}\right)^2 + \left(\frac{\Delta C}{C}\right)^2} \tag{S6}$$

where $\Delta\nu'$ is the standard deviation of the size distributions at the particle diameter of $D_{\text{p},i}$ output by the device, $\nu'$, and $\Delta C$ is the uncertainty of the Nano-SMPS correction factor. $\Delta\nu'$ represents the standard deviation caused by the instability in the particle generation, as in the case of the PSM+CPC system, but also by the low counting statistics for the particles sizes having a very low overall detection efficiency and for the particle sizes having low concentration in the measured case. $\Delta C$ is

calculated by

$$\frac{\Delta C}{C} = \sqrt{\left(\frac{\partial C}{\partial f_{charger}}\frac{\Delta f_{charger}}{C}\right)^2 + \left(\frac{\partial C}{\partial f_{CPC}}\frac{\Delta f_{CPC}}{C}\right)^2 + \left(\frac{\partial C}{\partial p_{device}}\frac{\Delta p_{device}}{C}\right)^2 + \left(\frac{\partial C}{\partial p_{lines}}\frac{\Delta p_{lines}}{C}\right)^2 + \left(\frac{\partial C}{\partial D_{p,i}}\frac{\Delta D_{p,i}}{C}\right)^2}$$

$$= \sqrt{\left(\frac{\Delta f_{charger}}{f_{charger}}\right)^2 + \left(\frac{\Delta f_{CPC}}{f_{CPC}}\right)^2 + \left(\frac{\Delta p_{device}}{p_{device}}\right)^2 + \left(\frac{\Delta p_{lines}}{p_{lines}}\right)^2 + \left(\frac{\partial C}{\partial D_{p,i}}\frac{\Delta D_{p,i}}{C}\right)^2} \qquad (S7)$$

where $\Delta f_{charger}/f_{charger}$ and $\Delta f_{CPC}/f_{CPC}$ are the relative uncertainties for $f_{charger}$ and $f_{CPC}$ having the values mentioned before, $\Delta p_{device}$ and $\Delta p_{lines}$ are the uncertainties for the penetration efficiencies, $p_{device}$ and $p_{lines}$, and the last term represents the uncertainty of $C$ caused by the uncertainty of particle diameters. $\Delta p_{device}$ and $\Delta p_{lines}$ depend on the particle diameter and are calculated by

$$\Delta p_{device} = \frac{\partial p_{device}}{\partial (L_{device}/Q_{device})} \cdot \Delta (L_{device}/Q_{device})$$

$$\Delta p_{lines} = \frac{\partial p_{lines}}{\partial (L_{lines}/Q_{lines})} \cdot \Delta (L_{lines}/Q_{lines}) \qquad (S8)$$

which differ from Eg. (S4) by missing the effect of the particle diameter because that effect is included in the last term of Eq. (S7). The last term represent the total effect of the particle diameter on $C$ because the particle diameter is involved in all other four terms.

**2.3 Calculated error bars for the size distributions**

The relative uncertainties for the size distributions between 2 and 10 nm after the ED caused by the different uncertainties involved in the size distribution measurements are presented in Tab. S1. For the PSM+CPC system, the most significant systematic errors arise from the uncertainty in the cut-diameters (12 … 30 %), partly due to correcting the diffusional losses in sampling lines needed in backwards-correcting the measured distributions to represent the distributions after the ED. For the Nano-SMPS, the uncertainty of the charger efficiency plays a major role in the systematic errors (40 … 66 %), but the uncertainty of the CPC 3776 detection efficiency curve has also a significant role for the smallest particles (55 % for 3.7 nm). The error bars decrease steeply when measuring particles sized 10 nm or larger using the Nano-SMPS. Both devices are, in theory, capable in measuring the size distribution at 3.7 nm, but the error bars for the Nano-SMPS are clearly higher compared to the PSM+CPC system. Therefore, the PSM+CPC system suits better in measuring near that diameter.

Random errors caused by the instability of the particle generation and low counting statistics of the Nano-SMPS also have significant effects for the both devices if there is not a notable amount of concentration in a specific size range. In the case with $D_{\bar{m}} = 3.6\,nm$, there is a notable amount of concentration in the PSM+CPC size range, and thus, the error bars are relatively low (22 … 61 %) for the PSM+CPC system. For the Nano-SMPS, the standard deviation for 6 nm particles is 72 %, which is, however, the most accurately measured particle size in that case: larger particles are inexistent and smaller particles are not detected; thus, low counting statistics cause high errors. In the case with $D_{\bar{m}} = 19\,nm$, the Nano-SMPS suits well in measuring at the particle size of 10 nm (the standard deviation of 10 % originating mainly from the instability in the particle generation) and also relatively well at the particle size of 6 nm (the standard deviation of 24 %), but the errors increase with the particles smaller than 6 nm. Conversely, the PSM+CPC system has high error bars because the concentration in the PSM+CPC size range is so low that the difference between the concentrations measured with different cut-diameters are smaller than the standard deviation of the concentrations (see Fig. 10), which is always a problem with the PSM having a cumulative nature in measuring concentrations, if the cut-diameters of the adjacent saturator flow rates are too near or the measured signal is too instable. This issue can be overcome by skipping the data measured with the adjacent cut-diameters or even by considering only the data measured with the smallest and with the largest cut-diameter. However, while the error bars will be narrower in this alternative method, the information on the shape of the size distribution in that size range will diminish.

**Table S1.** The percentual uncertainties for the size distributions after the ED, $\Delta\nu/\nu$ (%), for the selected particle diameters. The first seven lines represent the systematical error of the devices and they are independent of the measurement case. The last two lines represent the effect of the standard deviation for two measurement cases having small and large particles.

| Device $D_\mathrm{p}$ | PSM+CPC 2 nm | PSM+CPC 3.7 nm | Nano-SMPS 3.7 nm | Nano-SMPS 6 nm | Nano-SMPS 10 nm |
|---|---|---|---|---|---|
| Diffusional losses in sampling lines $\left(\frac{\Delta(L_\mathrm{lines}/Q_\mathrm{lines})}{L_\mathrm{lines}/Q_\mathrm{lines}}=10\,\%\right)$ | 8 | 3 | 4 | 2 | 1 |
| Diffusional losses in sampling lines $\left(\frac{\Delta D_{\mathrm{p},i}}{D_{\mathrm{p},i}}=20\,\%\right)$ | 30 | 12 | | | |
| PSM detection efficiency $\left(\frac{\Delta(D_{\mathrm{p},i}^{\mathrm{larger}}/D_{\mathrm{p},i}^{\mathrm{smaller}})}{D_{\mathrm{p},i}^{\mathrm{larger}}/D_{\mathrm{p},i}^{\mathrm{smaller}}}=20\,\%\right)$ | 20 | 20 | | | |
| Kr-85 charger efficiency | | | 66 | 61 | 40 |
| CPC 3776 detection efficiency | | | 55 | 8 | 0.7 |
| Diffusional losses inside the DMA $\left(\frac{\Delta(L_\mathrm{lines}/Q_\mathrm{lines})}{L_\mathrm{lines}/Q_\mathrm{lines}}=10\,\%\right)$ | | | 16 | 7 | 3 |
| Nano-SMPS correction factor $\left(\frac{\Delta D_{\mathrm{p},i}}{D_{\mathrm{p},i}}=5\,\%\right)$ | | | 32 | 17 | 10 |
| Random error in $D_{\bar{m}}=3.6\,\mathrm{nm}$ case (102 °C) | 61 | 22 | –[a] | 72 | –[a] |
| Random error in $D_{\bar{m}}=19\,\mathrm{nm}$ case (157.2 °C) | $\infty$[b] | 250 | –[a] | 24 | 10 |

[a] Cannot be calculated due to insufficient particle counts.
[b] For this point, $\nu(2\,\mathrm{nm})=0$ but $\Delta\nu$ is a non-zero number due to standard deviation.

The error bars for the size distributions shown in Fig. 4 are presented in Fig. S5. By considering the error bars, the distributions from the both devices agree for the cases in panes (a) and (c), whereas the case in pane (b) has still some disagreement implying that other error sources than accounted here can be involved in the measurements using these devices. According to the error bars near the particle size of 4 nm connecting the two size distributions, the PSM+CPC system provides more reliable results in the cases in panes (a) and (b). Conversely, in the case in pane (c), the Nano-SMPS provides more reliable results because, although there are two points in the PSM+CPC distribution, the alternative method shows no particles at all. However, the error bars for the alternative method are high; thus, the probability of the existence of particles in the PSM+CPC size range is high. Nevertheless, the fraction of particles in that size range compared to the total particle count is, definitely, some orders of magnitude smaller than in the cases in panes (a) and (b). Figure S6 presents the error bars for the distributions shown in Fig. 11.

In conclusion, the suitability of the Nano-SMPS for the particle sizes smaller than ~10 nm is weak especially due to the uncertainties involved in the radioactive charger efficiency and the CPC 3776 detection efficiency. Wiedensohler et al. (2012) have performed an intercomparison of several mobility particle sizers, in which the different devices provided a good agreement for the particle sizes larger than ~15 nm but had significant disagreements for the smaller particle sizes, without explanation. Due to the incapability of the Nano-SMPS in determining the size distribution reliably in sub-10 nm diameter range, in all of the cases studied here and elsewhere, the PSM+CPC system suits clearly better in determining the size distribution in that particle size range, or at least the total number concentration of particles larger than ~1 nm.

[Figure]

**Figure S5.** The measured size distributions for the measurement cases having the $T_{sa}$ of (a) 102 °C, (b) 135.5 °C, and (c) 157.2 °C, as shown in Fig. 4, with the error bars. The alternative PSM+CPC distributions represent the distributions using only the concentrations measured with the smallest and the largest cut-diameters, in order to narrow the error bars. The green shaded areas denote the error bars for the distributions from the alternative method. The error bars for the particle sizes from the Nano-SMPS ($\pm 5$ %) are not shown for clarity.

[Figure]

**Figure S6.** The simulated and measured size distributions for the measurement cases having the $T_{sa}$ of (a) 102 °C, (b) 135.5 °C, and (c) 157.2 °C, as shown in Fig. 11, with the error bars. The error bars for the particle sizes from the Nano-SMPS (±5 %) are not shown for clarity.

**2.4 Calculating the error bars for the diameters with the average mass**

The diameter with the average mass of a distribution is calculated by

$$D_{\bar{m}} = \left( \frac{M'}{N} \right)^{\frac{1}{3}} = \left( \frac{\sum_i^{\mathrm{PC}} \nu(D_{\mathrm{p},i}) \cdot \mathrm{d}\log D_{\mathrm{p},i} \cdot D_{\mathrm{p},i}^3 + \sum_i^{\mathrm{NS}} \nu(D_{\mathrm{p},i}) \cdot \mathrm{d}\log D_{\mathrm{p},i} \cdot D_{\mathrm{p},i}^3}{\sum_i \nu(D_{\mathrm{p},i}) \cdot \mathrm{d}\log D_{\mathrm{p},i}} \right)^{\frac{1}{3}} = \left( \frac{M'_{\mathrm{PC}} + M'_{\mathrm{NS}}}{N} \right)^{\frac{1}{3}} \tag{S9}$$

where $M'$ is the third moment of the distribution, $M'_{\mathrm{PC}}$ and $M'_{\mathrm{NS}}$ are the parts of the third moment from the PSM+CPC data and from the Nano-SMPS data, respectively, and $N$ is the total number concentration.

The error bars for $D_{\bar{m}}$ can be calculated with the equation

$$\frac{\Delta D_{\bar{m}}}{D_{\bar{m}}} = \sqrt{ \left( \left. \frac{\Delta D_{\bar{m}}}{D_{\bar{m}}} \right|_{\Delta D_{\mathrm{p}}} \right)^2 + \left( \left. \frac{\Delta D_{\bar{m}}}{D_{\bar{m}}} \right|_{\Delta \nu_{\mathrm{s}}} \right)^2 + \left( \left. \frac{\Delta D_{\bar{m}}}{D_{\bar{m}}} \right|_{\Delta \nu_{\mathrm{r}}} \right)^2 } \tag{S10}$$

where the first term represents the uncertainty caused by the uncertainty of the interpreted particle diameters, the second term the uncertainty caused by the uncertainty of the number size distribution due to the systematic errors of the devices, and the last term the uncertainty caused by the uncertainty of the number size distribution due to the random error.

The first term in Eq. (S10) is separated to the effects of the PSM+CPC system and of the Nano-SMPS, respectively:

$$\left. \frac{\Delta D_{\bar{m}}}{D_{\bar{m}}} \right|_{\Delta D_{\mathrm{p}}} = \sqrt{ \left( \left. \frac{\Delta D_{\bar{m}}}{D_{\bar{m}}} \right|_{\Delta D_{\mathrm{p},\mathrm{PC}_1}} \right)^2 + \left( \left. \frac{\Delta D_{\bar{m}}}{D_{\bar{m}}} \right|_{\Delta D_{\mathrm{p},\mathrm{NS}_1}} \right)^2 } . \tag{S11}$$

Because particle diameters are dependent variables for a specific device, i.e., if one diameter is shifted to a direction, other diameters are most probably shifted to the same direction and with almost the same magnitude, the diameters in Eq. (S9) are separated to dependent and independent parts:

$$D_{\mathrm{p},i} = D_{\mathrm{p},\mathrm{PC}_1} \cdot D'_{\mathrm{p},i}$$

$$D_{\mathrm{p},i} = D_{\mathrm{p},\mathrm{NS}_1} \cdot D'_{\mathrm{p},i}. \tag{S12}$$

where $D_{\mathrm{p},\mathrm{PC}_1}$ and $D_{\mathrm{p},\mathrm{NS}_1}$ denote the smallest diameters measured by the PSM+CPC system and by the Nano-SMPS, respectively, and $D'_{\mathrm{p},i}$ is a dimensionless variable denoting the ratios of all other diameters to the smallest diameter. Hence, the third moments can be expressed as

$$M'_{\mathrm{PC}} = \sum_i^{\mathrm{PC}} \nu(D_{\mathrm{p},i}) \cdot \mathrm{d}\log D_{\mathrm{p},i} \cdot D_{\mathrm{p},i}^3 = \sum_i^{\mathrm{PC}} \nu(D_{\mathrm{p},i}) \cdot \mathrm{d}\log D_{\mathrm{p},i} \cdot (D_{\mathrm{p},\mathrm{PC}_1} \cdot D'_{\mathrm{p},i})^3 = D_{\mathrm{p},\mathrm{PC}_1}^3 \cdot \sum_i^{\mathrm{PC}} \nu(D_{\mathrm{p},i}) \cdot \mathrm{d}\log D_{\mathrm{p},i} \cdot D'^3_{\mathrm{p},i}$$

$$M'_{\mathrm{NS}} = \sum_i^{\mathrm{NS}} \nu(D_{\mathrm{p},i}) \cdot \mathrm{d}\log D_{\mathrm{p},i} \cdot D_{\mathrm{p},i}^3 = \sum_i^{\mathrm{NS}} \nu(D_{\mathrm{p},i}) \cdot \mathrm{d}\log D_{\mathrm{p},i} \cdot (D_{\mathrm{p},\mathrm{NS}_1} \cdot D'_{\mathrm{p},i})^3 = D_{\mathrm{p},\mathrm{NS}_1}^3 \cdot \sum_i^{\mathrm{NS}} \nu(D_{\mathrm{p},i}) \cdot \mathrm{d}\log D_{\mathrm{p},i} \cdot D'^3_{\mathrm{p},i}. \tag{S13}$$

The uncertainties in Eq. (S11) can now be calculated by

$$\frac{\Delta D_{\bar{m}}}{D_{\bar{m}}}\bigg|_{\Delta D_{\mathrm{p,PC_1}}} = \frac{\partial D_{\bar{m}}}{\partial D_{\mathrm{p,PC_1}}} \cdot \frac{\Delta D_{\mathrm{p,PC_1}}}{D_{\bar{m}}} = \frac{1}{3}\frac{D_{\bar{m}}}{M'} \cdot \frac{\partial M'_{\mathrm{PC}}}{\partial D_{\mathrm{p,PC_1}}} \cdot \frac{\Delta D_{\mathrm{p,PC_1}}}{D_{\bar{m}}} = \frac{M'_{\mathrm{PC}}}{M'} \cdot \frac{\Delta D_{\mathrm{p,PC_1}}}{D_{\mathrm{p,PC_1}}}$$

$$\frac{\Delta D_{\bar{m}}}{D_{\bar{m}}}\bigg|_{\Delta D_{\mathrm{p,NS_1}}} = \frac{\partial D_{\bar{m}}}{\partial D_{\mathrm{p,NS_1}}} \cdot \frac{\Delta D_{\mathrm{p,NS_1}}}{D_{\bar{m}}} = \frac{1}{3}\frac{D_{\bar{m}}}{M'} \cdot \frac{\partial M'_{\mathrm{NS}}}{\partial D_{\mathrm{p,NS_1}}} \cdot \frac{\Delta D_{\mathrm{p,NS_1}}}{D_{\bar{m}}} = \frac{M'_{\mathrm{NS}}}{M'} \cdot \frac{\Delta D_{\mathrm{p,NS_1}}}{D_{\mathrm{p,NS_1}}}. \tag{S14}$$

As 20 and 5 % uncertainties for the diameters for the PSM+CPC system and for the Nano-SMPS, respectively, were estimated, Eq. (S11) becomes

$$\frac{\Delta D_{\bar{m}}}{D_{\bar{m}}}\bigg|_{\Delta D_{\mathrm{p}}} = \frac{1}{M'}\sqrt{(M'_{\mathrm{PC}} \cdot 0.2)^2 + (M'_{\mathrm{NS}} \cdot 0.05)^2}. \tag{S15}$$

The last two terms in Eq. (S10) related to the systematic and random errors separate the errors because the systematic errors for all size bins are presumably to the same direction and in almost the same magnitude and the random errors are randomly directed between different size bins because they are measured at different times. The number size distributions can be separated to the parts involving the sources for the systematic ($\Delta \nu_{\mathrm{s}}$) and for the random ($\Delta \nu_{\mathrm{r}}$) errors, respectively, using $\nu = \nu_{\mathrm{s}} \cdot \nu_{\mathrm{r}}$. The systematic errors for $\nu_{\mathrm{s}}$ are independent variables for the different devices, but dependent variables for the different size bins of a specific device. Hence, the second term in Eq. (S10) is further separated to the PSM+CPC system and to the Nano-SMPS, respectively, using

$$\frac{\Delta D_{\bar{m}}}{D_{\bar{m}}}\bigg|_{\Delta \nu_{\mathrm{s}}} = \sqrt{\left(\frac{\Delta D_{\bar{m}}}{D_{\bar{m}}}\bigg|_{\Delta \nu_{\mathrm{s,PC}}}\right)^2 + \left(\frac{\Delta D_{\bar{m}}}{D_{\bar{m}}}\bigg|_{\Delta \nu_{\mathrm{s,NS}}}\right)^2}$$

$$= \sqrt{\left[\frac{D_{\bar{m}}(\nu + \Delta \nu_{\mathrm{s,PC}}) - D_{\bar{m}}(\nu - \Delta \nu_{\mathrm{s,PC}})}{2D_{\bar{m}}}\right]^2 + \left[\frac{D_{\bar{m}}(\nu + \Delta \nu_{\mathrm{s,NS}}) - D_{\bar{m}}(\nu - \Delta \nu_{\mathrm{s,NS}})}{2D_{\bar{m}}}\right]^2}. \tag{S16}$$

The last term in Eq. (S10), related to the random error, is calculated by

$$\frac{\Delta D_{\bar{m}}}{D_{\bar{m}}}\bigg|_{\Delta \nu_{\mathrm{r}}} = \sqrt{\sum_i \left[\frac{\partial D_{\bar{m}}}{\partial \nu_{\mathrm{r}}(D_{\mathrm{p},i})} \cdot \frac{\Delta \nu_{\mathrm{r}}(D_{\mathrm{p},i})}{D_{\bar{m}}}\right]^2} = \sqrt{\sum_i \left[\left(\frac{1}{3M'}\frac{\partial M'}{\partial \nu_{\mathrm{r}}(D_{\mathrm{p},i})} - \frac{1}{3N}\frac{\partial N}{\partial \nu_{\mathrm{r}}(D_{\mathrm{p},i})}\right)\Delta \nu_{\mathrm{r}}(D_{\mathrm{p},i})\right]^2}$$

$$= \frac{1}{3}\sqrt{\sum_i \left[\left(\frac{\mathrm{d}\log D_{\mathrm{p},i} \cdot D_{\mathrm{p},i}^3}{M'} - \frac{\mathrm{d}\log D_{\mathrm{p},i}}{N}\right)\Delta \nu_{\mathrm{r}}(D_{\mathrm{p},i})\right]^2}. \tag{S17}$$

The calculated error bars for the diameters with the average mass are presented in Fig. 13.

[revised manuscript text omitted]

Nevertheless, both the measured data sets agree well with the shape of the theoretical curve, which implies that $[\text{H}_2\text{SO}_4]_\text{raw}$ can be estimated using $T_\text{sa}$. However, the absolute value for $[\text{H}_2\text{SO}_4]_\text{raw}$ cannot be satisfactorily estimated using neither $T_\text{sa}$ nor the measured concentrations due to the discrepancy of the measured concentrations. Therefore, the simulations of this study did not use the measured concentrations as the boundary conditions; instead, the $[\text{H}_2\text{SO}_4]_\text{raw}$ values were obtained through inverse modeling.

[Figure]

**Figure S1.** Simulated sulfuric acid concentrations in the raw sample compared to the measured and the theoretical concentrations with different sulfuric acid evaporator temperatures.

**2 Error estimation for particle size distribution measurements**

The disagreement of sub-6 nm particle size distributions measured by the combination of the PSM and the CPC 3775 and by the Nano-SMPS is examined by estimating the sources and the magnitudes of errors in measured concentrations for these devices.

**2.1 Calculating the error bars for the combination of the PSM and the CPC 3775**

The particle number size distributions are calculated using the step-wise method according to Lehtipalo et al. (2014). Backwards-correcting the measured distributions to represent the distributions after the ED requires multiplying the data with the DR of the BD and dividing by the penetration efficiency of particles in the sampling lines between the ED and the measurement devices. Finally, the equation to obtain the distribution at particle size $D_{\text{p},i}$ after the ED from the measured concentrations is

$$\nu(D_{\text{p},i}) = \frac{N_i^\text{smaller} - N_i^\text{larger}}{\log(D_{\text{p},i}^\text{larger}/D_{\text{p},i}^\text{smaller}) \cdot p(D_{\text{p},i}, L/Q)} \tag{S2}$$

where $N_i^\text{smaller}$ and $N_i^\text{larger}$ are particle number concentrations measured with the $D_{50}$-cut-sizes of $D_{\text{p},i}^\text{smaller}$ and $D_{\text{p},i}^\text{larger}$, respectively. $D_{\text{p},i}$ is the geometric mean diameter of the $D_{50}$-cut-sizes and $p(D_{\text{p},i}, L/Q)$ is the penetration efficiency of particles with

a diameter of $D_{\mathrm{p},i}$ in a sampling line with a length of $L_{\mathrm{lines}} = L$ and flow rate of $Q_{\mathrm{lines}} = Q$, according to diffusional losses calculated with the equations of Gormley and Kennedy (1948). This penetration efficiency takes also the BD into account because its operation principle is also based on diffusional losses; thus, $L$ denotes the effective length of the combined effect of the sampling lines and the BD.

Systematic errors for the calculated size distributions after the ED include the uncertainty of the cut-diameters and the uncertainty of the value of $L/Q$. Because the detection efficiency curves of the PSM and CPC 3775 are measured using particles having a different composition than $H_2SO_4$-$H_2O$, as in these measurements, and because environmental parameters, such as temperature, can have effects on the detection efficiency curves, the reported cut-diameters may not hold exactly. The uncertainty of 20 % is estimated for the cut-diameters and also for the ratio of the cut-diameters $D_{\mathrm{p},i}^{\mathrm{larger}}/D_{\mathrm{p},i}^{\mathrm{smaller}}$ because it is expected that if one of the cut-diameters is deviated towards smaller or larger particle sizes, another one is deviated towards the same direction. 10 % uncertainty is estimated for the value of $L/Q$, which includes the measurement uncertainty for both $L$ and $Q$ and the uncertainty in the equations of Gormley and Kennedy (1948).

The standard deviations of the measured concentrations caused by the instability of the particle generation are in the range of 1 ... 25 %, depending on the concentration level and particle sizes: higher concentrations and larger particle sizes provided more stable particle generation compared to lower concentrations and smaller particle sizes.

The error bars for $\nu(D_{\mathrm{p},i})$ can be calculated with the equation

$$
\frac{\Delta\nu}{\nu} = \sqrt{\left(\frac{\partial\nu}{\partial N_i^{\mathrm{smaller}}}\frac{\Delta N_i^{\mathrm{smaller}}}{\nu}\right)^2 + \left(\frac{\partial\nu}{\partial N_i^{\mathrm{larger}}}\frac{\Delta N_i^{\mathrm{larger}}}{\nu}\right)^2 + \left(\frac{\partial\nu}{\partial\log(D_{\mathrm{p},i}^{\mathrm{larger}}/D_{\mathrm{p},i}^{\mathrm{smaller}})}\frac{\Delta\log(D_{\mathrm{p},i}^{\mathrm{larger}}/D_{\mathrm{p},i}^{\mathrm{smaller}})}{\nu}\right)^2 + \left(\frac{\partial\nu}{\partial p}\frac{\Delta p}{\nu}\right)^2}
$$

$$
= \sqrt{\left(\frac{\Delta N_i^{\mathrm{smaller}}}{N_i^{\mathrm{smaller}} - N_i^{\mathrm{larger}}}\right)^2 + \left(\frac{\Delta N_i^{\mathrm{larger}}}{N_i^{\mathrm{smaller}} - N_i^{\mathrm{larger}}}\right)^2 + \left(\frac{\Delta\log(D_{\mathrm{p},i}^{\mathrm{larger}}/D_{\mathrm{p},i}^{\mathrm{smaller}})}{\log(D_{\mathrm{p},i}^{\mathrm{larger}}/D_{\mathrm{p},i}^{\mathrm{smaller}})}\right)^2 + \left(\frac{\Delta p}{p}\right)^2} \tag{S3}
$$

where $\Delta N_i^{\mathrm{smaller}}$ and $\Delta N_i^{\mathrm{smaller}}$ are the standard deviations of the concentrations, depending on the measurement case, the third term is $0.2^2$ because $\Delta(D_{\mathrm{p},i}^{\mathrm{larger}}/D_{\mathrm{p},i}^{\mathrm{smaller}})/(D_{\mathrm{p},i}^{\mathrm{larger}}/D_{\mathrm{p},i}^{\mathrm{smaller}}) = 20\,\%$, and $\Delta p$ depends on the particle size and is calculated with the equation

$$
\frac{\Delta p}{p} = \sqrt{\left(\frac{\partial p}{\partial(L/Q)}\frac{\Delta(L/Q)}{p}\right)^2 + \left(\frac{\partial p}{\partial D_{\mathrm{p},i}}\frac{\Delta D_{\mathrm{p},i}}{p}\right)^2} \tag{S4}
$$

using $\Delta(L/Q)/(L/Q) = 10\,\%$ and $\Delta D_{\mathrm{p},i}/D_{\mathrm{p},i} = 20\,\%$.

**2.2 Calculating the error bars for the Nano-SMPS**

The particle number size distributions reported by the Nano-SMPS device have already went through the manufacturer's inversion algorithm. Thus, the inverse modeling of this work does not try to predict the concentration measured as a function of time measured by the CPC 3776, acting as a particle counter in the Nano-SMPS system. Instead, the inverse modeling takes only the diffusional losses in the sampling lines and the CPC 3776 detection efficiency curve into account, but not, e.g., the radioactive charger efficiency and the diffusional losses inside the device. It is partly unknown what is included in the manufacturer's inversion algorithm, but at least the charger efficiency and the diffusional losses inside the device are included. The inversion algorithm probably includes also the CPC 3776 detection efficiency curve, $f_{\mathrm{CPC}}$, but it is, however, included in the inverse modeling of this work because it seems that it may differ significantly from the curve reported by the manufacturer, according to Hermann et al. (2007) and Mordas et al. (2008), as presented in Fig. S2. Unfortunately, the curve for the device used in these measurements is not measured; therefore, the inversion modeling uses the one reported by Mordas et al. (2008) because it lies between the other two curves, representing an average one. The uncertainty of the detection efficiency at a specific diameter is calculated from the maximum range of variation of the detection efficiencies from these three different sources.

[Figure]

**Figure S2.** The CPC 3776 detection efficiency curves as a function of particle size reported by the manufacturer, Hermann et al. (2007), and Mordas et al. (2008). Additionally, a hypothetical curve correcting the disagreement between sub-6 nm particle size distributions measured by the Nano-SMPS and PSM+CPC system is presented.

The hypothetical detection efficiency curve presented in Fig. S2 is based on the curve reported by Mordas et al. (2008) but with different parameters. If this hypothetical curve is the actual curve of the device used, the size distributions as in Fig. 4 will be corrected to the distributions presented in Fig. S3, from which it can be seen that the size distributions measured by different devices correspond clearly better, at least for the two cases having the lowest $T_{sa}$. The PSM+CPC distribution for the case having the highest $T_{sa}$ is probably overestimated because, according to Fig. 10 (c), the concentrations measured with different cut-diameters are on the same level, implying that there is not a notable amount of particles in that size range.

Other systematical errors, in addition to the uncertainty involved in the CPC 3776 detection efficiency curve, include the uncertainties of the charger efficiency, the diffusional losses, and the particle sizes interpreted by the manufacturer's inversion algorithm. The charger of the Nano-SMPS used was a TSI 3077 radioactive Kr-85 charger, which is based on charging particles bipolarly to the charge equilibrium state. The inversion algorithm uses the positive charge distribution function, $f_{charger}$, reported by Wiedensohler (1988). It is a semi-empirical function in which the mobilities and masses of positive and negative ions in the carrier gas are fitted based on the charge distribution measurements (Hussin et al., 1983; Adachi et al., 1985; Wiedensohler et al., 1986) made for particles larger than 5 nm in diameters. Alonso et al. (1997) have measured the charge distributions down to particle diameters of 2.5 nm. Unfortunately, the charger distributions from all these measurements differ, especially for the smallest particle sizes, and have thus different ion parameters, due to different particle compositions, carrier gas compositions, and the accuracies of the particle size measurements. Therefore, the charger efficiency function selected in the manufacturer's inversion algorithm is not exact. Based on the differences between the results of these charge distribution measurements, the uncertainties of 30, 20, and 10 % for the charger efficiencies at particle diameters of 6, 10, and 20 nm, respectively, are estimated. Another factor causing uncertainty for the charger efficiency is how satisfactorily the charge distribution is developed to the equilibrium state. If the residence time inside the charger is too short (Alonso et al., 1997), the activity of the charger is too low (de La Verpilliere et al., 2015) (e.g., if the activity of the charger is depleted due to a long operating life), or if the particle concentration is too high compared to the ion concentration (Wiedensohler et al., 2012), the equilibrium state may not be reached and the charger efficiency is overestimated. According to the deviations of the charge distributions in the measurements of de La Verpilliere et al. (2015) from the charge distribution function of Wiedensohler (1988), the uncertainties of 40, 30, and 20 % for the charger efficiencies at particle diameters of 2, 6, and 10 nm, respectively, are estimated. Because the particles before the Nano-SMPS are supposedly uncharged in this work, the possible incomplete reaching of the charge equilibrium state causes that the particles are less charged than predicted. Therefore, the concentrations would be underestimated, and thus the error bars related to this are considered only towards the positive direction.

[Figure]

**Figure S3.** The corrected particle size distributions as in Fig. 4 if the detection efficiency curve of the CPC 3776 would be the hypothetical curve presented in Fig. S2.

The diffusional losses of the particles in the sampling lines of this work are based on the equations of Gormley and Kennedy (1948) using the $L_{\text{lines}}/Q_{\text{lines}}$ parameter as in the case of the PSM+CPC system (the uncertainty of 10 % for the $L_{\text{lines}}/Q_{\text{lines}}$ parameter in this case is again estimated). The correction of the diffusional losses inside the Nano-SMPS device is also based on those equations in the manufacturer's inversion algorithm. The algorithm uses an empirically fitted $L_{\text{device}}/Q_{\text{device}}$ value which included the whole route of the particles inside the device even though the route is not a perfect laminar circular tube flow, for which the analytical solution by Gormley and Kennedy (1948) is based on. Therefore, the penetration function for particles inside the device, $p_{\text{device}}$, may not be exact and the uncertainty of 10 % for the $L_{\text{device}}/Q_{\text{device}}$ parameter is estimated.

The correction factor assumed to exist in the inversion algorithm of the Nano-SMPS, to which the penetration in the sampling lines, $p_{\text{lines}}$, is added, is

$$C(D_{\text{p},i}) = \frac{1}{f_{\text{charger}}(D_{\text{p},i}) \cdot f_{\text{CPC}}(D_{\text{p},i}) \cdot p_{\text{device}}(D_{\text{p},i}, L_{\text{device}}/Q_{\text{device}}) \cdot p_{\text{lines}}(D_{\text{p},i}, L_{\text{lines}}/Q_{\text{lines}})}. \tag{S5}$$

The concentration measured with a specific DMA (Differential Mobility Analyzer) voltage at a specific time, related to the particle diameter of $D_{\text{p},i}$ (obtained though the inversion algorithm), is multiplied with $C(D_{\text{p},i})$ in order to obtain the size distribution in a location of $L_{\text{lines}}$ before the device, i.e., after the ED in this case. For very small particles, all the four functions in Eq. (S5) have very low value; and thus, the value of $C(D_{\text{p},i})$ is extremely high. This is illustrated in Fig. S4 from which it can be observed that the value for sub-6 nm particles is several orders of magnitude. Very high correction factor denotes very low number of particle counts detected by the CPC at a specific diameter, and very low counts do not provide good accuracy due to statistics: there may be only a few randomly detected single particles or there may be even not a single detection at all during the time dedicated to that particle size, even though multiple scans have been performed for one measurement case.

In the case of no or very low detection of single particles, the error bars cannot be calculated. Because the correction factor increases very steeply with decreasing particle size, the uncertainties involved in the functions in Eq. (S5) can deviate it in high extent. Another consequence of the steep behavior of the correction factor is that if there is even a minor error in the interpreted particle diameters, the value of the correction factor can be significantly misestimated. There are several factors that can cause error to the particle diameters measured by the Nano-SMPS (Wiedensohler et al., 2012); here, the uncertainty of 5 % is estimated for the diameters.

[Figure]

**Figure S4.** Nano-SMPS correction factor, as in Eq. (S5), used to correct the measured particle concentrations in the data inversion.

The error bars for the size distributions, $\nu(D_{\mathrm{p},i})$, can be calculated with the equation

$$\frac{\Delta \nu}{\nu} = \sqrt{\left(\frac{\Delta \nu'}{\nu'}\right)^2 + \left(\frac{\Delta C}{C}\right)^2} \tag{S6}$$

where $\Delta \nu'$ is the standard deviation of the size distributions at the particle diameter of $D_{\mathrm{p},i}$ output by the device, $\nu'$, and $\Delta C$ is the uncertainty of the Nano-SMPS correction factor. $\Delta \nu'$ represents the standard deviation caused by the instability in the particle generation, as in the case of the PSM+CPC system, but also by the low counting statistics for the particles sizes having a very low overall detection efficiency and for the particle sizes having low concentration in the measured case. $\Delta C$ is calculated by

$$
\begin{aligned}
\frac{\Delta C}{C} &= \sqrt{\left(\frac{\partial C}{\partial f_{\mathrm{charger}}}\frac{\Delta f_{\mathrm{charger}}}{C}\right)^2 + \left(\frac{\partial C}{\partial f_{\mathrm{CPC}}}\frac{\Delta f_{\mathrm{CPC}}}{C}\right)^2 + \left(\frac{\partial C}{\partial p_{\mathrm{device}}}\frac{\Delta p_{\mathrm{device}}}{C}\right)^2 + \left(\frac{\partial C}{\partial p_{\mathrm{lines}}}\frac{\Delta p_{\mathrm{lines}}}{C}\right)^2 + \left(\frac{\partial C}{\partial D_{\mathrm{p},i}}\frac{\Delta D_{\mathrm{p},i}}{C}\right)^2} \\
&= \sqrt{\left(\frac{\Delta f_{\mathrm{charger}}}{f_{\mathrm{charger}}}\right)^2 + \left(\frac{\Delta f_{\mathrm{CPC}}}{f_{\mathrm{CPC}}}\right)^2 + \left(\frac{\Delta p_{\mathrm{device}}}{p_{\mathrm{device}}}\right)^2 + \left(\frac{\Delta p_{\mathrm{lines}}}{p_{\mathrm{lines}}}\right)^2 + \left(\frac{\partial C}{\partial D_{\mathrm{p},i}}\frac{\Delta D_{\mathrm{p},i}}{C}\right)^2}
\end{aligned} \tag{S7}
$$

where $\Delta f_{\mathrm{charger}}/f_{\mathrm{charger}}$ and $\Delta f_{\mathrm{CPC}}/f_{\mathrm{CPC}}$ are the relative uncertainties for $f_{\mathrm{charger}}$ and $f_{\mathrm{CPC}}$ having the values mentioned before, $\Delta p_{\mathrm{device}}$ and $\Delta p_{\mathrm{lines}}$ are the uncertainties for the penetration efficiencies, $p_{\mathrm{device}}$ and $p_{\mathrm{lines}}$, and the last term represents the uncertainty of $C$ caused by the uncertainty of particle diameters. $\Delta p_{\mathrm{device}}$ and $\Delta p_{\mathrm{lines}}$ depend on the particle diameter and are

calculated by

$$\Delta p_{\text{device}} = \frac{\partial p_{\text{device}}}{\partial (L_{\text{device}}/Q_{\text{device}})} \cdot \Delta(L_{\text{device}}/Q_{\text{device}})$$

$$\Delta p_{\text{lines}} = \frac{\partial p_{\text{lines}}}{\partial (L_{\text{lines}}/Q_{\text{lines}})} \cdot \Delta(L_{\text{lines}}/Q_{\text{lines}}) \tag{S8}$$

which differ from Eg. (S4) by missing the effect of the particle diameter because that effect is included in the last term of Eq. (S7). The last term represent the total effect of the particle diameter on $C$ because the particle diameter is involved in all other four terms.

**2.3 Calculated error bars for the size distributions**

The relative uncertainties for the size distributions between 2 and 10 nm after the ED caused by the different uncertainties involved in the size distribution measurements are presented in Tab. S1. For the PSM+CPC system, the most significant systematic errors arise from the uncertainty in the cut-diameters (12 ... 30 %), partly due to correcting the diffusional losses in sampling lines needed in backwards-correcting the measured distributions to represent the distributions after the ED. For the Nano-SMPS, the uncertainty of the charger efficiency plays a major role in the systematic errors (40 ... 66 %), but the uncertainty of the CPC 3776 detection efficiency curve has also a significant role for the smallest particles (55 % for 3.7 nm). The error bars decrease steeply when measuring particles sized 10 nm or larger using the Nano-SMPS. Both devices are, in theory, capable in measuring the size distribution at 3.7 nm, but the error bars for the Nano-SMPS are clearly higher compared to the PSM+CPC system. Therefore, the PSM+CPC system suits better in measuring near that diameter.

**Table S1.** The percentual uncertainties for the size distributions after the ED, $\Delta\nu/\nu$ (%), for the selected particle diameters. The first seven lines represent the systematical error of the devices and they are independent of the measurement case. The last two lines represent the effect of the standard deviation for two measurement cases having small and large particles.

| Device $D_{\text{p}}$ | PSM+CPC 2 nm | 3.7 nm | Nano-SMPS 3.7 nm | 6 nm | 10 nm |
|---|---|---|---|---|---|
| Diffusional losses in sampling lines $\left(\frac{\Delta(L_{\text{lines}}/Q_{\text{lines}})}{L_{\text{lines}}/Q_{\text{lines}}} = 10\,\%\right)$ | 8 | 3 | 4 | 2 | 1 |
| Diffusional losses in sampling lines $\left(\frac{\Delta D_{\text{p},i}}{D_{\text{p},i}} = 20\,\%\right)$ | 30 | 12 | | | |
| PSM detection efficiency $\left(\frac{\Delta(D_{\text{p},i}^{\text{larger}}/D_{\text{p},i}^{\text{smaller}})}{D_{\text{p},i}^{\text{larger}}/D_{\text{p},i}^{\text{smaller}}} = 20\,\%\right)$ | 20 | 20 | | | |
| Kr-85 charger efficiency | | | 66 | 61 | 40 |
| CPC 3776 detection efficiency | | | 55 | 8 | 0.7 |
| Diffusional losses inside the DMA $\left(\frac{\Delta(L_{\text{lines}}/Q_{\text{lines}})}{L_{\text{lines}}/Q_{\text{lines}}} = 10\,\%\right)$ | | | 16 | 7 | 3 |
| Nano-SMPS correction factor $\left(\frac{\Delta D_{\text{p},i}}{D_{\text{p},i}} = 5\,\%\right)$ | | | 32 | 17 | 10 |
| Random error in $D_{\bar{m}} = 3.6$ nm case (102 °C) | 61 | 22 | _[a] | 72 | _[a] |
| Random error in $D_{\bar{m}} = 19$ nm case (157.2 °C) | $\infty$[b] | 250 | _[a] | 24 | 10 |

[a] Cannot be calculated due to insufficient particle counts.
[b] For this point, $\nu(2\,\text{nm}) = 0$ but $\Delta\nu$ is a non-zero number due to standard deviation.

Random errors caused by the instability of the particle generation and low counting statistics of the Nano-SMPS also have significant effects for the both devices if there is not a notable amount of concentration in a specific size range. In the case with $D_{\bar{m}} = 3.6\,\text{nm}$, there is a notable amount of concentration in the PSM+CPC size range, and thus, the error bars are relatively low (22 ... 61 %) for the PSM+CPC system. For the Nano-SMPS, the standard deviation for 6 nm particles is 72 %, which is, however, the most accurately measured particle size in that case: larger particles are inexistent and smaller particles are not detected; thus, low counting statistics cause high errors. In the case with $D_{\bar{m}} = 19\,\text{nm}$, the Nano-SMPS suits well in measuring at the particle size of 10 nm (the standard deviation of 10 % originating mainly from the instability in the particle generation) and also relatively well at the particle size of 6 nm (the standard deviation of 24 %), but the errors increase with the particles smaller than 6 nm. Conversely, the PSM+CPC system has high error bars because the concentration in the PSM+CPC size range is so low that the difference between the concentrations measured with different cut-diameters are smaller than the standard deviation of the concentrations (see Fig. 10), which is always a problem with the PSM having a cumulative nature in measuring concentrations, if the cut-diameters of the adjacent saturator flow rates are too near or the measured signal is too instable. This issue can be overcome by skipping the data measured with the adjacent cut-diameters or even by considering only the data measured with the smallest and with the largest cut-diameter. However, while the error bars will be narrower in this alternative method, the information on the shape of the size distribution in that size range will diminish.

The error bars for the size distributions shown in Fig. 4 are presented in Fig. S5. By considering the error bars, the distributions from the both devices agree for the cases in panes (a) and (c), whereas the case in pane (b) has still some disagreement implying that other error sources than accounted here can be involved in the measurements using these devices. According to the error bars near the particle size of 4 nm connecting the two size distributions, the PSM+CPC system provides more reliable results in the cases in panes (a) and (b). Conversely, in the case in pane (c), the Nano-SMPS provides more reliable results because, although there are two points in the PSM+CPC distribution, the alternative method shows no particles at all. However, the error bars for the alternative method are high; thus, the probability of the existence of particles in the PSM+CPC size range is high. Nevertheless, the fraction of particles in that size range compared to the total particle count is, definitely, some orders of magnitude smaller than in the cases in panes (a) and (b). Figure S6 presents the error bars for the distributions shown in Fig. 11.

In conclusion, the suitability of the Nano-SMPS for the particle sizes smaller than $\sim$10 nm is weak especially due to the uncertainties involved in the radioactive charger efficiency and the CPC 3776 detection efficiency. Wiedensohler et al. (2012) have performed an intercomparison of several mobility particle sizers, in which the different devices provided a good agreement for the particle sizes larger than $\sim$15 nm but had significant disagreements for the smaller particle sizes, without explanation. Due to the incapability of the Nano-SMPS in determining the size distribution reliably in sub-10 nm diameter range, in all of the cases studied here and elsewhere, the PSM+CPC system suits clearly better in determining the size distribution in that particle size range, or at least the total number concentration of particles larger than $\sim$1 nm.

[Figure]

**Figure S5.** The measured size distributions for the measurement cases having the $T_{sa}$ of (a) 102 °C, (b) 135.5 °C, and (c) 157.2 °C, as shown in Fig. 4, with the error bars. The alternative PSM+CPC distributions represent the distributions using only the concentrations measured with the smallest and the largest cut-diameters, in order to narrow the error bars. The green shaded areas denote the error bars for the distributions from the alternative method. The error bars for the particle sizes from the Nano-SMPS ($\pm 5\,\%$) are not shown for clarity.

[Figure]

**Figure S6.** The simulated and measured size distributions for the measurement cases having the $T_{\mathrm{sa}}$ of (a) 102 °C, (b) 135.5 °C, and (c) 157.2 °C, as shown in Fig. 11, with the error bars. The error bars for the particle sizes from the Nano-SMPS ($\pm 5\,\%$) are not shown for clarity.

**2.4 Calculating the error bars for the diameters with the average mass**

The diameter with the average mass of a distribution is calculated by

$$D_{\bar{m}} = \left(\frac{M'}{N}\right)^{\frac{1}{3}} = \left(\frac{\sum\limits_i^{\text{PC}} \nu(D_{\text{p},i}) \cdot \text{d}\log D_{\text{p},i} \cdot D_{\text{p},i}^3 + \sum\limits_i^{\text{NS}} \nu(D_{\text{p},i}) \cdot \text{d}\log D_{\text{p},i} \cdot D_{\text{p},i}^3}{\sum\limits_i \nu(D_{\text{p},i}) \cdot \text{d}\log D_{\text{p},i}}\right)^{\frac{1}{3}} = \left(\frac{M'_{\text{PC}} + M'_{\text{NS}}}{N}\right)^{\frac{1}{3}} \tag{S9}$$

where $M'$ is the third moment of the distribution, $M'_{\text{PC}}$ and $M'_{\text{NS}}$ are the parts of the third moment from the PSM+CPC data and from the Nano-SMPS data, respectively, and $N$ is the total number concentration.

The error bars for $D_{\bar{m}}$ can be calculated with the equation

$$\frac{\Delta D_{\bar{m}}}{D_{\bar{m}}} = \sqrt{\left(\left.\frac{\Delta D_{\bar{m}}}{D_{\bar{m}}}\right|_{\Delta D_{\text{p}}}\right)^2 + \left(\left.\frac{\Delta D_{\bar{m}}}{D_{\bar{m}}}\right|_{\Delta \nu_{\text{s}}}\right)^2 + \left(\left.\frac{\Delta D_{\bar{m}}}{D_{\bar{m}}}\right|_{\Delta \nu_{\text{r}}}\right)^2} \tag{S10}$$

where the first term represents the uncertainty caused by the uncertainty of the interpreted particle diameters, the second term the uncertainty caused by the uncertainty of the number size distribution due to the systematic errors of the devices, and the last term the uncertainty caused by the uncertainty of the number size distribution due to the random error.

The first term in Eq. (S10) is separated to the effects of the PSM+CPC system and of the Nano-SMPS, respectively:

$$\left.\frac{\Delta D_{\bar{m}}}{D_{\bar{m}}}\right|_{\Delta D_{\text{p}}} = \sqrt{\left(\left.\frac{\Delta D_{\bar{m}}}{D_{\bar{m}}}\right|_{\Delta D_{\text{p},\text{PC}_1}}\right)^2 + \left(\left.\frac{\Delta D_{\bar{m}}}{D_{\bar{m}}}\right|_{\Delta D_{\text{p},\text{NS}_1}}\right)^2}. \tag{S11}$$

Because particle diameters are dependent variables for a specific device, i.e., if one diameter is shifted to a direction, other diameters are most probably shifted to the same direction and with almost the same magnitude, the diameters in Eq. (S9) are separated to dependent and independent parts:

$$D_{\text{p},i} = D_{\text{p},\text{PC}_1} \cdot D'_{\text{p},i}$$
$$D_{\text{p},i} = D_{\text{p},\text{NS}_1} \cdot D'_{\text{p},i}. \tag{S12}$$

where $D_{\text{p},\text{PC}_1}$ and $D_{\text{p},\text{NS}_1}$ denote the smallest diameters measured by the PSM+CPC system and by the Nano-SMPS, respectively, and $D'_{\text{p},i}$ is a dimensionless variable denoting the ratios of all other diameters to the smallest diameter. Hence, the third moments can be expressed as

$$M'_{\text{PC}} = \sum\limits_i^{\text{PC}} \nu(D_{\text{p},i}) \cdot \text{d}\log D_{\text{p},i} \cdot D_{\text{p},i}^3 = \sum\limits_i^{\text{PC}} \nu(D_{\text{p},i}) \cdot \text{d}\log D_{\text{p},i} \cdot (D_{\text{p},\text{PC}_1} \cdot D'_{\text{p},i})^3 = D_{\text{p},\text{PC}_1}^3 \cdot \sum\limits_i^{\text{PC}} \nu(D_{\text{p},i}) \cdot \text{d}\log D_{\text{p},i} \cdot D'^3_{\text{p},i}$$

$$M'_{\text{NS}} = \sum\limits_i^{\text{NS}} \nu(D_{\text{p},i}) \cdot \text{d}\log D_{\text{p},i} \cdot D_{\text{p},i}^3 = \sum\limits_i^{\text{NS}} \nu(D_{\text{p},i}) \cdot \text{d}\log D_{\text{p},i} \cdot (D_{\text{p},\text{NS}_1} \cdot D'_{\text{p},i})^3 = D_{\text{p},\text{NS}_1}^3 \cdot \sum\limits_i^{\text{NS}} \nu(D_{\text{p},i}) \cdot \text{d}\log D_{\text{p},i} \cdot D'^3_{\text{p},i}. \tag{S13}$$

The uncertainties in Eq. (S11) can now be calculated by

$$\left.\frac{\Delta D_{\bar{m}}}{D_{\bar{m}}}\right|_{\Delta D_{\text{p},\text{PC}_1}} = \frac{\partial D_{\bar{m}}}{\partial D_{\text{p},\text{PC}_1}} \cdot \frac{\Delta D_{\text{p},\text{PC}_1}}{D_{\bar{m}}} = \frac{1}{3}\frac{D_{\bar{m}}}{M'} \cdot \frac{\partial M'_{\text{PC}}}{\partial D_{\text{p},\text{PC}_1}} \cdot \frac{\Delta D_{\text{p},\text{PC}_1}}{D_{\bar{m}}} = \frac{M'_{\text{PC}}}{M'} \cdot \frac{\Delta D_{\text{p},\text{PC}_1}}{D_{\text{p},\text{PC}_1}}$$

$$\left.\frac{\Delta D_{\bar{m}}}{D_{\bar{m}}}\right|_{\Delta D_{\text{p},\text{NS}_1}} = \frac{\partial D_{\bar{m}}}{\partial D_{\text{p},\text{NS}_1}} \cdot \frac{\Delta D_{\text{p},\text{NS}_1}}{D_{\bar{m}}} = \frac{1}{3}\frac{D_{\bar{m}}}{M'} \cdot \frac{\partial M'_{\text{NS}}}{\partial D_{\text{p},\text{NS}_1}} \cdot \frac{\Delta D_{\text{p},\text{NS}_1}}{D_{\bar{m}}} = \frac{M'_{\text{NS}}}{M'} \cdot \frac{\Delta D_{\text{p},\text{NS}_1}}{D_{\text{p},\text{NS}_1}}. \tag{S14}$$

As 20 and 5 % uncertainties for the diameters for the PSM+CPC system and for the Nano-SMPS, respectively, were estimated, Eq. (S11) becomes

$$\left.\frac{\Delta D_{\bar{m}}}{D_{\bar{m}}}\right|_{\Delta D_{\mathrm{p}}} = \frac{1}{M'}\sqrt{\left(M'_{\mathrm{PC}}\cdot 0.2\right)^2 + \left(M'_{\mathrm{NS}}\cdot 0.05\right)^2}. \tag{S15}$$

The last two terms in Eq. (S10) related to the systematic and random errors separate the errors because the systematic errors for all size bins are presumably to the same direction and in almost the same magnitude and the random errors are randomly directed between different size bins because they are measured at different times. The number size distributions can be separated to the parts involving the sources for the systematic ($\Delta\nu_{\mathrm{s}}$) and for the random ($\Delta\nu_{\mathrm{r}}$) errors, respectively, using $\nu = \nu_{\mathrm{s}}\cdot\nu_{\mathrm{r}}$. The systematic errors for $\nu_{\mathrm{s}}$ are independent variables for the different devices, but dependent variables for the different size bins of a specific device. Hence, the second term in Eq. (S10) is further separated to the PSM+CPC system and to the Nano-SMPS, respectively, using

$$\left.\frac{\Delta D_{\bar{m}}}{D_{\bar{m}}}\right|_{\Delta\nu_{\mathrm{s}}} = \sqrt{\left(\left.\frac{\Delta D_{\bar{m}}}{D_{\bar{m}}}\right|_{\Delta\nu_{\mathrm{s,PC}}}\right)^2 + \left(\left.\frac{\Delta D_{\bar{m}}}{D_{\bar{m}}}\right|_{\Delta\nu_{\mathrm{s,NS}}}\right)^2}$$

$$= \sqrt{\left[\frac{D_{\bar{m}}\left(\nu + \Delta\nu_{\mathrm{s,PC}}\right) - D_{\bar{m}}\left(\nu - \Delta\nu_{\mathrm{s,PC}}\right)}{2D_{\bar{m}}}\right]^2 + \left[\frac{D_{\bar{m}}\left(\nu + \Delta\nu_{\mathrm{s,NS}}\right) - D_{\bar{m}}\left(\nu - \Delta\nu_{\mathrm{s,NS}}\right)}{2D_{\bar{m}}}\right]^2}. \tag{S16}$$

The last term in Eq. (S10), related to the random error, is calculated by

$$\left.\frac{\Delta D_{\bar{m}}}{D_{\bar{m}}}\right|_{\Delta\nu_{\mathrm{r}}} = \sqrt{\sum_i\left[\frac{\partial D_{\bar{m}}}{\partial\nu_{\mathrm{r}}(D_{\mathrm{p},i})}\cdot\frac{\Delta\nu_{\mathrm{r}}(D_{\mathrm{p},i})}{D_{\bar{m}}}\right]^2} = \sqrt{\sum_i\left[\left(\frac{1}{3M'}\frac{\partial M'}{\partial\nu_{\mathrm{r}}(D_{\mathrm{p},i})} - \frac{1}{3N}\frac{\partial N}{\partial\nu_{\mathrm{r}}(D_{\mathrm{p},i})}\right)\Delta\nu_{\mathrm{r}}(D_{\mathrm{p},i})\right]^2}$$

$$= \frac{1}{3}\sqrt{\sum_i\left[\left(\frac{\mathrm{d}\log D_{\mathrm{p},i}\cdot D_{\mathrm{p},i}^3}{M'} - \frac{\mathrm{d}\log D_{\mathrm{p},i}}{N}\right)\Delta\nu_{\mathrm{r}}(D_{\mathrm{p},i})\right]^2}. \tag{S17}$$

The calculated error bars for the diameters with the average mass are presented in Fig. 13.

---

## Author Response (AR3)

**Authors' response to the referee's comment for Olin et al.: "Inversely modeling homogeneous $H_2SO_4$-$H_2O$ nucleation rate in exhaust-related conditions"**

We thank the referee for a beneficial comment and have corrected the manuscript according to it.

Referee report is in *black italic* and authors' response is in blue roman font. The marked-up manuscript and Supplement showing the changes and the new versions are included at the end of this file.

**Referee 1 comments:**

*The authors have carefully addressed my criticism and revised the manuscript accordingly. Its message has been significantly clarified now and it might therefore be suitable for publication in ACP. Still, I have some concerns about the complexity of uncertainties in this study. It's not only the measurement of the sulfuric acid concentration that is still a challenging task but also size distribution measurement in the few nanometer size range is typically far from being quantitative. The size distribution results presented in this study are the best example. The authors themselves summarize all the critical aspects of size distribution measurement such as charging probability, sampling losses and instrument related transfer functions. The measured signal (number concentration at a certain DMA voltage) needs a sophisticated INVERSION algorithm to finally obtain a size distribution. In that sense it may be questioned whether the inverse modeling really brings the benefit one would hope for. Does the inversion of the inversion reduce error bars after all? I guess a careful analysis of measurement uncertainties is obligatory to get some feeling on the reliability of the results.*

We have made a more careful examination on the particle size distributions in the smallest particle size range. Firstly, we found that there has been an error in our code used to import the Nano-SMPS data: the particle diameter vector has been misaligned with the concentration matrix; thus, all size distributions have been shifted towards larger diameters with a factor of about 1.2. After correcting this error, all Nano-SMPS distributions are now in 1.2 times smaller sizes. Fortunately, this correction narrows the gaps between the distributions obtained from the PSM+CPC system and from the Nano-SMPS, as is seen in Fig. AR1. Additionally, we have changed the CPC 3776 detection efficiency curve from the one reported by the manufacturer to the curve measured by Mordas et al. (2008) because it seems that the curve can deviate clearly from the manufacturer's curve, as seen by Hermann et al. (2007) and Mordas et al. (2008).

[Figure]

**Figure AR1.** The change of the example size distributions after the correction of the error in the code.

However, there are still some discrepancies between the distributions after the correction. We have made a careful analysis of the uncertainties involved in the size distribution measurements by calculating the uncertainties for all the size distribution points at specific diameters. For the PSM+CPC system, systematic effects, such as the uncertainties in the detection efficiency curves and in the diffusional loss correction function, are taken into account. For the Nano-SMPS system, systematic effects, such as the uncertainties in the radioactive charger efficiency, in the CPC 3776 detection efficiency curve, and in the diffusional loss correction function, are taken into account. Uncertainties associated with the random effects caused by the noise in the measured concentrations due to instability in particle generation for the both devices and by low counting statistics of the Nano-SMPS at the particle sizes having very low detection efficiency or low concentration are also taken into account. The detailed information on calculating the uncertainties is now included in the Supplement; the error bars representing the uncertainties associated with both the systematic and random effects for the example distributions are also shown here in Fig. AR2. Error bars are now added also to some of the main manuscript's figures for which the clarity of the figures can be maintained; the remaining figures with the error bars are added to the Supplement.

High uncertainties in the PSM+CPC distributions arise when the standard deviations of the measured concentrations are at a same level as the difference between the concentrations measured with the adjacent cut-diameters. Therefore, using all the measured concentrations typically cause high uncertainties if there is instability in the measured signal. Alternative way to obtain the size distribution in the PSM+CPC size range is to use only the concentrations measured with the smallest and with the largest cut-diameter. The alternative method will produce better precision (error bars shown as green shaded areas), but this will, of course, diminish the information on the shape of the distribution within that size range. Considering the error bars of the distributions, it seems that in the cases in panes (a) and (b), there are particles in the PSM+CPC size range although the Nano-SMPS distributions show log-normal-like edges for the smaller sizes. In the case in pane (c), although there are two size distribution points in the PSM+CPC distribution, particles in that size range are, according to the alternative method, inexistent. However, high error bars for the alternative method, caused by the noise in the measured concentrations due to instabilities in particle generation, denote that the existence of particles in that size range is probable. Nevertheless, the fraction of particles in that size range compared to the total particle count is, definitely, some orders of magnitude smaller than in the cases in panes (a) and (b). Due to the difficulties that the Nano-SMPS has in determining the size distribution reliably in sub-10 nm diameter range, in the cases studied here, we found that the PSM+CPC system was better suited in determining the size distribution in that particle size range.

[Figure]

**Figure AR2.** The example size distributions shown with the error bars representing the uncertainties associated with both the systematic and random effects.

The inverse modeling in this work is based on predicting the concentrations measured with the different saturator flow rates of the PSM and with the CPC 3775 and on predicting the size distribution which is reported by the Nano-SMPS software. In other words, the inverse modeling does not try to predict the concentrations measured by the CPC 3776, acting as a particle counter in the Nano-SMPS system, as a function of time, or as a function of a specific DMA voltage. Concluding, there are two distinct parts of inversion involved: (1) the inverse modeling performed in this work and (2) the inversion algorithm which is included in the software of the Nano-SMPS device. These two inversion parts are not overlapped; thus, there is no "inversion of the inversion" in the analysis. The inverse modeling in this work takes the diffusional losses in the sampling lines and the detection efficiencies of the particle counters into account, while the inversion algorithm of the Nano-SMPS device takes at least the charger efficiency and the diffusional losses inside the device into account.

A consequence from correcting the error in the Nano-SMPS importing code is that because the particle diameters are now smaller, the diameters with the average mass of the distributions are also smaller now. Therefore, Fig. 13(b) and its interpretation is slightly changed now, as seen in Fig. AR3. The measured $D_{\bar{m}}$ values are now shifted towards the smaller diameters, which causes that the smallest diameters have now better agreement with the simulated diameters, but however, some diameters become less agreed with the simulated ones. The lining of the points is, nevertheless, the same: the fitted points form a slightly curved line in which the mid-ranged sizes are slightly overestimated. As some of the points now lie on the other side of the 1:1 line, minor changes to the text related to this figure are also made, but the final interpretation still remains as before.

[Figure]

**Figure AR3.** The change of the diameters with the average mass after the correction of the error in the code. The error bars, representing the uncertainties associated with both the systematic and random effects, are also added, and the colors are also changed for better clarity.

**References**

Hermann, M., Wehner, B., Bischof, O., Han, H.-S., Krinke, T., Liu, W., Zerrath, A., and Wiedensohler, A.: Particle counting efficiencies of new TSI condensation particle counters, J. Aerosol Sci., 38, 674 – 682, https://doi.org/https://doi.org/10.1016/j.jaerosci.2007.05.001, http://www.sciencedirect.com/science/article/pii/S0021850207000705, 2007.

[revised manuscript text omitted]

Nevertheless, both the measured data sets agree well with the shape of the theoretical curve, which implies that $[H_2SO_4]_{raw}$ can be estimated using $T_{sa}$. However, the absolute value for $[H_2SO_4]_{raw}$ cannot be satisfactorily estimated using neither $T_{sa}$ nor the measured concentrations due to the discrepancy of the measured concentrations. Therefore, the simulations of this study did not use the measured concentrations as the boundary conditions; instead, the $[H_2SO_4]_{raw}$ values were obtained through inverse modeling.

[Figure]

**Figure S1.** Simulated sulfuric acid concentrations in the raw sample compared to the measured and the theoretical concentrations with different sulfuric acid evaporator temperatures. The concentrations are presented as the concentrations in NTP (normal temperature and pressure) conditions rather than in a hot raw sample.

**2   Uncertainty estimation for particle size distribution measurements**

The disagreement of sub-6 nm particle size distributions measured by the combination of the PSM and the CPC 3775 and by the Nano-SMPS is examined by investigating the sources causing uncertainties to the size distributions obtained from these devices. Uncertainties associated with both the systematic and random effects in the calculated size distributions after the ejector diluter are calculated as follows.

**2.1   Calculation of the uncertainties for the size distributions measured by the combination of the PSM and the CPC 3775**

The particle number size distributions are calculated using the step-wise method according to Lehtipalo et al. (2014) . Backwards-correcting the measured distributions to represent the distributions after the ejector diluter requires multiplying the data with the dilution ratio of the bridge diluter and dividing by the penetration efficiency of particles in the sampling lines between the ejector diluter and the measurement devices. Finally, the equation to obtain the distribution at particle size $D_{p,i}$

after the ejector diluter from the measured concentrations is

$$\nu(D_{p,i}) = \frac{N_i^{\text{smaller}} - N_i^{\text{larger}}}{\log(D_{p,i}^{\text{larger}}/D_{p,i}^{\text{smaller}}) \cdot p(D_{p,i}, L/Q)} \tag{S2}$$

where $N_i^{\text{smaller}}$ and $N_i^{\text{larger}}$ are particle number concentrations measured with the $D_{50}$-cut-sizes of $D_{p,i}^{\text{smaller}}$ and $D_{p,i}^{\text{larger}}$, respectively. $D_{p,i}$ is the geometric mean diameter of the $D_{50}$-cut-sizes and $p(D_{p,i}, L/Q)$ is the penetration efficiency of particles with a diameter of $D_{p,i}$ in a sampling line with a length of $L_{\text{lines}} = L$ and flow rate of $Q_{\text{lines}} = Q$, according to diffusional losses calculated with the equations of Gormley and Kennedy (1948). This penetration efficiency takes also the bridge diluter into account because its operation principle is also based on diffusional losses; thus, $L$ denotes the effective length of the combined effect of the sampling lines and the bridge diluter.

Uncertainties associated with the systematic effects in the calculated size distributions after the ejector diluter include the uncertainty of the cut-diameters and the uncertainty of the value of $L/Q$. Because the detection efficiency curves of the PSM and CPC 3775 are measured using particles having a different composition than $H_2SO_4$-$H_2O$, as in these measurements, and because environmental parameters, such as temperature, can have effects on the detection efficiency curves, the reported cut-diameters may not hold exactly. The relative uncertainty of 20 % is estimated for the cut-diameters and also for the ratio of the cut-diameters $D_{p,i}^{\text{larger}}/D_{p,i}^{\text{smaller}}$ because it is expected that if one of the cut-diameters is deviated towards smaller or larger particle sizes, another one is deviated towards the same direction. 10 % relative uncertainty is estimated for the value of $L/Q$, which includes the uncertainty of the measurement of both $L$ and $Q$ and the uncertainty in the equations of Gormley and Kennedy (1948).

Uncertainties associated with the random effects arise from the noise in the measured concentrations caused by the instability of the particle generation. The relative standard deviations of the measured concentrations are in the range of $1 \ldots 25\,\%$, depending on the concentration level and particle sizes: higher concentrations and larger particle sizes provided more stable particle generation compared to lower concentrations and smaller particle sizes.

The uncertainty associated with both the systematic and random effects for $\nu(D_{p,i})$ can be calculated with the equation

$$\frac{\Delta\nu}{\nu} = \sqrt{\left(\frac{\partial\nu}{\partial N_i^{\text{smaller}}}\frac{\Delta N_i^{\text{smaller}}}{\nu}\right)^2 + \left(\frac{\partial\nu}{\partial N_i^{\text{larger}}}\frac{\Delta N_i^{\text{larger}}}{\nu}\right)^2 + \left(\frac{\partial\nu}{\partial\log(D_{p,i}^{\text{larger}}/D_{p,i}^{\text{smaller}})}\frac{\Delta\log(D_{p,i}^{\text{larger}}/D_{p,i}^{\text{smaller}})}{\nu}\right)^2 + \left(\frac{\partial\nu}{\partial p}\frac{\Delta p}{\nu}\right)^2}$$

$$= \sqrt{\left(\frac{\Delta N_i^{\text{smaller}}}{N_i^{\text{smaller}} - N_i^{\text{larger}}}\right)^2 + \left(\frac{\Delta N_i^{\text{larger}}}{N_i^{\text{smaller}} - N_i^{\text{larger}}}\right)^2 + \left(\frac{\Delta\log(D_{p,i}^{\text{larger}}/D_{p,i}^{\text{smaller}})}{\log(D_{p,i}^{\text{larger}}/D_{p,i}^{\text{smaller}})}\right)^2 + \left(\frac{\Delta p}{p}\right)^2} \tag{S3}$$

where $\Delta N_i^{\text{smaller}}$ and $\Delta N_i^{\text{smaller}}$ are the standard deviations of the measured concentrations, depending on the measurement case, the third term is $0.2^2$ because $\Delta(D_{p,i}^{\text{larger}}/D_{p,i}^{\text{smaller}})/(D_{p,i}^{\text{larger}}/D_{p,i}^{\text{smaller}}) = 20\,\%$, and $\Delta p/p$ is the relative uncertainty for $p$ depending on the particle size and is calculated with the equation

$$\frac{\Delta p}{p} = \sqrt{\left(\frac{\partial p}{\partial(L/Q)}\frac{\Delta(L/Q)}{p}\right)^2 + \left(\frac{\partial p}{\partial D_{p,i}}\frac{\Delta D_{p,i}}{p}\right)^2} \tag{S4}$$

using $\Delta(L/Q)/(L/Q) = 10\,\%$ and $\Delta D_{p,i}/D_{p,i} = 20\,\%$.

**2.2 Calculation of the uncertainties for the size distributions measured by the Nano-SMPS**

The particle number size distributions reported by the Nano-SMPS device have already went through the manufacturer's inversion algorithm. Thus, the inverse modeling of this work does not try to predict the concentration measured as a function

of time measured by the CPC 3776, acting as a particle counter in the Nano-SMPS system. Instead, the inverse modeling takes only the diffusional losses in the sampling lines and the CPC 3776 detection efficiency curve into account, but not, e.g., the radioactive charger efficiency and the diffusional losses inside the device. It is partly unknown what is included in the manufacturer's inversion algorithm, but at least the charger efficiency and the diffusional losses inside the device are included. The inversion algorithm probably includes also the CPC 3776 detection efficiency curve, $f_{CPC}$, but it is, however, included in the inverse modeling of this work because it seems that it may differ significantly from the curve reported by the manufacturer, according to Hermann et al. (2007) and Mordas et al. (2008), as presented in Fig. S2. Unfortunately, the curve for the device used in these measurements is not measured; therefore, the inversion modeling uses the one reported by Mordas et al. (2008) because it lies between the other two curves, representing an average one. The uncertainty of the detection efficiency at a specific diameter is calculated from the maximum range of variation of the detection efficiencies from these three different sources.

[Figure]

**Figure S2.** The CPC 3776 detection efficiency curves as a function of particle size reported by the manufacturer, Hermann et al. (2007), and Mordas et al. (2008). Additionally, a hypothetical curve correcting the disagreement between sub-6 nm particle size distributions measured by the Nano-SMPS and PSM+CPC system is presented.

The hypothetical detection efficiency curve presented in Fig. S2 is based on the curve reported by Mordas et al. (2008) but with different parameters. If this hypothetical curve is the actual curve of the device used, the size distributions as in Fig. 4 will be corrected to the distributions presented in Fig. S3, from which it can be seen that the size distributions measured by different devices correspond clearly better, at least for the two cases having the lowest $T_{sa}$. The PSM+CPC distribution for the case having the highest $T_{sa}$ is probably overestimated due to noise in the measured concentration because, according to Fig. 10 (c), the concentrations measured with different cut-diameters are on the same level, implying that there should not be a notable amount of particles in that size range.

Other uncertainties associated with the systematic effects, in addition to the uncertainty involved in the CPC 3776 detection efficiency curve, include the uncertainties of the charger efficiency, the diffusional losses correction, and the particle sizes interpreted by the manufacturer's inversion algorithm. The charger of the Nano-SMPS used was a TSI 3077 radioactive Kr-85 charger, which is based on charging particles bipolarly to the charge equilibrium state. The inversion algorithm uses the positive charge distribution function, $f_{charger}$, reported by Wiedensohler (1988). It is a semi-empirical function in which the mobilities and masses of positive and negative ions in the carrier gas are fitted based on the charge distribution measurements (Hussin et al., 1983; Adachi et al., 1985; Wiedensohler et al., 1986) made for particles larger than 5 nm in diameters.

Alonso et al. (1997) have measured the charge distributions down to particle diameters of 2.5 nm. Unfortunately, the charger distributions from all these measurements differ, especially for the smallest particle sizes, and have thus different ion parameters, due to different particle compositions, carrier gas compositions, and the accuracies of the particle size measurements. Therefore,

[Figure]

**Figure S3.** The corrected particle size distributions as in Fig. 4 if the detection efficiency curve of the CPC 3776 would be the hypothetical curve presented in Fig. S2.

the charger efficiency function selected in the manufacturer's inversion algorithm may not be very accurate. Based on the differences between the results of these charge distribution measurements, the relative uncertainties of 30, 20, and 10 % for the charger efficiencies at particle diameters of 6, 10, and 20 nm, respectively, are estimated. Another factor causing uncertainty for the charger efficiency is how satisfactorily the charge distribution is developed to the equilibrium state. If the residence time inside the charger is too short (Alonso et al., 1997), the activity of the charger is too low (de La Verpilliere et al., 2015) (e.g, if the activity of the charger is depleted due to a long operating life), or if the particle concentration is too high compared to the ion concentration (Wiedensohler et al., 2012), the equilibrium state may not be reached and the charger efficiency is overestimated. According to the deviations of the charge distributions in the measurements of de La Verpilliere et al. (2015) from the charge distribution function of Wiedensohler (1988), the relative uncertainties of 40, 30, and 20 % for the charger efficiencies at particle diameters of 2, 6, and 10 nm, respectively, are estimated. Because the particles before the Nano-SMPS are supposedly uncharged in this work, the possible incomplete reaching of the charge equilibrium state causes that the particles are less charged than predicted. Therefore, the concentrations would be underestimated, and thus the possible error related to this is considered only negative.

The diffusional losses of the particles in the sampling lines of this work are based on the equations of Gormley and Kennedy (1948) using the $L_{\text{lines}}/Q_{\text{lines}}$ parameter as in the case of the PSM+CPC system (the relative uncertainty of 10 % for the $L_{\text{lines}}/Q_{\text{lines}}$ parameter in this case is again estimated). The correction of the diffusional losses inside the Nano-SMPS device is also based on those equations in the manufacturer's inversion algorithm. The algorithm uses an empirically fitted $L_{\text{device}}/Q_{\text{device}}$ value which included the whole route of the particles inside the device even though the route is not a perfect laminar circular tube flow, for which the analytical solution by Gormley and Kennedy (1948) is based on. Therefore, the penetration function for particles inside the device, $p_{\text{device}}$, may not be very accurate and the relative uncertainty of 10 % for the $L_{\text{device}}/Q_{\text{device}}$ parameter is estimated.

The correction factor assumed to exist in the inversion algorithm of the Nano-SMPS, to which the penetration in the sampling lines, $p_{\text{lines}}$, is added, is

$$C(D_{\text{p},i}) = \frac{1}{f_{\text{charger}}(D_{\text{p},i}) \cdot f_{\text{CPC}}(D_{\text{p},i}) \cdot p_{\text{device}}(D_{\text{p},i}, L_{\text{device}}/Q_{\text{device}}) \cdot p_{\text{lines}}(D_{\text{p},i}, L_{\text{lines}}/Q_{\text{lines}})}. \tag{S5}$$

The concentration measured with a specific DMA (Differential Mobility Analyzer) voltage at a specific time, related to the particle diameter of $D_{\text{p},i}$ (obtained though the inversion algorithm), is multiplied with $C(D_{\text{p},i})$ in order to obtain the size distribution in a location of $L_{\text{lines}}$ before the device, i.e., after the ejector diluter in this case. For very small particles, all the four functions in Eq. (S5) have very low value; and thus, the value of $C(D_{\text{p},i})$ is extremely high. This is illustrated in Fig. S4 from which it can be observed that the value for sub-6 nm particles is several orders of magnitude. Very high correction factor denotes very low number of particle counts detected by the CPC at a specific diameter, and very low counts do not provide good precision due to statistics: there may be only a few randomly detected single particles or there may be even not a single detection at all during the time dedicated to that particle size, even though multiple scans have been performed for one measurement case. In the case of no or very low detection of single particles, uncertainties cannot be calculated. Because the correction factor increases very steeply with decreasing particle size, the uncertainties involved in the functions in Eq. (S5) can deviate it in high extent. Another consequence of the steep behavior of the correction factor is that if there is even a minor error in the interpreted particle diameters, the value of the correction factor can be significantly misestimated. There are several factors that can cause error to the particle diameters measured by the Nano-SMPS (Wiedensohler et al., 2012) ; here, the relative uncertainty of 5 % is estimated for the diameters.

[Figure]

**Figure S4.** Nano-SMPS correction factor, as in Eq. (S5), used to correct the measured particle concentrations in the data inversion.

The uncertainty associated with both the systematic and random effects for the size distributions, $\nu(D_{\text{p},i})$, can be calculated with the equation

$$\frac{\Delta\nu}{\nu} = \sqrt{\left(\frac{\Delta\nu'}{\nu'}\right)^2 + \left(\frac{\Delta C}{C}\right)^2} \tag{S6}$$

where $\Delta\nu'/\nu'$ is the relative standard deviation of the size distributions at the particle diameter of $D_{\text{p},i}$ output by the device and $\Delta C/C$ is the relative uncertainty of the Nano-SMPS correction factor. $\Delta\nu'/\nu'$ represents uncertainties associated with the

random effects arisen from the noise in the measured concentrations caused by the instability of the particle generation, as in the case of the PSM+CPC system, but also from low precision in measuring particles sizes having a very low overall detection efficiency and particle sizes having low concentration in the measured case. $\Delta C/C$ is calculated with the equation

$$\frac{\Delta C}{C} = \sqrt{\left(\frac{\partial C}{\partial f_{charger}}\frac{\Delta f_{charger}}{C}\right)^2 + \left(\frac{\partial C}{\partial f_{CPC}}\frac{\Delta f_{CPC}}{C}\right)^2 + \left(\frac{\partial C}{\partial p_{device}}\frac{\Delta p_{device}}{C}\right)^2 + \left(\frac{\partial C}{\partial p_{lines}}\frac{\Delta p_{lines}}{C}\right)^2 + \left(\frac{\partial C}{\partial D_{p,i}}\frac{\Delta D_{p,i}}{C}\right)^2}$$

$$= \sqrt{\left(\frac{\Delta f_{charger}}{f_{charger}}\right)^2 + \left(\frac{\Delta f_{CPC}}{f_{CPC}}\right)^2 + \left(\frac{\Delta p_{device}}{p_{device}}\right)^2 + \left(\frac{\Delta p_{lines}}{p_{lines}}\right)^2 + \left(\frac{\partial C}{\partial D_{p,i}}\frac{\Delta D_{p,i}}{C}\right)^2} \tag{S7}$$

where $\Delta f_{charger}/f_{charger}$ and $\Delta f_{CPC}/f_{CPC}$ are the relative uncertainties for $f_{charger}$ and $f_{CPC}$ having the values mentioned before, $\Delta p_{device}/p_{device}$ and $\Delta p_{lines}/p_{lines}$ are the relative uncertainties for the penetration efficiencies, $p_{device}$ and $p_{lines}$, and the last term represents the relative uncertainty of $C$ caused by the uncertainty of particle diameters. $\Delta p_{device}/p_{device}$ and $\Delta p_{lines}/p_{lines}$ depend on the particle diameter and are calculated with the equations

$$\frac{\Delta p_{device}}{p_{device}} = \frac{\partial p_{device}}{\partial(L_{device}/Q_{device})} \cdot \frac{\Delta(L_{device}/Q_{device})}{p_{device}}$$

$$\frac{\Delta p_{lines}}{p_{lines}} = \frac{\partial p_{lines}}{\partial(L_{lines}/Q_{lines})} \cdot \frac{\Delta(L_{lines}/Q_{lines})}{p_{lines}} \tag{S8}$$

which differ from Eg. (S4) by missing the effect of the particle diameter because that effect is included in the last term of Eq. (S7). The last term represents the total effect of the uncertainty of particle diameter on the uncertainty of $C$ because particle diameter is involved in all the other four terms.

**2.3 Calculated uncertainties for the size distributions**

The relative uncertainties for the size distributions between 2 and 10 nm after the ejector diluter caused by the uncertainties associated with the different effects involved in the size distribution measurements are presented in Tab. S1. For the PSM+CPC system, the most significant relative uncertainties associated with the systematic effects arise from the uncertainty of the cut-diameters (12 ... 30 %), partly due to correcting the diffusional losses in sampling lines needed in backwards-correcting the measured distributions to represent the distributions after the ejector diluter. For the Nano-SMPS, the uncertainty of the charger efficiency plays a major role in the relative uncertainties associated with the systematic effects (40 ... 66 %), but the uncertainty of the CPC 3776 detection efficiency curve has also a significant role for the smallest particles (55 % for 3.7 nm). The relative uncertainties decrease steeply when measuring particles sized 10 nm or larger using the Nano-SMPS. Both devices are, in theory, capable in measuring the size distribution at 3.7 nm, but the uncertainties with the Nano-SMPS are clearly higher compared to the PSM+CPC system. Therefore, the PSM+CPC system suits better in measuring near that diameter.

Uncertainties associated with the random effects caused by the noise in the measured concentrations due to instability in particle generation for the both devices and by low counting statistics of the Nano-SMPS also have significant effects if there is not a notable amount of concentration in a specific size range. In the case with $D_{\bar{m}} = 3.6\,\text{nm}$, there is a notable amount of concentration in the PSM+CPC size range, and thus, the relative uncertainties are relatively low (22 ... 61 %) for the PSM+CPC system. For the Nano-SMPS, the relative standard deviation of the size distribution for the 6 nm particles is 72 %, which is, however, the particle size measured with the highest precision in that case: larger particles are inexistent and smaller particles are not detected. In the case with $D_{\bar{m}} = 19\,\text{nm}$, the Nano-SMPS suits well in measuring the size distribution at the particle size of 10 nm (the relative standard deviation of 10 % originating mainly from the instability in the particle generation) and also relatively well at the particle size of 6 nm (the relative standard deviation of 24 %), but the uncertainties increase with the particles smaller than 6 nm. Conversely, the PSM+CPC system has high uncertainties because the concentration in the PSM+CPC size range is so low that the difference between the concentrations measured with different cut-diameters are

**Table S1.** The relative uncertainties (in percents) for the size distributions after the ejector diluter, $\Delta\nu/\nu$ (%), for the selected particle diameters. The first seven lines represent the relative uncertainties associated with the systematic effects and are thus independent of the measurement cases. The last two lines represent the relative uncertainties associated with the random effects for two selected measurement cases having small and large particles.

| Device | PSM+CPC | | Nano-SMPS | | |
|---|---|---|---|---|---|
| $D_\mathrm{p}$ | 2 nm | 3.7 nm | 3.7 nm | 6 nm | 10 nm |
| Diffusional losses in sampling lines $\left(\frac{\Delta(L_\mathrm{lines}/Q_\mathrm{lines})}{L_\mathrm{lines}/Q_\mathrm{lines}}=10\,\%\right)$ | 8 | 3 | 4 | 2 | 1 |
| Diffusional losses in sampling lines $\left(\frac{\Delta D_{\mathrm{p},i}}{D_{\mathrm{p},i}}=20\,\%\right)$ | 30 | 12 | | | |
| PSM detection efficiency $\left(\frac{\Delta(D_{\mathrm{p},i}^{\mathrm{larger}}/D_{\mathrm{p},i}^{\mathrm{smaller}})}{D_{\mathrm{p},i}^{\mathrm{larger}}/D_{\mathrm{p},i}^{\mathrm{smaller}}}=20\,\%\right)$ | 20 | 20 | | | |
| Kr-85 charger efficiency | | | 66 | 61 | 40 |
| CPC 3776 detection efficiency | | | 55 | 8 | 0.7 |
| Diffusional losses inside the DMA $\left(\frac{\Delta(L_\mathrm{lines}/Q_\mathrm{lines})}{L_\mathrm{lines}/Q_\mathrm{lines}}=10\,\%\right)$ | | | 16 | 7 | 3 |
| Nano-SMPS correction factor $\left(\frac{\Delta D_{\mathrm{p},i}}{D_{\mathrm{p},i}}=5\,\%\right)$ | | | 32 | 17 | 10 |
| Random effects in $D_{\bar{m}}=3.6\,\mathrm{nm}$ case (102 °C) | 61 | 22 | —[a] | 72 | —[a] |
| Ramdom effects in $D_{\bar{m}}=19\,\mathrm{nm}$ case (157.2 °C) | $\infty$[b] | 250 | —[a] | 24 | 10 |

[a] Cannot be calculated due to insufficient particle counts.
[b] For this point, $\nu(2\,\mathrm{nm})=0$ but $\Delta\nu$ is a non-zero number due to the standard deviations of $N_i^{\mathrm{smaller}}$ and $N_i^{\mathrm{smaller}}$.

smaller than the standard deviation of the concentrations (see Fig. 10), which is always a problem with the PSM having a cumulative nature in measuring concentrations, if the cut-diameters of the adjacent saturator flow rates are too near or the measured signal is too unstable. This issue can be overcome by skipping the data measured with the adjacent cut-diameters or even by considering only the data measured with the smallest and with the largest cut-diameter. However, while the precision will be higher in this alternative method, the information on the shape of the size distribution in that size range will diminish.

The error bars representing the uncertainties associated with both the systematic and random effects for the size distributions shown in Fig. 4 are presented in Fig. S5. By considering the error bars, the distributions from the both devices agree for the cases in panes (a) and (c), whereas the case in pane (b) has still some disagreement implying that other sources of uncertainty than accounted here can be involved in the measurements using these devices. According to the error bars near the particle size of 4 nm connecting the two size distributions, the PSM+CPC system provides more reliable results in the cases in panes (a) and (b). Conversely, in the case in pane (c), the Nano-SMPS provides more reliable results because, although there are two points in the PSM+CPC distribution, the alternative method shows no particles at all. However, the error bars for the alternative method are high; thus, the existence of particles in the PSM+CPC size range is still probable. Nevertheless, the fraction of particles in that size range compared to the total particle count is, definitely, some orders of magnitude smaller than in the cases in panes (a) and (b). The distributions and their uncertainties near the particle size of 4 nm, where the distributions from the both devices are available, are decided to keep separated here due to high systematic error possible in the Nano-SMPS data, although by combining the distributions would cause lower overall uncertainties. Figure S6 presents the error bars for the distributions shown in Fig. 11.

In conclusion, the reliability of our Nano-SMPS system was low for the particle sizes smaller than ∼10 nm, for the most part due to the uncertainties involved in the radioactive charger efficiency and the CPC 3776 detection efficiency. Wiedensohler et al. (2012) have performed an intercomparison of several mobility particle sizers, in which the different devices

[Figure]

**Figure S5.** The measured size distributions for the measurement cases having the $T_{sa}$ of (a) 102 °C, (b) 135.5 °C, and (c) 157.2 °C, as shown in Fig. 4, with the error bars representing the uncertainties associated with both the systematic and random effects. The alternative PSM+CPC distributions represent the distributions using only the concentrations measured with the smallest and the largest cut-diameters, in order to increase the precision. The green shaded areas denote the error bars for the distributions from the alternative method. The error bars for the particle sizes obtained from the Nano-SMPS (±5 %) are not shown for clarity.

[Figure]

**Figure S6.** The simulated and measured size distributions for the measurement cases having the $T_{sa}$ of (a) 102 °C, (b) 135.5 °C, and (c) 157.2 °C, as shown in Fig. 11, with the error bars representing the uncertainties associated with both the systematic and random effects. The error bars for the particle sizes obtained from the Nano-SMPS ($\pm 5$ %) are not shown for clarity.

provided a good agreement for the particle sizes larger than ~15 nm but had significant disagreements for the smaller particle sizes, without explanation. Due to the difficulties that the Nano-SMPS has in determining the size distribution reliably in sub-10 nm diameter range, in the cases studied here and elsewhere, we found that the PSM+CPC system was better suited in determining the size distribution in that particle size range, or at least in determining the total number concentration of particles larger than ~1 nm.

**2.4 Calculation of the uncertainties for the diameters with the average mass**

The diameter with the average mass of a distribution is calculated by

$$
D_{\bar{m}} = \left( \frac{M'}{N} \right)^{\frac{1}{3}} = \left( \frac{\sum_i^{PC} \nu(D_{p,i}) \cdot d\log D_{p,i} \cdot D_{p,i}^3 + \sum_i^{NS} \nu(D_{p,i}) \cdot d\log D_{p,i} \cdot D_{p,i}^3}{\sum_i \nu(D_{p,i}) \cdot d\log D_{p,i}} \right)^{\frac{1}{3}} = \left( \frac{M'_{PC} + M'_{NS}}{N} \right)^{\frac{1}{3}} \tag{S9}
$$

where $M'$ is the third moment of the distribution, $M'_{PC}$ and $M'_{NS}$ are the parts of the third moment from the PSM+CPC data and from the Nano-SMPS data, respectively, and $N$ is the total number concentration.

The relative uncertainty for $D_{\bar{m}}$ can be calculated with the equation

$$
\frac{\Delta D_{\bar{m}}}{D_{\bar{m}}} = \sqrt{ \left( \left. \frac{\Delta D_{\bar{m}}}{D_{\bar{m}}} \right|_{\Delta D_p} \right)^2 + \left( \left. \frac{\Delta D_{\bar{m}}}{D_{\bar{m}}} \right|_{\Delta \nu_s} \right)^2 + \left( \left. \frac{\Delta D_{\bar{m}}}{D_{\bar{m}}} \right|_{\Delta \nu_r} \right)^2 } \tag{S10}
$$

where the first term represents the relative uncertainty caused by the uncertainty of the interpreted particle diameters, the second term the relative uncertainty caused by the uncertainty associated with the systematic effects for the number size distribution, and the last term the relative uncertainty caused by the uncertainty associated with the random effects for the number size distribution.

The first term in Eq. (S10) is separated to the effects of the PSM+CPC system and of the Nano-SMPS, respectively:

$$
\left. \frac{\Delta D_{\bar{m}}}{D_{\bar{m}}} \right|_{\Delta D_p} = \sqrt{ \left( \left. \frac{\Delta D_{\bar{m}}}{D_{\bar{m}}} \right|_{\Delta D_{p,PC_1}} \right)^2 + \left( \left. \frac{\Delta D_{\bar{m}}}{D_{\bar{m}}} \right|_{\Delta D_{p,NS_1}} \right)^2 } . \tag{S11}
$$

Because particle diameters are dependent variables for a specific device, i.e., if one diameter is shifted to a direction, other diameters are most probably shifted to the same direction and with almost the same magnitude, the diameters in Eq. (S9) are separated to dependent and independent parts:

$$
D_{p,i} = D_{p,PC_1} \cdot D'_{p,i}
$$

$$
D_{p,i} = D_{p,NS_1} \cdot D'_{p,i} . \tag{S12}
$$

where $D_{p,PC_1}$ and $D_{p,NS_1}$ denote the smallest diameters measured by the PSM+CPC system and by the Nano-SMPS, respectively, and $D'_{p,i}$ is a dimensionless variable denoting the ratios of all other diameters to the smallest diameter. Hence, the third moments can be expressed as

$$
M'_{PC} = \sum_i^{PC} \nu(D_{p,i}) \cdot d\log D_{p,i} \cdot D_{p,i}^3 = \sum_i^{PC} \nu(D_{p,i}) \cdot d\log D_{p,i} \cdot (D_{p,PC_1} \cdot D'_{p,i})^3 = D_{p,PC_1}^3 \cdot \sum_i^{PC} \nu(D_{p,i}) \cdot d\log D_{p,i} \cdot D'^3_{p,i}
$$

$$
M'_{NS} = \sum_i^{NS} \nu(D_{p,i}) \cdot d\log D_{p,i} \cdot D_{p,i}^3 = \sum_i^{NS} \nu(D_{p,i}) \cdot d\log D_{p,i} \cdot (D_{p,NS_1} \cdot D'_{p,i})^3 = D_{p,NS_1}^3 \cdot \sum_i^{NS} \nu(D_{p,i}) \cdot d\log D_{p,i} \cdot D'^3_{p,i} . \tag{S13}
$$

The relative uncertainties in Eq. (S11) can now be calculated by

$$\frac{\Delta D_{\bar{m}}}{D_{\bar{m}}}\bigg|_{\Delta D_{p,PC_1}} = \frac{\partial D_{\bar{m}}}{\partial D_{p,PC_1}} \cdot \frac{\Delta D_{p,PC_1}}{D_{\bar{m}}} = \frac{1}{3}\frac{D_{\bar{m}}}{M'} \cdot \frac{\partial M'_{PC}}{\partial D_{p,PC_1}} \cdot \frac{\Delta D_{p,PC_1}}{D_{\bar{m}}} = \frac{M'_{PC}}{M'} \cdot \frac{\Delta D_{p,PC_1}}{D_{p,PC_1}}$$

$$\frac{\Delta D_{\bar{m}}}{D_{\bar{m}}}\bigg|_{\Delta D_{p,NS_1}} = \frac{\partial D_{\bar{m}}}{\partial D_{p,NS_1}} \cdot \frac{\Delta D_{p,NS_1}}{D_{\bar{m}}} = \frac{1}{3}\frac{D_{\bar{m}}}{M'} \cdot \frac{\partial M'_{NS}}{\partial D_{p,NS_1}} \cdot \frac{\Delta D_{p,NS_1}}{D_{\bar{m}}} = \frac{M'_{NS}}{M'} \cdot \frac{\Delta D_{p,NS_1}}{D_{p,NS_1}}. \tag{S14}$$

As 20 and 5 % relative uncertainties for the diameters for the PSM+CPC system and for the Nano-SMPS, respectively, were estimated, Eq. (S11) becomes

$$\frac{\Delta D_{\bar{m}}}{D_{\bar{m}}}\bigg|_{\Delta D_p} = \frac{1}{M'}\sqrt{\left(M'_{PC} \cdot 0.2\right)^2 + \left(M'_{NS} \cdot 0.05\right)^2}. \tag{S15}$$

The relative uncertainties associated with the systematic and random effects are separated in the last two terms in Eq. (S10) because the possible errors due to the systematic effects for all size bins are presumably to the same direction and in almost of the same magnitude and the possible errors due to the random effects are randomly directed between different size bins because they are measured at different times. The number size distributions can be separated to the parts involving the sources of uncertainties associated with the systematic ($\Delta\nu_s$) and with the random ($\Delta\nu_r$) effects, respectively, using $\nu = \nu_s \cdot \nu_r$. The systematic effects for the uncertainty of $\nu_s$ involve independent variables between the different devices, but dependent variables between the different size bins of a specific device. Hence, the second term in Eq. (S10) is further separated to the PSM+CPC system and to the Nano-SMPS, respectively, using

$$\frac{\Delta D_{\bar{m}}}{D_{\bar{m}}}\bigg|_{\Delta\nu_s} = \sqrt{\left(\frac{\Delta D_{\bar{m}}}{D_{\bar{m}}}\bigg|_{\Delta\nu_{s,PC}}\right)^2 + \left(\frac{\Delta D_{\bar{m}}}{D_{\bar{m}}}\bigg|_{\Delta\nu_{s,NS}}\right)^2}$$

$$= \sqrt{\left[\frac{D_{\bar{m}}(\nu + \Delta\nu_{s,PC}) - D_{\bar{m}}(\nu - \Delta\nu_{s,PC})}{2D_{\bar{m}}}\right]^2 + \left[\frac{D_{\bar{m}}(\nu + \Delta\nu_{s,NS}) - D_{\bar{m}}(\nu - \Delta\nu_{s,NS})}{2D_{\bar{m}}}\right]^2}. \tag{S16}$$

The last term in Eq. (S10), related to the relative uncertainties associated with the random effects, is calculated by

$$\frac{\Delta D_{\bar{m}}}{D_{\bar{m}}}\bigg|_{\Delta\nu_r} = \sqrt{\sum_i \left[\frac{\partial D_{\bar{m}}}{\partial\nu_r(D_{p,i})} \cdot \frac{\Delta\nu_r(D_{p,i})}{D_{\bar{m}}}\right]^2} = \sqrt{\sum_i \left[\left(\frac{1}{3M'}\frac{\partial M'}{\partial\nu_r(D_{p,i})} - \frac{1}{3N}\frac{\partial N}{\partial\nu_r(D_{p,i})}\right)\Delta\nu_r(D_{p,i})\right]^2}$$

$$= \frac{1}{3}\sqrt{\sum_i \left[\left(\frac{d\log D_{p,i} \cdot D_{p,i}^3}{M'} - \frac{d\log D_{p,i}}{N}\right)\Delta\nu_r(D_{p,i})\right]^2}. \tag{S17}$$

The calculated error bars representing the uncertainties associated with both the systematic and random effects for the diameters with the average mass are presented in Fig. 13.

[revised manuscript text omitted]

Nevertheless, both the measured data sets agree well with the shape of the theoretical curve, which implies that $[\mathrm{H_2SO_4}]_\mathrm{raw}$ can be estimated using $T_\mathrm{sa}$. However, the absolute value for $[\mathrm{H_2SO_4}]_\mathrm{raw}$ cannot be satisfactorily estimated using neither $T_\mathrm{sa}$ nor the measured concentrations due to the discrepancy of the measured concentrations. Therefore, the simulations of this study did not use the measured concentrations as the boundary conditions; instead, the $[\mathrm{H_2SO_4}]_\mathrm{raw}$ values were obtained through inverse modeling.

[Figure]

**Figure S1.** Simulated sulfuric acid concentrations in the raw sample compared to the measured and the theoretical concentrations with different sulfuric acid evaporator temperatures. The concentrations are presented as the concentrations in NTP (normal temperature and pressure) conditions rather than in a hot raw sample.

**2 Uncertainty estimation for particle size distribution measurements**

The disagreement of sub-6 nm particle size distributions measured by the combination of the PSM and the CPC 3775 and by the Nano-SMPS is examined by investigating the sources causing uncertainties to the size distributions obtained from these devices. Uncertainties associated with both the systematic and random effects in the calculated size distributions after the ejector diluter are calculated as follows.

**2.1 Calculation of the uncertainties for the size distributions measured by the combination of the PSM and the CPC 3775**

The particle number size distributions are calculated using the step-wise method according to Lehtipalo et al. (2014). Backwards-correcting the measured distributions to represent the distributions after the ejector diluter requires multiplying the data with the dilution ratio of the bridge diluter and dividing by the penetration efficiency of particles in the sampling lines between the ejector diluter and the measurement devices. Finally, the equation to obtain the distribution at particle size $D_{\mathrm{p},i}$ after the ejector

diluter from the measured concentrations is

$$\nu(D_{\mathrm{p},i}) = \frac{N_i^{\mathrm{smaller}} - N_i^{\mathrm{larger}}}{\log(D_{\mathrm{p},i}^{\mathrm{larger}}/D_{\mathrm{p},i}^{\mathrm{smaller}}) \cdot p(D_{\mathrm{p},i}, L/Q)} \tag{S2}$$

where $N_i^{\mathrm{smaller}}$ and $N_i^{\mathrm{larger}}$ are particle number concentrations measured with the $D_{50}$-cut-sizes of $D_{\mathrm{p},i}^{\mathrm{smaller}}$ and $D_{\mathrm{p},i}^{\mathrm{larger}}$, respectively. $D_{\mathrm{p},i}$ is the geometric mean diameter of the $D_{50}$-cut-sizes and $p(D_{\mathrm{p},i}, L/Q)$ is the penetration efficiency of particles with a diameter of $D_{\mathrm{p},i}$ in a sampling line with a length of $L_{\mathrm{lines}} = L$ and flow rate of $Q_{\mathrm{lines}} = Q$, according to diffusional losses calculated with the equations of Gormley and Kennedy (1948). This penetration efficiency takes also the bridge diluter into account because its operation principle is also based on diffusional losses; thus, $L$ denotes the effective length of the combined effect of the sampling lines and the bridge diluter.

Uncertainties associated with the systematic effects in the calculated size distributions after the ejector diluter include the uncertainty of the cut-diameters and the uncertainty of the value of $L/Q$. Because the detection efficiency curves of the PSM and CPC 3775 are measured using particles having a different composition than $H_2SO_4$-$H_2O$, as in these measurements, and because environmental parameters, such as temperature, can have effects on the detection efficiency curves, the reported cut-diameters may not hold exactly. The relative uncertainty of 20 % is estimated for the cut-diameters and also for the ratio of the cut-diameters $D_{\mathrm{p},i}^{\mathrm{larger}}/D_{\mathrm{p},i}^{\mathrm{smaller}}$ because it is expected that if one of the cut-diameters is deviated towards smaller or larger particle sizes, another one is deviated towards the same direction. 10 % relative uncertainty is estimated for the value of $L/Q$, which includes the uncertainty of the measurement of both $L$ and $Q$ and the uncertainty in the equations of Gormley and Kennedy (1948).

Uncertainties associated with the random effects arise from the noise in the measured concentrations caused by the instability of the particle generation. The relative standard deviations of the measured concentrations are in the range of 1 ... 25 %, depending on the concentration level and particle sizes: higher concentrations and larger particle sizes provided more stable particle generation compared to lower concentrations and smaller particle sizes.

The uncertainty associated with both the systematic and random effects for $\nu(D_{\mathrm{p},i})$ can be calculated with the equation

$$\frac{\Delta\nu}{\nu} = \sqrt{\left(\frac{\partial\nu}{\partial N_i^{\mathrm{smaller}}}\frac{\Delta N_i^{\mathrm{smaller}}}{\nu}\right)^2 + \left(\frac{\partial\nu}{\partial N_i^{\mathrm{larger}}}\frac{\Delta N_i^{\mathrm{larger}}}{\nu}\right)^2 + \left(\frac{\partial\nu}{\partial\log(D_{\mathrm{p},i}^{\mathrm{larger}}/D_{\mathrm{p},i}^{\mathrm{smaller}})}\frac{\Delta\log(D_{\mathrm{p},i}^{\mathrm{larger}}/D_{\mathrm{p},i}^{\mathrm{smaller}})}{\nu}\right)^2 + \left(\frac{\partial\nu}{\partial p}\frac{\Delta p}{\nu}\right)^2}$$

$$= \sqrt{\left(\frac{\Delta N_i^{\mathrm{smaller}}}{N_i^{\mathrm{smaller}} - N_i^{\mathrm{larger}}}\right)^2 + \left(\frac{\Delta N_i^{\mathrm{larger}}}{N_i^{\mathrm{smaller}} - N_i^{\mathrm{larger}}}\right)^2 + \left(\frac{\Delta\log(D_{\mathrm{p},i}^{\mathrm{larger}}/D_{\mathrm{p},i}^{\mathrm{smaller}})}{\log(D_{\mathrm{p},i}^{\mathrm{larger}}/D_{\mathrm{p},i}^{\mathrm{smaller}})}\right)^2 + \left(\frac{\Delta p}{p}\right)^2} \tag{S3}$$

where $\Delta N_i^{\mathrm{smaller}}$ and $\Delta N_i^{\mathrm{smaller}}$ are the standard deviations of the measured concentrations, depending on the measurement case, the third term is $0.2^2$ because $\Delta(D_{\mathrm{p},i}^{\mathrm{larger}}/D_{\mathrm{p},i}^{\mathrm{smaller}})/(D_{\mathrm{p},i}^{\mathrm{larger}}/D_{\mathrm{p},i}^{\mathrm{smaller}}) = 20\%$, and $\Delta p/p$ is the relative uncertainty for $p$ depending on the particle size and is calculated with the equation

$$\frac{\Delta p}{p} = \sqrt{\left(\frac{\partial p}{\partial(L/Q)}\frac{\Delta(L/Q)}{p}\right)^2 + \left(\frac{\partial p}{\partial D_{\mathrm{p},i}}\frac{\Delta D_{\mathrm{p},i}}{p}\right)^2} \tag{S4}$$

using $\Delta(L/Q)/(L/Q) = 10\%$ and $\Delta D_{\mathrm{p},i}/D_{\mathrm{p},i} = 20\%$.

**2.2 Calculation of the uncertainties for the size distributions measured by the Nano-SMPS**

The particle number size distributions reported by the Nano-SMPS device have already went through the manufacturer's inversion algorithm. Thus, the inverse modeling of this work does not try to predict the concentration measured as a function of time measured by the CPC 3776, acting as a particle counter in the Nano-SMPS system. Instead, the inverse modeling takes only the diffusional losses in the sampling lines and the CPC 3776 detection efficiency curve into account, but not, e.g., the radioactive charger efficiency and the diffusional losses inside the device. It is partly unknown what is included in the

manufacturer's inversion algorithm, but at least the charger efficiency and the diffusional losses inside the device are included. The inversion algorithm probably includes also the CPC 3776 detection efficiency curve, $f_{CPC}$, but it is, however, included in the inverse modeling of this work because it seems that it may differ significantly from the curve reported by the manufacturer, according to Hermann et al. (2007) and Mordas et al. (2008), as presented in Fig. S2. Unfortunately, the curve for the device used in these measurements is not measured; therefore, the inversion modeling uses the one reported by Mordas et al. (2008) because it lies between the other two curves, representing an average one. The uncertainty of the detection efficiency at a specific diameter is calculated from the maximum range of variation of the detection efficiencies from these three different sources.

[Figure]

**Figure S2.** The CPC 3776 detection efficiency curves as a function of particle size reported by the manufacturer, Hermann et al. (2007), and Mordas et al. (2008). Additionally, a hypothetical curve correcting the disagreement between sub-6 nm particle size distributions measured by the Nano-SMPS and PSM+CPC system is presented.

The hypothetical detection efficiency curve presented in Fig. S2 is based on the curve reported by Mordas et al. (2008) but with different parameters. If this hypothetical curve is the actual curve of the device used, the size distributions as in Fig. 4 will be corrected to the distributions presented in Fig. S3, from which it can be seen that the size distributions measured by different devices correspond clearly better, at least for the two cases having the lowest $T_{sa}$. The PSM+CPC distribution for the case having the highest $T_{sa}$ is probably overestimated due to noise in the measured concentration because, according to Fig. 10 (c), the concentrations measured with different cut-diameters are on the same level, implying that there should not be a notable amount of particles in that size range.

Other uncertainties associated with the systematic effects, in addition to the uncertainty involved in the CPC 3776 detection efficiency curve, include the uncertainties of the charger efficiency, the diffusional losses correction, and the particle sizes interpreted by the manufacturer's inversion algorithm. The charger of the Nano-SMPS used was a TSI 3077 radioactive Kr-85 charger, which is based on charging particles bipolarly to the charge equilibrium state. The inversion algorithm uses the positive charge distribution function, $f_{charger}$, reported by Wiedensohler (1988). It is a semi-empirical function in which the mobilities and masses of positive and negative ions in the carrier gas are fitted based on the charge distribution measurements (Hussin et al., 1983; Adachi et al., 1985; Wiedensohler et al., 1986) made for particles larger than 5 nm in diameters. Alonso et al. (1997) have measured the charge distributions down to particle diameters of 2.5 nm. Unfortunately, the charger distributions from all these measurements differ, especially for the smallest particle sizes, and have thus different ion parameters, due to different particle compositions, carrier gas compositions, and the accuracies of the particle size measurements. Therefore, the charger efficiency function selected in the manufacturer's inversion algorithm may not be very accurate. Based on the differences between the results of these charge distribution measurements, the relative uncertainties of 30, 20, and 10 % for the charger efficiencies at particle diameters of 6, 10, and 20 nm, respectively, are estimated. Another factor causing uncertainty for the charger efficiency is how satisfactorily the charge distribution is developed to the equilibrium state. If the residence

[Figure]

**Figure S3.** The corrected particle size distributions as in Fig. 4 if the detection efficiency curve of the CPC 3776 would be the hypothetical curve presented in Fig. S2.

time inside the charger is too short (Alonso et al., 1997), the activity of the charger is too low (de La Verpilliere et al., 2015) (e.g, if the activity of the charger is depleted due to a long operating life), or if the particle concentration is too high compared to the ion concentration (Wiedensohler et al., 2012), the equilibrium state may not be reached and the charger efficiency is overestimated. According to the deviations of the charge distributions in the measurements of de La Verpilliere et al. (2015) from the charge distribution function of Wiedensohler (1988), the relative uncertainties of 40, 30, and 20 % for the charger efficiencies at particle diameters of 2, 6, and 10 nm, respectively, are estimated. Because the particles before the Nano-SMPS are supposedly uncharged in this work, the possible incomplete reaching of the charge equilibrium state causes that the particles are less charged than predicted. Therefore, the concentrations would be underestimated, and thus the possible error related to this is considered only negative.

The diffusional losses of the particles in the sampling lines of this work are based on the equations of Gormley and Kennedy (1948) using the $L_{\text{lines}}/Q_{\text{lines}}$ parameter as in the case of the PSM+CPC system (the relative uncertainty of 10 % for the $L_{\text{lines}}/Q_{\text{lines}}$ parameter in this case is again estimated). The correction of the diffusional losses inside the Nano-SMPS device is also based on those equations in the manufacturer's inversion algorithm. The algorithm uses an empirically fitted $L_{\text{device}}/Q_{\text{device}}$ value which included the whole route of the particles inside the device even though the route is not a perfect laminar circular tube flow, for which the analytical solution by Gormley and Kennedy (1948) is based on. Therefore, the penetration function for particles inside the device, $p_{\text{device}}$, may not be very accurate and the relative uncertainty of 10 % for the $L_{\text{device}}/Q_{\text{device}}$ parameter is estimated.

The correction factor assumed to exist in the inversion algorithm of the Nano-SMPS, to which the penetration in the sampling lines, $p_{\text{lines}}$, is added, is

$$C(D_{\text{p},i}) = \frac{1}{f_{\text{charger}}(D_{\text{p},i}) \cdot f_{\text{CPC}}(D_{\text{p},i}) \cdot p_{\text{device}}(D_{\text{p},i}, L_{\text{device}}/Q_{\text{device}}) \cdot p_{\text{lines}}(D_{\text{p},i}, L_{\text{lines}}/Q_{\text{lines}})}. \tag{S5}$$

The concentration measured with a specific DMA (Differential Mobility Analyzer) voltage at a specific time, related to the particle diameter of $D_{p,i}$ (obtained though the inversion algorithm), is multiplied with $C(D_{p,i})$ in order to obtain the size distribution in a location of $L_{\text{lines}}$ before the device, i.e., after the ejector diluter in this case. For very small particles, all the four functions in Eq. (S5) have very low value; and thus, the value of $C(D_{p,i})$ is extremely high. This is illustrated in Fig. S4 from which it can be observed that the value for sub-6 nm particles is several orders of magnitude. Very high correction factor denotes very low number of particle counts detected by the CPC at a specific diameter, and very low counts do not provide good precision due to statistics: there may be only a few randomly detected single particles or there may be even not a single detection at all during the time dedicated to that particle size, even though multiple scans have been performed for one measurement case. In the case of no or very low detection of single particles, uncertainties cannot be calculated. Because the correction factor increases very steeply with decreasing particle size, the uncertainties involved in the functions in Eq. (S5) can deviate it in high extent. Another consequence of the steep behavior of the correction factor is that if there is even a minor error in the interpreted particle diameters, the value of the correction factor can be significantly misestimated. There are several factors that can cause error to the particle diameters measured by the Nano-SMPS (Wiedensohler et al., 2012); here, the relative uncertainty of 5 % is estimated for the diameters.

[Figure]

**Figure S4.** Nano-SMPS correction factor, as in Eq. (S5), used to correct the measured particle concentrations in the data inversion.

The uncertainty associated with both the systematic and random effects for the size distributions, $\nu(D_{p,i})$, can be calculated with the equation

$$\frac{\Delta\nu}{\nu} = \sqrt{\left(\frac{\Delta\nu'}{\nu'}\right)^2 + \left(\frac{\Delta C}{C}\right)^2} \tag{S6}$$

where $\Delta\nu'/\nu'$ is the relative standard deviation of the size distributions at the particle diameter of $D_{p,i}$ output by the device and $\Delta C/C$ is the relative uncertainty of the Nano-SMPS correction factor. $\Delta\nu'/\nu'$ represents uncertainties associated with the random effects arisen from the noise in the measured concentrations caused by the instability of the particle generation, as in the case of the PSM+CPC system, but also from low precision in measuring particles sizes having a very low overall detection

efficiency and particle sizes having low concentration in the measured case. $\Delta C/C$ is calculated with the equation

$$\frac{\Delta C}{C} = \sqrt{\left(\frac{\partial C}{\partial f_{\text{charger}}}\frac{\Delta f_{\text{charger}}}{C}\right)^2 + \left(\frac{\partial C}{\partial f_{\text{CPC}}}\frac{\Delta f_{\text{CPC}}}{C}\right)^2 + \left(\frac{\partial C}{\partial p_{\text{device}}}\frac{\Delta p_{\text{device}}}{C}\right)^2 + \left(\frac{\partial C}{\partial p_{\text{lines}}}\frac{\Delta p_{\text{lines}}}{C}\right)^2 + \left(\frac{\partial C}{\partial D_{\text{p},i}}\frac{\Delta D_{\text{p},i}}{C}\right)^2}$$

$$= \sqrt{\left(\frac{\Delta f_{\text{charger}}}{f_{\text{charger}}}\right)^2 + \left(\frac{\Delta f_{\text{CPC}}}{f_{\text{CPC}}}\right)^2 + \left(\frac{\Delta p_{\text{device}}}{p_{\text{device}}}\right)^2 + \left(\frac{\Delta p_{\text{lines}}}{p_{\text{lines}}}\right)^2 + \left(\frac{\partial C}{\partial D_{\text{p},i}}\frac{\Delta D_{\text{p},i}}{C}\right)^2} \tag{S7}$$

where $\Delta f_{\text{charger}}/f_{\text{charger}}$ and $\Delta f_{\text{CPC}}/f_{\text{CPC}}$ are the relative uncertainties for $f_{\text{charger}}$ and $f_{\text{CPC}}$ having the values mentioned before, $\Delta p_{\text{device}}/p_{\text{device}}$ and $\Delta p_{\text{lines}}/p_{\text{lines}}$ are the relative uncertainties for the penetration efficiencies, $p_{\text{device}}$ and $p_{\text{lines}}$, and the last term represents the relative uncertainty of $C$ caused by the uncertainty of particle diameters. $\Delta p_{\text{device}}/p_{\text{device}}$ and $\Delta p_{\text{lines}}/p_{\text{lines}}$ depend on the particle diameter and are calculated with the equations

$$\frac{\Delta p_{\text{device}}}{p_{\text{device}}} = \frac{\partial p_{\text{device}}}{\partial (L_{\text{device}}/Q_{\text{device}})} \cdot \frac{\Delta(L_{\text{device}}/Q_{\text{device}})}{p_{\text{device}}}$$

$$\frac{\Delta p_{\text{lines}}}{p_{\text{lines}}} = \frac{\partial p_{\text{lines}}}{\partial (L_{\text{lines}}/Q_{\text{lines}})} \cdot \frac{\Delta(L_{\text{lines}}/Q_{\text{lines}})}{p_{\text{lines}}} \tag{S8}$$

which differ from Eg. (S4) by missing the effect of the particle diameter because that effect is included in the last term of Eq. (S7). The last term represents the total effect of the uncertainty of particle diameter on the uncertainty of $C$ because particle diameter is involved in all the other four terms.

**2.3 Calculated uncertainties for the size distributions**

The relative uncertainties for the size distributions between 2 and 10 nm after the ejector diluter caused by the uncertainties associated with the different effects involved in the size distribution measurements are presented in Tab. S1. For the PSM+CPC system, the most significant relative uncertainties associated with the systematic effects arise from the uncertainty of the cut-diameters (12 ... 30 %), partly due to correcting the diffusional losses in sampling lines needed in backwards-correcting the measured distributions to represent the distributions after the ejector diluter. For the Nano-SMPS, the uncertainty of the charger efficiency plays a major role in the relative uncertainties associated with the systematic effects (40 ... 66 %), but the uncertainty of the CPC 3776 detection efficiency curve has also a significant role for the smallest particles (55 % for 3.7 nm). The relative uncertainties decrease steeply when measuring particles sized 10 nm or larger using the Nano-SMPS. Both devices are, in theory, capable in measuring the size distribution at 3.7 nm, but the uncertainties with the Nano-SMPS are clearly higher compared to the PSM+CPC system. Therefore, the PSM+CPC system suits better in measuring near that diameter.

Uncertainties associated with the random effects caused by the noise in the measured concentrations due to instability in particle generation for the both devices and by low counting statistics of the Nano-SMPS also have significant effects if there is not a notable amount of concentration in a specific size range. In the case with $D_{\bar{m}} = 3.6\,\text{nm}$, there is a notable amount of concentration in the PSM+CPC size range, and thus, the relative uncertainties are relatively low (22 ... 61 %) for the PSM+CPC system. For the Nano-SMPS, the relative standard deviation of the size distribution for the 6 nm particles is 72 %, which is, however, the particle size measured with the highest precision in that case: larger particles are inexistent and smaller particles are not detected. In the case with $D_{\bar{m}} = 19\,\text{nm}$, the Nano-SMPS suits well in measuring the size distribution at the particle size of 10 nm (the relative standard deviation of 10 % originating mainly from the instability in the particle generation) and also relatively well at the particle size of 6 nm (the relative standard deviation of 24 %), but the uncertainties increase with the particles smaller than 6 nm. Conversely, the PSM+CPC system has high uncertainties because the concentration in the PSM+CPC size range is so low that the difference between the concentrations measured with different cut-diameters are smaller than the standard deviation of the concentrations (see Fig. 10), which is always a problem with the PSM having a cumulative nature in measuring concentrations, if the cut-diameters of the adjacent saturator flow rates are too near or the measured signal is too unstable. This issue can be overcome by skipping the data measured with the adjacent cut-diameters or even by considering only the data measured with the smallest and with the largest cut-diameter. However, while the precision will be higher in this alternative method, the information on the shape of the size distribution in that size range will diminish.

**Table S1.** The relative uncertainties (in percents) for the size distributions after the ejector diluter, $\Delta\nu/\nu$ (%), for the selected particle diameters. The first seven lines represent the relative uncertainties associated with the systematic effects and are thus independent of the measurement cases. The last two lines represent the relative uncertainties associated with the random effects for two selected measurement cases having small and large particles.

| Device $D_{\mathrm{p}}$ | PSM+CPC | | Nano-SMPS | | |
|---|---|---|---|---|---|
| | 2 nm | 3.7 nm | 3.7 nm | 6 nm | 10 nm |
| Diffusional losses in sampling lines $\left(\frac{\Delta(L_{\mathrm{lines}}/Q_{\mathrm{lines}})}{L_{\mathrm{lines}}/Q_{\mathrm{lines}}}=10\,\%\right)$ | 8 | 3 | 4 | 2 | 1 |
| Diffusional losses in sampling lines $\left(\frac{\Delta D_{\mathrm{p},i}}{D_{\mathrm{p},i}}=20\,\%\right)$ | 30 | 12 | | | |
| PSM detection efficiency $\left(\frac{\Delta(D_{\mathrm{p},i}^{\mathrm{larger}}/D_{\mathrm{p},i}^{\mathrm{smaller}})}{D_{\mathrm{p},i}^{\mathrm{larger}}/D_{\mathrm{p},i}^{\mathrm{smaller}}}=20\,\%\right)$ | 20 | 20 | | | |
| Kr-85 charger efficiency | | | 66 | 61 | 40 |
| CPC 3776 detection efficiency | | | 55 | 8 | 0.7 |
| Diffusional losses inside the DMA $\left(\frac{\Delta(L_{\mathrm{lines}}/Q_{\mathrm{lines}})}{L_{\mathrm{lines}}/Q_{\mathrm{lines}}}=10\,\%\right)$ | | | 16 | 7 | 3 |
| Nano-SMPS correction factor $\left(\frac{\Delta D_{\mathrm{p},i}}{D_{\mathrm{p},i}}=5\,\%\right)$ | | | 32 | 17 | 10 |
| Random effects in $D_{\bar{m}}=3.6$ nm case (102 °C) | 61 | 22 | _[a] | 72 | _[a] |
| Ramdom effects in $D_{\bar{m}}=19$ nm case (157.2 °C) | $\infty$[b] | 250 | _[a] | 24 | 10 |

[a] Cannot be calculated due to insufficient particle counts.
[b] For this point, $\nu(2\,\mathrm{nm})=0$ but $\Delta\nu$ is a non-zero number due to the standard deviations of $N_i^{\mathrm{smaller}}$ and $N_i^{\mathrm{smaller}}$.

The error bars representing the uncertainties associated with both the systematic and random effects for the size distributions shown in Fig. 4 are presented in Fig. S5. By considering the error bars, the distributions from the both devices agree for the cases in panes (a) and (c), whereas the case in pane (b) has still some disagreement implying that other sources of uncertainty than accounted here can be involved in the measurements using these devices. According to the error bars near the particle size of 4 nm connecting the two size distributions, the PSM+CPC system provides more reliable results in the cases in panes (a) and (b). Conversely, in the case in pane (c), the Nano-SMPS provides more reliable results because, although there are two points in the PSM+CPC distribution, the alternative method shows no particles at all. However, the error bars for the alternative method are high; thus, the existence of particles in the PSM+CPC size range is still probable. Nevertheless, the fraction of particles in that size range compared to the total particle count is, definitely, some orders of magnitude smaller than in the cases in panes (a) and (b). The distributions and their uncertainties near the particle size of 4 nm, where the distributions from the both devices are available, are decided to keep separated here due to high systematic error possible in the Nano-SMPS data, although by combining the distributions would cause lower overall uncertainties. Figure S6 presents the error bars for the distributions shown in Fig. 11.

In conclusion, the reliability of our Nano-SMPS system was low for the particle sizes smaller than ∼10 nm, for the most part due to the uncertainties involved in the radioactive charger efficiency and the CPC 3776 detection efficiency. Wiedensohler et al. (2012) have performed an intercomparison of several mobility particle sizers, in which the different devices provided a good agreement for the particle sizes larger than ∼15 nm but had significant disagreements for the smaller particle sizes, without explanation. Due to the difficulties that the Nano-SMPS has in determining the size distribution reliably in sub-10 nm diameter range, in the cases studied here and elsewhere, we found that the PSM+CPC system was better suited in determining the size distribution in that particle size range, or at least in determining the total number concentration of particles larger than ∼1 nm.

[Figure]

**Figure S5.** The measured size distributions for the measurement cases having the $T_{sa}$ of (a) 102 °C, (b) 135.5 °C, and (c) 157.2 °C, as shown in Fig. 4, with the error bars representing the uncertainties associated with both the systematic and random effects. The alternative PSM+CPC distributions represent the distributions using only the concentrations measured with the smallest and the largest cut-diameters, in order to increase the precision. The green shaded areas denote the error bars for the distributions from the alternative method. The error bars for the particle sizes obtained from the Nano-SMPS ($\pm 5\%$) are not shown for clarity.

[Figure]

**Figure S6.** The simulated and measured size distributions for the measurement cases having the $T_{sa}$ of (a) 102 °C, (b) 135.5 °C, and (c) 157.2 °C, as shown in Fig. 11, with the error bars representing the uncertainties associated with both the systematic and random effects. The error bars for the particle sizes obtained from the Nano-SMPS ($\pm 5\,\%$) are not shown for clarity.

**2.4 Calculation of the uncertainties for the diameters with the average mass**

The diameter with the average mass of a distribution is calculated by

$$
D_{\bar{m}} = \left( \frac{M'}{N} \right)^{\frac{1}{3}} = \left( \frac{\sum\limits_{i}^{\mathrm{PC}} \nu(D_{\mathrm{p},i}) \cdot \mathrm{d}\log D_{\mathrm{p},i} \cdot D_{\mathrm{p},i}^3 + \sum\limits_{i}^{\mathrm{NS}} \nu(D_{\mathrm{p},i}) \cdot \mathrm{d}\log D_{\mathrm{p},i} \cdot D_{\mathrm{p},i}^3}{\sum\limits_{i} \nu(D_{\mathrm{p},i}) \cdot \mathrm{d}\log D_{\mathrm{p},i}} \right)^{\frac{1}{3}} = \left( \frac{M'_{\mathrm{PC}} + M'_{\mathrm{NS}}}{N} \right)^{\frac{1}{3}} \tag{S9}
$$

where $M'$ is the third moment of the distribution, $M'_{\mathrm{PC}}$ and $M'_{\mathrm{NS}}$ are the parts of the third moment from the PSM+CPC data and from the Nano-SMPS data, respectively, and $N$ is the total number concentration.

The relative uncertainty for $D_{\bar{m}}$ can be calculated with the equation

$$
\frac{\Delta D_{\bar{m}}}{D_{\bar{m}}} = \sqrt{\left( \left. \frac{\Delta D_{\bar{m}}}{D_{\bar{m}}} \right|_{\Delta D_{\mathrm{p}}} \right)^2 + \left( \left. \frac{\Delta D_{\bar{m}}}{D_{\bar{m}}} \right|_{\Delta \nu_{\mathrm{s}}} \right)^2 + \left( \left. \frac{\Delta D_{\bar{m}}}{D_{\bar{m}}} \right|_{\Delta \nu_{\mathrm{r}}} \right)^2} \tag{S10}
$$

where the first term represents the relative uncertainty caused by the uncertainty of the interpreted particle diameters, the second term the relative uncertainty caused by the uncertainty associated with the systematic effects for the number size distribution, and the last term the relative uncertainty caused by the uncertainty associated with the random effects for the number size distribution.

The first term in Eq. (S10) is separated to the effects of the PSM+CPC system and of the Nano-SMPS, respectively:

$$
\left. \frac{\Delta D_{\bar{m}}}{D_{\bar{m}}} \right|_{\Delta D_{\mathrm{p}}} = \sqrt{\left( \left. \frac{\Delta D_{\bar{m}}}{D_{\bar{m}}} \right|_{\Delta D_{\mathrm{p},\mathrm{PC}_1}} \right)^2 + \left( \left. \frac{\Delta D_{\bar{m}}}{D_{\bar{m}}} \right|_{\Delta D_{\mathrm{p},\mathrm{NS}_1}} \right)^2}. \tag{S11}
$$

Because particle diameters are dependent variables for a specific device, i.e., if one diameter is shifted to a direction, other diameters are most probably shifted to the same direction and with almost the same magnitude, the diameters in Eq. (S9) are separated to dependent and independent parts:

$$
\begin{aligned}
D_{\mathrm{p},i} &= D_{\mathrm{p},\mathrm{PC}_1} \cdot D'_{\mathrm{p},i} \\
D_{\mathrm{p},i} &= D_{\mathrm{p},\mathrm{NS}_1} \cdot D'_{\mathrm{p},i}.
\end{aligned} \tag{S12}
$$

where $D_{\mathrm{p},\mathrm{PC}_1}$ and $D_{\mathrm{p},\mathrm{NS}_1}$ denote the smallest diameters measured by the PSM+CPC system and by the Nano-SMPS, respectively, and $D'_{\mathrm{p},i}$ is a dimensionless variable denoting the ratios of all other diameters to the smallest diameter. Hence, the third moments can be expressed as

$$
M'_{\mathrm{PC}} = \sum\limits_{i}^{\mathrm{PC}} \nu(D_{\mathrm{p},i}) \cdot \mathrm{d}\log D_{\mathrm{p},i} \cdot D_{\mathrm{p},i}^3 = \sum\limits_{i}^{\mathrm{PC}} \nu(D_{\mathrm{p},i}) \cdot \mathrm{d}\log D_{\mathrm{p},i} \cdot (D_{\mathrm{p},\mathrm{PC}_1} \cdot D'_{\mathrm{p},i})^3 = D_{\mathrm{p},\mathrm{PC}_1}^3 \cdot \sum\limits_{i}^{\mathrm{PC}} \nu(D_{\mathrm{p},i}) \cdot \mathrm{d}\log D_{\mathrm{p},i} \cdot D'^3_{\mathrm{p},i}
$$

$$
M'_{\mathrm{NS}} = \sum\limits_{i}^{\mathrm{NS}} \nu(D_{\mathrm{p},i}) \cdot \mathrm{d}\log D_{\mathrm{p},i} \cdot D_{\mathrm{p},i}^3 = \sum\limits_{i}^{\mathrm{NS}} \nu(D_{\mathrm{p},i}) \cdot \mathrm{d}\log D_{\mathrm{p},i} \cdot (D_{\mathrm{p},\mathrm{NS}_1} \cdot D'_{\mathrm{p},i})^3 = D_{\mathrm{p},\mathrm{NS}_1}^3 \cdot \sum\limits_{i}^{\mathrm{NS}} \nu(D_{\mathrm{p},i}) \cdot \mathrm{d}\log D_{\mathrm{p},i} \cdot D'^3_{\mathrm{p},i}. \tag{S13}
$$

The relative uncertainties in Eq. (S11) can now be calculated by

$$
\begin{aligned}
\left. \frac{\Delta D_{\bar{m}}}{D_{\bar{m}}} \right|_{\Delta D_{\mathrm{p},\mathrm{PC}_1}} &= \frac{\partial D_{\bar{m}}}{\partial D_{\mathrm{p},\mathrm{PC}_1}} \cdot \frac{\Delta D_{\mathrm{p},\mathrm{PC}_1}}{D_{\bar{m}}} = \frac{1}{3} \frac{D_{\bar{m}}}{M'} \cdot \frac{\partial M'_{\mathrm{PC}}}{\partial D_{\mathrm{p},\mathrm{PC}_1}} \cdot \frac{\Delta D_{\mathrm{p},\mathrm{PC}_1}}{D_{\bar{m}}} = \frac{M'_{\mathrm{PC}}}{M'} \cdot \frac{\Delta D_{\mathrm{p},\mathrm{PC}_1}}{D_{\mathrm{p},\mathrm{PC}_1}} \\
\left. \frac{\Delta D_{\bar{m}}}{D_{\bar{m}}} \right|_{\Delta D_{\mathrm{p},\mathrm{NS}_1}} &= \frac{\partial D_{\bar{m}}}{\partial D_{\mathrm{p},\mathrm{NS}_1}} \cdot \frac{\Delta D_{\mathrm{p},\mathrm{NS}_1}}{D_{\bar{m}}} = \frac{1}{3} \frac{D_{\bar{m}}}{M'} \cdot \frac{\partial M'_{\mathrm{NS}}}{\partial D_{\mathrm{p},\mathrm{NS}_1}} \cdot \frac{\Delta D_{\mathrm{p},\mathrm{NS}_1}}{D_{\bar{m}}} = \frac{M'_{\mathrm{NS}}}{M'} \cdot \frac{\Delta D_{\mathrm{p},\mathrm{NS}_1}}{D_{\mathrm{p},\mathrm{NS}_1}}.
\end{aligned} \tag{S14}
$$

As 20 and 5 % relative uncertainties for the diameters for the PSM+CPC system and for the Nano-SMPS, respectively, were estimated, Eq. (S11) becomes

$$\left. \frac{\Delta D_{\bar{m}}}{D_{\bar{m}}} \right|_{\Delta D_{\mathrm{p}}} = \frac{1}{M'} \sqrt{\left( M'_{\mathrm{PC}} \cdot 0.2 \right)^2 + \left( M'_{\mathrm{NS}} \cdot 0.05 \right)^2}. \tag{S15}$$

The relative uncertainties associated with the systematic and random effects are separated in the last two terms in Eq. (S10) because the possible errors due to the systematic effects for all size bins are presumably to the same direction and in almost of the same magnitude and the possible errors due to the random effects are randomly directed between different size bins because they are measured at different times. The number size distributions can be separated to the parts involving the sources of uncertainties associated with the systematic ($\Delta\nu_{\mathrm{s}}$) and with the random ($\Delta\nu_{\mathrm{r}}$) effects, respectively, using $\nu = \nu_{\mathrm{s}} \cdot \nu_{\mathrm{r}}$. The systematic effects for the uncertainty of $\nu_{\mathrm{s}}$ involve independent variables between the different devices, but dependent variables between the different size bins of a specific device. Hence, the second term in Eq. (S10) is further separated to the PSM+CPC system and to the Nano-SMPS, respectively, using

$$\left. \frac{\Delta D_{\bar{m}}}{D_{\bar{m}}} \right|_{\Delta\nu_{\mathrm{s}}} = \sqrt{ \left( \left. \frac{\Delta D_{\bar{m}}}{D_{\bar{m}}} \right|_{\Delta\nu_{\mathrm{s,PC}}} \right)^2 + \left( \left. \frac{\Delta D_{\bar{m}}}{D_{\bar{m}}} \right|_{\Delta\nu_{\mathrm{s,NS}}} \right)^2 }$$
$$= \sqrt{ \left[ \frac{D_{\bar{m}} \left( \nu + \Delta\nu_{\mathrm{s,PC}} \right) - D_{\bar{m}} \left( \nu - \Delta\nu_{\mathrm{s,PC}} \right)}{2 D_{\bar{m}}} \right]^2 + \left[ \frac{D_{\bar{m}} \left( \nu + \Delta\nu_{\mathrm{s,NS}} \right) - D_{\bar{m}} \left( \nu - \Delta\nu_{\mathrm{s,NS}} \right)}{2 D_{\bar{m}}} \right]^2 }. \tag{S16}$$

The last term in Eq. (S10), related to the relative uncertainties associated with the random effects, is calculated by

$$\left. \frac{\Delta D_{\bar{m}}}{D_{\bar{m}}} \right|_{\Delta\nu_{\mathrm{r}}} = \sqrt{ \sum_i \left[ \frac{\partial D_{\bar{m}}}{\partial \nu_{\mathrm{r}}(D_{\mathrm{p},i})} \cdot \frac{\Delta\nu_{\mathrm{r}}(D_{\mathrm{p},i})}{D_{\bar{m}}} \right]^2 } = \sqrt{ \sum_i \left[ \left( \frac{1}{3M'} \frac{\partial M'}{\partial \nu_{\mathrm{r}}(D_{\mathrm{p},i})} - \frac{1}{3N} \frac{\partial N}{\partial \nu_{\mathrm{r}}(D_{\mathrm{p},i})} \right) \Delta\nu_{\mathrm{r}}(D_{\mathrm{p},i}) \right]^2 }$$
$$= \frac{1}{3} \sqrt{ \sum_i \left[ \left( \frac{\mathrm{d}\log D_{\mathrm{p},i} \cdot D_{\mathrm{p},i}^3}{M'} - \frac{\mathrm{d}\log D_{\mathrm{p},i}}{N} \right) \Delta\nu_{\mathrm{r}}(D_{\mathrm{p},i}) \right]^2 }. \tag{S17}$$

The calculated error bars representing the uncertainties associated with both the systematic and random effects for the diameters with the average mass are presented in Fig. 13.